# Reviews and syntheses: Benthic foraminifera and gromiids from oxygen depleted environments - Survival strategies, biogeochemistry and trophic interactions

Nicolaas Glock[1]

[1]: Institute for Geology, University of Hamburg, Bundesstraße 55, 20146 Hamburg, Germany

*Correspondence to*: Nicolaas Glock (nicolaas.glock@uni-hamburg.de)

**Abstract**

The oceans are losing oxygen ($O_2$) and oxygen minimum zones are expanding, due to climate warming (lower $O_2$ solubility) and eutrophication related to agriculture. This trend is challenging for most marine taxa that are not well adapted to $O_2$ depletion. For other taxa this trend might be advantageous, because they can withstand low $O_2$ concentrations or thrive under $O_2$ depleted or even anoxic conditions. Benthic foraminifera are a group of protists that include taxa with adaptations to partly extreme environmental conditions. Several species possess adaptations to $O_2$ depletion that are rare amongst eukaryotes and these species might benefit from ongoing ocean deoxygenation. In addition, since some foraminifera can calcify even under anoxic conditions they are important archives for paleoceanographic reconstruction in $O_2$ depleted environments. This paper reviews the current state of knowledge about foraminifera from low $O_2$ environments. Recent advances to understand specific survival strategies of foraminifera to withstand $O_2$ depletion are summarized and discussed. These adaptations include an anaerobic metabolism, heterotrophic denitrification, symbiosis with bacteria, kleptoplasty and dormancy and have a strong impact on their preferred microhabitat in the sediments, especially the ability to denitrify by some benthic foraminiferal species. Benthic foraminifera also differ regarding their trophic strategies which has additional impact on the selection of their microhabitat. For example, some species are strict herbivores that feed exclusively on fresh phytodetritus and live close to the sediment surface, while some species are non-selective detrivores that occupy intermediate to deep infaunal habitats. There is evidence that foraminifers have the capacity of phagocytosis, even under anoxia, and some foraminiferal species, which can withstand low $O_2$ conditions, seem to prey on meiofauna. Also, due to their high abundances in $O_2$ depleted environments and their metabolic adaptations, benthic foraminifera are key players in marine nutrient cycling, especially within the marine N and P cycles. This review summarizes the denitrification rates for the species that are known to denitrify and the intracellular nitrate concentrations of the species that are known to intracelullary store nitrate. Finally, equations are provided that can be used to estimate the intracellular nutrient storage and denitrification rates of foraminifera and might be integrated into biogeochemical models.

**1 Introduction**

More than two decades have passed since Bernhard and Sen Gupta (1999) provided a comprehensive review about the history of research on foraminifera from $O_2$ depleted environments. About a decade later, Koho and Piña-Ochoa (2012) published another overview about benthic foraminifera as inhabitants in low-$O_2$ habitats, mainly focusing on the species distribution in different environments and the different depth layers in the sediment. They also summarized the early work on foraminiferal denitrification, kleptoplasty and evidence for bacterial symbiosis. Nevertheless, advances in methods to analyze the metabolic rates, intracellular nitrate storage and molecular genetics of foraminifera have changed our understanding of strategies such as an anaerobic metabolism that help them to withstand $O_2$ depletion. This paper aims to summarize these developments, mainly focusing on benthic foraminifera. For the discussion about life in habitats, where $O_2$ is scarce or absent it is important to define the range of $O_2$ concentration for terms such as anoxia, hypoxia, suboxic or oxic conditions. The concentration range for these terms varies with literature. To avoid confusion, this review will only use the following definitions from literature:

**Anoxia** usually indicate the complete absence of $O_2$ ([$O_2$] = 0 µM; Diaz, 2016)

**Suboxic conditions** indicate habitats, where $O_2$ is low enough that denitrification and Mn- and Fe-reduction but sulfide concentrations are still low, due to the absence of sulfate reduction ([$O_2$] ~ 1-10 µM; Oakley et al., 2007)

**Hypoxia** in aquatic environments indicate habitats where $O_2$ is present but the $O_2$ saturation is less than 30%, since most fish cannot survive below 30% saturation ([$O_2$] < 62.5 µM Levin et al., 2009).

**Low $O_2$** or **$O_2$ depleted** habitats will summarize all environments that fulfil one of the above definitions (i.e., every environment where [$O_2$] is < 62.5 µM)

Knowledge about planktic foraminifera from $O_2$ depleted habitats is scarce compared to the knowledge about benthic foraminifera. Nonetheless, at least two species *Globorotaloides hexagonus* and *Hastigerina parapelagica* are known to live in pelagic oxygen minimum zones (OMZs) (Davis et al., 2021). As a result, *G. hexagonus* has proven to be a valuable paleo-indicator for the presence of pelagic OMZs during the Pliocene (Davis et al., 2023). Benthic foraminifera from low $O_2$ environments have also been established as an invaluable archive for paleoceanography. However, this review will touch on summarizing redox proxies based on benthic foraminifera only briefly, since there is work in progress to give a comprehensive review about proxies for $O_2$

concentrations in paleoceanography (Hoogakker et al., in prep). Due to their ability to precipitate their calcitic tests even under anoxic conditions, fossil benthic foraminifera became routine tools in paleoceanography to reconstruct past redox conditions (Nardelli et al., 2014; Orsi et al., 2020). Some morphological adaptations are very common for benthic foraminifera that thrive in $O_2$ depleted habitats. Small, more elongated and flattened morphologies are often characteristic for $O_2$ depletion, while more spherical forms can indicate oxygenated conditions (Bernhard, 1986; Bernhard et al., 1997). In addition, high porosity and thin test walls seem to be characteristic for foraminifera that live in low $O_2$ environments (Kaiho, 1994). The porosity, including pore size and pore density, of foraminiferal tests recently received more attention as possible paleoceanographic tool. Different foraminiferal species seem to adapt their pore characteristics in a different way to environmental conditions. *Cibicides* spp. for example mainly thrive in well oxygenated environments (Mackensen et al., 1995) and the porosity in epifaunal *Cibicides* spp. and *Planulina* spp. is significantly negatively correlated to the $O_2$ concentrations in the bottom water (Rathburn et al., 2018; Glock et al., 2022). If $O_2$ is too depleted, these foraminifers increase their porosity to optimize the $O_2$ uptake. Furthermore, the mechanism of biomineralisation in foraminifera can preserve the chemical signature of ambient seawater in their test calcite. These species precipitate their test calcite directly from vacuolized seawater (Erez, 2003; de Nooijer et al., 2014; Toyofuku et al., 2017) and thus the chemical composition of the test calcite reflects the chemical composition of the surrounding water in their habitats. Different element/Ca ratios are used as proxy for various parameters. Over the past decades several redox sensitive element/Ca ratios in foraminiferal calcite were identified as potential $O_2$ proxies, where Mn/Ca (Reichart et al., 2003; Barras et al., 2018; Brinkmann et al., 2021) and I/Ca (e.g., Zhou et al., 2014, 2022; Lu et al., 2016; Glock et al., 2019d; Winkelbauer et al., 2021; Cook et al., 2022) are amongst the most prominent examples. The offset of the stable carbon isotope fractionation ($\delta^{13}C$) between the tests of epifaunal and deep infaunal benthic foraminifera can also be used as a quantitative $[O_2]_{BW}$ proxy (e.g., McCorkle and Emerson, 1988; Schmiedl and Mackensen, 2006; Hoogakker et al., 2014, 2018). Finally, species compositions of benthic foraminifera assemblages are used to reconstruct past environmental conditions. Kaiho et al. (1994) developed the first benthic foraminifera $O_2$ index (BFOI). Further development of this index is still going on with recent developments by Tetard et al., 2021 and Kranner et al., 2022.

The first part of the present paper reviews recent advances in our understanding of the diverse strategies that foraminifera use to withstand $O_2$ depletion, focusing mainly on denitrification, dormancy and kleptoplasty. The part about foramiferal denitrification also incorporates denitrification into the conceptual TROX model of Jorissen et al. (1995). The TROX model explains the sediment microhabitats of benthic foraminifera in terms of an interplay in the supply of $O_2$ and non-refractory organic matter that can be used as food. The next section briefly summarizes the knowledge about ecological and trophic interactions of foraminifera from $O_2$ depleted environments. Finally, the role of foraminifera in marine biogeochemical cycling will be discussed, with a focus on nitrogen and phosphorous cycling.

**2.1 Survival strategies**

Some benthic foraminiferal species have very specific adaptations that provide the opportunity either to thrive in anoxia or at least to survive periods of $O_2$ depletion (see examples in fig. 1).

2.1.1 Foraminiferal denitrification

More than a decade ago first evidence emerged that some foraminifera from $O_2$ depleted environments are able to perform complete denitrification (Risgaard-Petersen et al., 2006). Heterotrophic denitrification describes the step-by-step reduction of nitrate ($NO_3^-$) to inert $N_2$ gas (Eq. 1 according to Jorgensen, 2006 & Fig. 2).

Eq.1: $$5\,[CH_2O] + 4\,NO_3^- \rightarrow 2\,N_2 + 4\,HCO_3^- + CO_2 + 3\,H_2O$$

,where $[CH_2O]$ symbolizes organic matter of unspecified composition.

Heterotrophic denitrification provides energy to an organism for oxidative phosphorylation in a similar way as $O_2$ respiration. The $\Delta G^0$ for heterotrophic denitrification per mol carbon at a pH of 7 is -453 kJ mol$^{-1}$, which is slightly less efficient $O_2$ respiration ($\Delta G^0$ = -479 kJ mol$^{-1}$ according to Jorgensen, 2006).

The discovery of foraminiferal denitrification by Risgaard-Petersen et al. (2006) was also the first evidence for complete denitrification in eukaryotic cells in general and it showed that they likely take up $NO_3^-$ from the surrounding pore water and store it within intracellular seawater vacuoles. Nevertheless, no later study

could actually proof a bonafide "complete" denitrification pathway in foraminifera and the eukaryotic foraminiferal denitrification pathway is today considered to be incomplete (Woehle et al., 2018; Orsi et al, 2020; Gomaa et al., 2021; see discussion below). Other eukaryotes that are known to perform incomplete denitrification are the primitive eukaryote *Loxodes* (Finlay et al., 1983) and two species of fungi (Usuda et al., 1995).

Four years after the study by Risgaard-Petersen et al. (2006), Pina-Ochoa et al. (2010b) documented that intracellular $NO_3^-$ storage and denitrification are not an exception, limited to a few specialized foraminiferal species, but actually a widespread phenomenon. Within a couple of years more studies either quantified denitrification rates or the intracellular $NO_3^-$ storage of various foraminifera and gromiid species (Høgslund et al., 2008; Glud et al., 2009; Piña-Ochoa et al., 2010b, a; Koho et al., 2011; Bernhard et al., 2012b). The intracellular $NO_3^-$ storage can reach concentrations up to 567 mM in gromiids (Piña-Ochoa et al., 2010b) and experiments with isotopically labeled $NO_3^-$ showed that *Globobulimina turgida* takes up $NO_3^-$ in a similar rate, independently of the presence or magnitude of the intracellular $NO_3^-$ pool (Koho et al., 2011). It has been hypothesized that at least some denitrifiying foraminifera seem to take up $NO_3^-$ through the pores in their tests and the pore-density (# of pores per area) of some denitrifying species, such as *Bolivina spissa* turned out to be a promising proxy for quantitative $NO_3^-$ reconstructions (Glock et al., 2011, 2018). However, not all benthic foraminifera are able to denitrify, even if they live in environments that are periodically exposed to anoxia such as representatives of the intertidal species morphogroup *Ammonia tepida* (either *Ammonia venata, Ammonia aberdoveyensis* or *Ammonia confertitesta* according to Hayward et al., 2021), which neither store $NO_3^-$ nor show any denitrification activities (Piña-Ochoa et al., 2010b). Some foraminifera from the Bering Sea have been shown to store $NO_3^-$ but did not denitrify in incubation experiments (Langlet et al., 2020). These species include *Nonionella pulchella, Uvigerina peregrina* and *Bolivinellina pseudopunctata*. Also, the $NO_3^-$ storage in *U. peregrina* shows a high variability, depending on the environment. Individuals of *U. peregrina* from the Bay of Biscay lack a significant $NO_3^-$ storage, while *U. peregrina* from the North Sea and the Bering Sea both show intracellular $NO_3^-$ enrichments (Piña-Ochoa et al., 2010b, Langlet et al., 2021). Other *Uvigerina* and *Nonionella* species have been shown to denitrify (Risgaard-Petersen et al., 2006; Høgslund et al., 2008; Piña-Ochoa et al., 2010b; Glock et al., 2019b; Gomaa et al., 2021). Many milliolids and allogromiids, several intertidal rotaliid species but also some other rotaliids and textulariids completely lack an intracellular $NO_3^-$ storage (Piña-Ochoa et al., 2010b).

The observations that some species store $NO_3^-$ and denitrify in some environments and in others not might have two reasons. One reason could be that these species belong to an opportunistic group of foraminifera that can well adapt to both oxygenated environments where they respire $O_2$ and do not denitrify and $O_2$ depleted environments where they switch to denitrification. The other reason could be that some of these foraminifera belong to morphogroups that are identified as a single species but indeed are a mixture of cryptic and pseudocryptic species that include denitrifying and non-denitrifying species. An example for such a morphogroup that has recently had a revision is *A. tepida*. This morphogroup includes three species (*A. venata, A. aberdoveyensis* or *A. confertitesta*) that now can be morphologically distinguished (Richirt et al., 2019; Hayward et al., 2021). A similar case concerns the morphogroup *Nonionella stella*, where representatives have been found to denitrify (Høgslund et al., 2008; Choquel et al., 2021). The morphogroup *N. stella* also consists out of several cryptic to pseudocryptic species (Deldicq et al., 2019). The situation might be similar with other *Nonionella* species and the widespread species *U. peregrina*.

There is strong evidence for symbiosis between foraminifera and prokaryotes in many hosts from $O_2$ depleted environments, which most likely are an adaptation to survive within the steep geochemical gradients close to the oxic/anoxic boundary (Bernhard et al., 2000; Bernhard, 2003; Bernhard et al., 2006; Nomaki et al., 2014; Bernhard et al., 2018). Most of the observed prokaryotic associates are endobionts within the foraminiferal cytoplasm but some are ectobionts that often are observed close to the pores in the foraminiferal shell (Bernhard et al., 2001, 2010a, 2018). For about a decade after the first discovery of foraminiferal denitrification it remained unclear if foraminifera indeed denitrify themselves, or if the bacterial symbionts are responsible for the denitrification. Evidence came up for both hypotheses. Bernhard et al. (2012b) showed that *Bolvina argentaea* consumed its intracellular $NO_3^-$ storage in $O_2$ free incubations even after a very harsh treatment with antibiotics, which indicates that this species can denitrify even, when the activity of potential bacterial symbionts would be inhibited. Other studies showed that bacterial endobionts likely perform denitrification in some allogromiid foraminifera and gromiid species (Bernhard et al., 2012a; Høgslund et al., 2017). Gromiida are a separate group of protists within the Rhizaria and closely related to foraminifera. With the recent advances in molecular biology, however, it became possible to analyze the transcriptome of denitrifying foraminifera and Woehle & Roy et al. (2018) showed that the enzymes responsible for denitrification in *Globobulimina spp.* from a Swedish $O_2$ depleted

Fjord basin are indeed transcribed by eukaryotic RNA. These enzymes are homologues of enzymes that are also
used by bacteria for denitrification, which indicates an ancient prokaryotic origin of denitrification in foraminifera.
Nevertheless, the homologues of the enzymes that catalyze the first and the last step of foraminiferal denitrification
(Reduction of $NO_3^-$ to nitrite ($NO_2^-$) and reduction of nitrous oxide ($N_2O$) to $N_2$ gas; fig. 2) have not been identified,
yet. This indicates that foraminifera use other enzymes to catalyze these steps, or that they rely on bacterial
symbionts for these steps, or that they use an alternative denitrification pathway in general.
One hypothesis, brought up by Woehle & Roy et al. (2018) is that the homologue of the nitric oxide
reductase (Nor) is indeed a nitric oxide dismutase that has been proposed to catalyze the enzymatic reaction 2 NO
→ $N_2$ + $O_2$ (alternative pathway in fig. 2) (Ettwig et al., 2012). The presence of the eukaryotic denitrification
pathway found in foraminifera (Woehle & Roy et al., 2018) has been confirmed through other analyses of
foraminiferal transcriptomes (Orsi et al., 2020; Gomaa et al., 2021). Gomaa et al. (2021) also identified an enzyme
of yet unknown functionality that might be responsible for the first step in the foraminiferal denitrification
pathway. Recent metagenomics and transcriptomics results of denitrifying foraminifera indicate that bacterial
symbionts might perform the missing steps in the foraminiferal denitrification pathway or that they at least partly
contribute to the amount of $NO_3^-$ that is denitrified within foraminiferal cells (Woehle & Roy et al., 2022). It has
already been hypothesized before that the ectobionts, found on *Bolivina pacifica* from the Santa Barbara Basin are
either sulfate reducing or sulfur oxidizing bacteria (Bernhard et al., 2010a). The possible complementation of the
foraminiferal denitrification with bacterial symbionts appears to be contradictory to the results by Bernhard et al.
(2012b) who showed that *B. argentaea* consumed its intracellular $NO_3^-$ storage (likely for denitrification) even
after the antibiotics treatment. Gomaa et al. (2021) confirmed that *B. argentaea* also lacks the first and last
denitrification step in its transcriptome, although it lacks intracellular bacterial symbionts (Bernhard et al., 2012b).
Future studies might decipher, if indeed bacteria are responsible for the missing denitrification step and are immune
to such antibiotic treatment, if an oxygenic nitric oxide dismutase skips the last denitrification step as discussed
by Woehle & Roy et al. (2018) and/or if foraminifera have unknown enzymes that catalyze the missing steps as
suggested by Gomaa et al. (2021). The study by Woehle & Roy et al. (2022) also reconstructed that the last
common ancestor of denitrifying foraminifera likely has its origin during the Cretaceous, possibly related to the
occurrence of the Cretaceous Anoxic Events. Since the foraminiferal denitrification pathway is incomplete and the
first and last steps might be performed by Desulfobacteraceae in their microbiome the authors suggested that the
acquisition of denitrification ability in foraminifera occurred in multiple stages (starting during the Cretaceous)
but is not yet complete (Woehle & Roy et al., 2022).
It is noteworthy that denitrifying foraminifera from the Peruvian OMZ show a metabolic preference of
$NO_3^-$ over $O_2$ as an electron acceptor (Glock et al., 2019b). These foraminifera show an increasing cell volume
with increasing ambient $NO_3^-$ and decreasing $O_2$ concentrations. Similar observations have been made at the
California Borderlands, where some benthic foraminifera also increase their cell-volume with decreasing ambient
$O_2$ concentrations (Keating-Bitonti and Payne, 2017). Additional evidence for the metabolic $NO_3^-$ preference came
from comparing denitrification and $O_2$ respiration rates and scaling them to their cell volume (Glock et al., 2019b).
The scaling is lower for $O_2$ respiration than for denitrification, indicating that the $NO_3^-$ metabolism during
denitrification is more efficient than the $O_2$ metabolism during aerobic respiration in foraminifera from the
Peruvian OMZ. This might explain, why some infaunal denitrifying foraminifera follow the oxycline within
sediments (Linke and Lutze, 1993; Duijnstee et al., 2003). We have to keep in mind that $O_2$ can be quite harmful
for organisms that are not adapted to higher $O_2$ concentrations, due to its strong reactivity. Even trace amounts of
$O_2$ can inhibit denitrification and $O_2$ can repress the denitrifying enzyme synthesis (Smith and Tiedje, 1979;
Knowles, 1981; Tiedje, 1988; Mckenney et al., 1994). Thus, if denitrifying foraminifera are exposed to small
amounts of $O_2$ they cannot denitrify but also do not have enough $O_2$ to supply their demands for electron acceptors.
Larger amounts of $O_2$ might supply this demand but also harm the cell. For example, $O_2$ can inhibit the growth of
some obligate anaerobes poison enzymes that are important for their metabolism (Lu and Imlay, 2021). Also for
aerobes $O_2$ can be harmful. "Hyperoxia", an excess supply of $O_2$, leads to damaging effects by highly-reactive
metabolic products of $O_2$ (free $O_2$ radicals) that inactivate enzymes in the cell, damage DNA and destroy lipid
membranes (Frank and Massaro, 1980). Furthermore, foraminifera are able to store $NO_3^-$ within vacuoles, due to
its lower reactivity and still have an electron acceptor reservoir if $NO_3^-$ is depleted in their microhabitat. This is
not possible for $O_2$ due to its high reactivity (Auten and Davis, 2009). Finally, a review by Zimorski et al. (2019)
addresses the common misconception that the presence of $O_2$ improves the overall energetic state of the cell. It is
a fact that the energy yield from remineralizing glucose or amino acids is higher in the presence of $O_2$ ("$O_2$
respiration") but it is also a fact that that the synthesis of biomass consumes thirteen times more energy per cell,
if $O_2$ is present, compared to anoxic conditions. This is related to the chemical equilibrium between organic matter
and $CO_2$, which strongly shifts to the side of $CO_2$ in the presence of $O_2$ (Zimorski et al., 2019). All this might
explain, why the metabolism of at least some foraminifera is better adapted to denitrification than to $O_2$ respiration.
The circumstance that some foraminifera have a metabolic preference of $NO_3^-$ over $O_2$ as electron
acceptor (Glock et al., 2019b) and that other species like *U. peregrina* denitrify in some environments but
completely lack an intracellular $NO_3^-$ storage in others (Piña-Ochoa et al., 2010b) might partly explain the
microhabitat selectivity of benthic foraminifera in the sediment. According to the conceptual TROX model,
benthic foraminifera can be divided into groups, due to their microhabitat preference: epifauna, shallow infauna,
intermediate infauna and deep infauna (Jorissen et al., 1995). The presence of this species specific microhabitat
structure has first been documented by Corliss (1985). These microhabitats are mainly controlled by bottom water
$O_2$ concentrations and the supply of non-refractory organic matter (i.e., food, Jorissen et al., 1995). Due to our
increasing understanding about the anaerobic metabolism of foraminifera we can now assume that $NO_3^-$
availability is another controlling factor (Fig.3). This is also indicated by a study coupled early diagenetic modeling
with foraminiferal ecology to model the microhabitats of benthic foraminifera (Jorissen et al., 2022). According
to their metabolic preference for $NO_3^-$ or $O_2$ as electron acceptors many benthic foraminifera species that typically
occupy a certain microhabitat (epifauna, shallow infauna and deep infauna) might partly be assigned to three
different attributes (Aerobe, facultative anaerobe and facultative aerobe). Most likely there are exceptions to these
classifications that will be discussed below. Another controlling factor on the microhabitat can be the specific
trophic strategy of the foraminiferal species, which is further discussed in section 3.
**Deep infaunal** species can most likely be considered as **facultative aerobes** that have a metabolic
preference of $NO_3^-$ over $O_2$ (Glock et al., 2019b) and try to avoid trace amounts of $O_2$. They cannot be accounted
as obligate anaerobes, though, since they can withstand periods of oxygenation. Many experiments show that
denitrifiying foraminifera can switch to $O_2$ respiration, if they are exposed to $O_2$ (i.e., Piña-Ochoa et al., 2010b),
Still, they follow the oxycline in the sediments to avoid the inhibition of denitrification by trace amounts of $O_2$.
The $\delta^{13}C$ signature of shells of deep infaunal globobuliminids indicates that they calcify in sediment depth where
the pore water $O_2$ level reaches zero or even deeper in the sediments. The offset between $\delta^{13}C$ of *Globobulima*
spp. tests and $\delta^{13}C$ of epifaunal foraminifera or of bottom water dissolved inorganic carbon (DIC) is nearly equal
to the offset between DIC at the zero $O_2$ layer and the bottom water (Schmiedl and Mackensen, 2006) and often
can be even higher (Costa et al., 2023), indicating that many globobuliminids live even below the oxycline. Even
though they can switch to $O_2$ respiration (Piña-Ochoa et al., 2010b), these species most likely would try to avoid
crossing the oxycline since denitrification would be already inhibited by nM $O_2$ concentrations (Dalsgaard et al.,
2014) and the $O_2$ concentration slightly above the oxycline is not high enough to fulfil their metabolic demands.
Indeed, the model by Jorissen et al. (2022) describes the distribution of deep infauna very well, by using the
presence of $O_2$ as an inhibiting factor, which also promotes that they can rather be considered faculatative aerobes
instead of facultative anaerobes. Taxa belonging to the deep infaunal group that might be considered as facultative
aerobes that prefer $NO_3^-$ over $O_2$ include for example *Valvulineria inflata* and *bradyana*, *Bolivina seminuda*,
*Globobulimina pyrula, Globobulimina affinis* and *Cancris carmenensis* (Jorissen et al., 1995; Schmiedl and
Mackensen, 2006; Mojtahid et al., 2010; Glock et al., 2019b).
**Shallow infauna** can in many cases be considered as **facultative anaerobes** that are well adapted to the
presence of low $O_2$ concentrations but can switch to denitrification if they are exposed to anoxic conditions or need
to enter the deeper sediment parts to find food or avoid competitive stress. These species have the advantage that
they can utilize both fresh phytodetritus from the top of the sediments and organic matter of lower quality from
the deeper parts of the sediments. A good example for a shallow infaunal – facultative anaerobe species is *U.*
*peregrina*, that is well known for its shallow infaunal lifestyle (Schmiedl and Mackensen, 2006) and has been
found with or without intracellular $NO_3^-$ storage in different environments (Piña-Ochoa et al., 2010b; Langlet et
al., 2020). Of course, it cannot be generalized that all foraminifera from a shallow infaunal habitat are indeed
facultative anaerobes. At least some species that can be considered shallow infaunal have been shown neither to
be able to store $NO_3^-$ nor to denitrify. As mentioned above all specimens from the *Ammonia tepida* morphogroup
that have been analyzed so far lack an intracellular $NO_3^-$ storage and cannot denitrify (Piña-Ochoa et al., 2010b).
Nevertheless, these taxa are often exposed to anoxia and can sometimes even be found alive in 4 to 26 cm sediment
depth (Alve and Murray, 2001; Thibault de Chanvalon et al., 2015). It is possible that these foraminifera indeed
only have an aerobe metabolism and just become dormant under exposure to anoxia (dormancy is discussed in
another section). Though, another possibility is that intertidal species such as *A. venata*, *A. aberdoveyensis* or *A.*
*confertitesta* have other adaptations to anoxia than denitrification. Recent studies revealed other possible anaerobic
metabolic pathways in foraminifera such as fermentation or dephosphorylation of creatine phosphate which are
discussed in section 2.1.3 (Orsi et al., 2020; Gomaa et al., 2021). Also, an *Ammonia* sp. has been shown to take up

nitrogen from $^{15}N$ labeled $NO_3^-$ under $O_2$ depletion and to assimilate it within cell organelles known as electron dense bodies (Nomaki et al., 2016). Eventually, studies on the transcriptome of non-denitrifying species from infaunal environments might be able to show, if some of these species can switch an alternative anaerobe metabolism under exposure to anoxia.

Many **epifaunal species** can most likely be considered as **aerobes** that typically occur at the sediment-water interface or on elevated surfaces. Typical epifaunal – aerobe taxa include *Cibicides* spp. and *Planulina* spp. (Corliss and Chen, 1988; Lutze and Thiel, 1989). These species have the advantage that they are well adapted to collect fresh food supply from above (Wollenburg et al., 2021) but usually cannot withstand longer $O_2$ depleted periods (Mackensen et al., 1995). Nevertheless, recent genetic data indicates that *Cibicidoides wuellerstorfi* clusters very close to known denitrifying species in the phylogenetic tree, so it cannot be excluded that some *Cibicides* spp. may denitrify under certain circumstances (Woehle & Roy et al., 2022). In the same way as for the other microhabitats, not all species with an epifaunal lifestyle should be automatically considered as aerobes. There are examples of epifaunal benthic foraminifera that have not been found in well oxygenated environments but reach high abundances in $O_2$ depleted environments. One example is *Epistominella smithi*, which has been described in low $O_2$ environments, such as the Santa Barbara Basin (Harman, 1964) or the Peruvian OMZ (Erdem and Schönfeld, 2017). Nevertheless, the morphology of *E. smithi* strongly suggests an epifaunal lifestyle. Another example is the epifaunal species *Planulina limbata*. This species is abundant only in $O_2$ depleted environments on continental margins within the East Pacific (Natland, 1938; Erdem and Schönfeld, 2017; Glock et al., 2022). Recent *P. limbata* specimens are present in severely $O_2$ depleted water masses within the Peruvian OMZ ([$O_2$] = 3 - 12 µmol/kg, Glock et al., 2022). Nevertheless, *P. limbata* also adapts its pore density to the availability of $O_2$ (Glock et al., 2022), which might indicate that it has an aerobic metabolism, despite that its presence appears to be limited to low $O_2$ environments. Another possibility is that species such as *E. smithi* or *P. limbata* may denitrify under certain circumstances and therefore can also be considered as faculatitive anaerobes. Hopefully, measurements of metabolic rates, intracellular nutrient content and enzymatic activity might bring further evidence in the future, if at least some epifaunal species can switch to an anaerobe metabolism, when $O_2$ is too depleted.

The **intermediate infauna** is somehow an exceptional case. Common representatives of intermediate infaunal taxa are *Melonis barleeanus* or *Pullenia* spp. (Corliss, 1991). The typical example for intermediate infaunal species *M. barleeanus* is interesting, since it either stores no or only very small amounts of $NO_3^-$ (See table 2 and 3). Still, several studies indicate that *M. barleeanus* lives deeper in the sediments than some *Uvigerina* spp. (Corliss, 1991; Ní Fhlaithearta et al., 2018) although many *Uvigerina* species have been shown to store $NO_3^-$ and denitrify (Tab. 1&2). This might give room to speculate if *M. barleeanus* has other metabolic adaptations to $O_2$ depletion than denitrification or if it simply does not store large amounts of $NO_3^-$ but denitrifies $NO_3^-$ directly after the uptake from the seawater. Indeed, a recent study predicted the microhabitats of infaunal benthic foraminifera using an early diagenetic model and showed that the intermediate infauna clusters around the $NO_3^-$ maximum in the pore water (Jorissen et al., 2022). Future perspectives on understanding the biology of intermediate infauna might include transcriptome analyses to decipher other anaerobe metabolic pathways and testing the denitrification capacity after incubation in $NO_3^-$-free and $NO_3^-$-containing seawater.

Note that the deep infauna can even migrate deeper into the sediments below the depth of $NO_3^-$ penetration, if they must, due to their ability to intracellulary store $NO_3^-$ as a reservoir (Fig. 3). The deeper boundary of the deep infauna might be controlled by the zone of sulfate reduction, where free sulfide is produced, which could be toxic for the foraminifers. Research to measure denitrification rates in different benthic foraminiferal species continues (Langlet et al., 2020; Choquel et al., 2021). This will add to the scarce available data and contribute to estimates of the role of foraminifera in oceanic N-cycling. This topic is discussed a bit further in section 4.

2.1.2 Kleptoplasty

Kleptoplasty describes a symbiosis between algal chloroplasts and a host organism that sequesters the chloroplasts from algae (Clark et al., 1990). The word originates from the Greek word "Kleptes", which means "thief". Kleptoplasty in foraminifera is most extensively studied for shallow *Elphidium* and *Haynesina* species that often thrive within the photic zone and this research originated in the 1970s (Lopez, 1979; Lee et al., 1988; Correia and Lee, 2000, 2002b, a; Goldstein et al., 2004; Pillet et al., 2011, 2013; Cevasco et al., 2015; Jauffrais et al., 2016, 2017, 2018; Cesbron et al., 2017; Goldstein and Richardson, 2018; Jesus et al., 2021). Several studies showed that the sequestered chloroplasts in the intertidal species *Haynesina germanica* are still capable of photosynthesis under light exposure (Lopez, 1979; Cesbron et al., 2017). *H. germanica* often shares the habitat with species from the *Ammonia tepida* morphogroup (*Ammonia aberdoveyensis* or *Ammonia confertitesta* according to Hayward et al.,

2021) which also tend ingest chloroplasts but these chloroplasts do not show any photosynthetic activity anymore (Jauffrais et al., 2016).

The kleptoplasts in foraminifera orginate from diatoms, which has been confirmed on the basis of the chloroplast shape in TEM-observations and by sequencing the chloroplasts using molecular biological methods (Lopez, 1979; Lee et al., 1988; Cedhagen, 1991; Lee and Anderson, 1991; Bernhard and Bowser, 1999; Grzymski et al., 2002; Goldstein et al., 2004). Austin et al., (2005) hypothesized that the toothplates in *H. germanica* are morphological adaptations to crack diatom frustules for access to their chloroplasts. Recently, LeKieffre et al. (2018) showed in (aerated) incubation experiments with $H^{13}CO_3^-$ and $^{15}NH_4^+$ during a light/dark cycle that *Haynesina germanica* is indeed able to fix inorganic carbon and nitrogen under light exposure. Intertidal foraminifera are often exposed to $O_2$ depleted or even anoxic conditions, when water stagnates during low tide or if they are transported to deeper anoxic sediment layers by bioturbation (Rybarczyk et al., 1996; Cesbron et al., 2017). Oxygen penetration depths in tidal flats can vary between a few mm during low tide to several cm during high tide (Jansen et al., 2009). Thus, intertidal foraminifera are often exposed to anoxia, even within the first cm of the sediment column. *H. germanica* is also supposed to occur in black sediments of the British salt marsh tide pools (Bernhard and Bowser, 1999), which likely become anoxic during a tidal cycle (Rybarczyk et al., 1996) and it was among the first recolonizers of a Fjord suffering of organic pollution (Cato et al., 1980; Bernhard and Bowser, 1999). Kleptoplasty might thus be an additional adaptation of foraminifera from photic environments to stay active during periods of $O_2$ depletion, which already has been hypothesized by Cesbron et al., 2017.

Less well understood is the phenomenon of kleptoplasty, observed in the benthic foraminifers *Nonionella stella, Virgulina fragilis* and *Nonionellina labradorica* that can thrive below the photic zone and often inhabit $O_2$ -depleted sediments (Cedhagen, 1991; Bernhard and Bowser, 1999; Grzymski et al., 2002; Bernhard, 2003; Tsuchiya et al., 2015; Jauffrais et al., 2019; Gomaa et al., 2021; Powers et al., 2022). Experiments to test if *N. labradorica* is able to photosynthesize with its sequestered chloroplasts have been inconclusive. While Cedhagen (1991) found active photosynthesis in *N. labradorica* specimens incubated with $^{14}C$, Jauffrais et al. (2019) showed an increased $O_2$ respiration rate instead of $O_2$ production and chloroplast degradation in specimens exposed to light. Recently, Gomaa et al. (2021) found chloroplast encoded in transcripts of *N. stella*, indicating that the kleptoplasts in this species are still active. Genetic analyses revealed that the kleptoplasts in *N. stella* and *N. labradorica* are also mainly sequestered from diatoms, most likely after ingestion and selective digestion of phytodetritus (Grzymski et al., 2002; Jauffrais et al., 2019; Gomaa et al., 2021). Grymzki et al. (2002) calculated that the required amount of light for *N. stella* specimens collected from aphotic depths at the Santa Barbara Basin is too low to sustain active photosynthesis. Instead, they suggested that the kleptoplasts in foraminifera from aphotic environments provide the ability to fix inorganic nitrogen via the glutamine synthetase and glutamate 2-oxo-glutarate amidotransferase (GOGAT) pathway. Indeed, Jauffrais et al. (2019) showed that kleptoplastic *N. labradorica* are able to fix inorganic nitrogen but coupled TEM/Nano-SIMS revealed that the assimilated nitrogen is associated with electron opaque bodies instead of sequestered chloroplasts. Analyses of the transcriptome of *N. stella* by Gomaa et al. (2021) support the observations by Grimzki et al. (2002), since *N. stella* appears to be able to fix ammonia by itself. They also found that the fucoxanthin-chlorophyll binding protein (FCP) was expressed in the transcriptome of *N. stella* and speculated that the ability to synthesize FCP was derived from the kleptoplasts by horizontal gene-transfer. FCP is a pigment, commonly found in chloroplasts of brown algae and allows a more efficient photosynthesis with a light absorption bandwidth especially useful in aquatic environments (Papagiannakis et al., 2005; Premvardhan et al., 2008). The true function of the kleptoplasts in deep-sea benthic foraminifera from aphotic, often $O_2$ depleted, environments still remains enigmatic, though.

2.1.3 Other strategies: Fermentation, utilization of high energy phosphates and peroxisome proliferation

Several recent publications based on advances in molecular biological methods (e.g., next generation sequencing) have revealed some other metabolic adaptations of foraminifera that thrive under $O_2$ depletion (examples see fig. 4) (Woehle & Roy et al., 2018, 2022; Orsi et al., 2020; Gomaa et al., 2021). In *N. stella* and *Bolivina argentea*, Gomaa et al. (2021) found evidence for the expression of proteins, including pyruvate -ferredoxin oxidoreductase (PFOR) and [FeFe]-hydrogenase, that are characteristic of anaerobic metabolism. These PFOR sequences were indeed eukaryotic and closely related to those of the facultative anaerobe polychaete *Capitella teleta* and the anaerobic protistan parasite *Blastocystis*. The [FeFe]-hydrogenase is very similar to those in the amoeba/flagellate *Naegleria gruberi*, which has experimentally been shown to be active and to produce molecular hydrogen even under aerobic conditions (Tsaousis et al., 2014). Due to these observations Gomaa et al. (2021) suggested that *N. stella* and *B. argentaea* might be able to produce $H_2$ gas and have the capacity for an anaerobic energy metabolism.

Another important observation was made by Orsi et al. (2020). They used metatranscriptomics on sediments from the Namibian shelf, where the foraminiferal community is dominated by *Bolivina* and *Stainforthia* species. Presumably living foraminifera were present in the sediment column up to 28 cm depth in an anoxic habitat with high sulfide concentrations. The gene expression of those foraminifers increased under sulfidic conditions, which indicates that they not only survive but thrive under anoxic conditions. The anaerobic energy metabolism of these foraminifers seems to be sufficient enough to support calcification and phagocytosis even under anoxic conditions. Evidence for foraminiferal calcification under anoxia already came up by a study by Nardelli et al. (2014). Orsi et al. (2020) suggested that the Namibian foraminifera use phagocytosis (vacuolic ingestion of food particles) to ingest prey cells even under anoxic conditions. These processes (calcification and the ingestion of prey cells by phagocytosis) require bursts of high energy, which the authors suggest is generated by dephosphorylation of an intracellular creatine phosphate storage to regenerate ATP from ADP. Evidence for the capacity for the dephosphorylation of creatine phosphate under anoxia was indicated by the metatranscriptomes. In addition, a high intracellular dissolved inorganic phosphate storage has been found in benthic foraminifera from the Peruvian OMZ, which might serve as a reservoir to synthesize creatine phosphate and/or to synthesize polyphosphates that might be broken down to harvest energy (Glock et al., 2020). Orsi et al. (2020) and Gomaa et al. 2021 also found evidence for another anaerobic metabolism. Their data indicates that the foraminifers metabolize hydrolyzed organics to produce ATP using fermentation and fumarate reduction.

Most foraminifera species from $O_2$ depleted habitats possess numerous peroxisomes that are usually associated with mitochondria and the endoplasmatic reticulum (Bernhard and Bowser, 2008). Bernhard and Bowser (2008) hypothesized that these peroxisome proliferations might be used to either metabolize $H_2O_2$ and other highly reactive oxygen species that are produced within the chemocline close to the oxic/anoxic boundary or to reduce the oxidative stress by these compounds. Indeed, they showed in an experiment that ATP concentrations in foraminifera increased proportional to ambient $H_2O_2$ concentrations. A recent study on transcriptome and metatranscriptome of *N. stella* and *B. argentaea* from the Santa Barbara Basin revealed that these species utilize an adaptable mitochondrial and peroxisomal metabolism, depending on the chemical treatment in the experiment (Powers et al., 2022). The high plasticity of their peroxisomal and mitochondrial metabolism might be substantial for survival at the highly variable conditions at the chemocline in the sediments. The results by Powers et al. (2022) indicate that at least some processes that are involved in foraminiferal denitrification are associated with mitochondria. Interestingly, the expression of denitrification related genes in both species was upregulated after incubation with elevated $H_2O_2$ but without $NO_3^-$ and downregulated, if they were incubated without $H_2O_2$ but with $NO_3^-$, compared to a control treatment with both $H_2O_2$ and $NO_3^-$. In the same way several peroxisomal processes were upregulated in the $H_2O_2$ only treatment. In addition, despite that both species are able to denitrify, Powers et al. (2022) found distinct metabolic adaptations to anoxia in both species. For example, a quinol:fumarate oxidoreductase, which is considered as an adaptive mechanism for anaerobic respiration in eukaryotic organisms, was present in *N. stella* but not in *B. argentaea*. Vice versa, *B. argentaea* has the capacity to digest food vacuole contents under $O_2$ depletion, while *N. stella* was lacking food vacuoles (Powers et al., 2022).

2.1.4 Dormancy

Dormancy is another strategy to survive anoxia or extreme $O_2$ depletion for some benthic foraminifera that cannot denitrify. Dormancy is defined as the reduced or suspended metabolic activity in response to exogenous factors (Ross and Hallock, 2016). Observations that indicate the potential of dormancy in foraminifera have been documented since the 1950s and are extensively reviewed by Ross & Hallock (2016). Nevertheless, many aspects of foraminiferal dormancy, such as role in foraminiferal life cycle or its role in structuring foraminiferal assemblages remained unexplored (Ross and Hallock, 2016).

In the 1990s some studies suggested that some foraminifera may become dormant when exposed to anoxia. Bernhard and Alve (1996) observed that the ATP concentration ([ATP]) of the benthic foraminiferal species *Bulimina marginata, Stainforthia fusiformis* and *Adercotryma glomerata* flushed with $N_2$ gas to drive out $O_2$ was significantly lower than in specimens from well-aerated conditions. They interpreted this observation as an indication that dormancy is a survival strategy for some foraminiferal species when they are exposed to periods of anoxia. Linke & Lutze (1993) observed cysts of *Elphidium incertum* from putative anoxic habitats that might be interpreted as a sign for dormancy and Hannah and Rogerson (1997) hypothesized that foraminifera transported to an anoxic sediment layer might become dormant until they return to aerated conditions by transport through bioturbation.

Recently, dormancy of foraminifera exposed to anoxia had gained more attention again. LeKieffre et al. (2017) did a feeding experiment with specimens from the *Ammonia tepida* morphogroup (*A. confertitesta*

according to Koho et al., 2018 and Hayward et al., 2021) using a $^{13}$C-labeled diatom film as food source. They
compared the metabolic differences of *Ammonia sp.* between oxic and anoxic conditions by mapping the
distribution of $^{13}$C within the cells using coupled TEM/Nano-SIMS and by analyzing the carbon concentration and
stable carbon isotopic composition of the total organic matter and individual fatty acids in the foraminifer. Nearly
the complete diatom biofilm was consumed and the foraminiferal cytoplasm was strongly enriched in $^{13}$C under
oxic conditions. Specimens from the anoxic incubation ingested only few of the diatoms and those were neither
assimilated nor metabolized further. In addition, the specimens from the oxic incubation produced a significant
amount of specific polyunsaturated fatty acids, which was not the case under anoxic conditions. *A. confertitesta*
reacted to the induced anoxia with a severely reduced metabolic rate within less than 24 hours. All these
observations provide solid evidence that dormancy is a survival strategy of *A. confertitesta* under anoxia.
Koho et al. (2018) further analyzed cell structural changes in *Ammonia* spp. under exposure to anoxia
collected from the field as well as from incubations. The specimens from anoxia showed an increase in lipid
droplets and electron dense bodies within their cytoplasm. The cytoplasm itself was thinned out, which was
interpreted as metabolization of the cytosol. In addition, while absent within the specimens from oxic
environments, various bacteria were present within the cytoplasm of the specimens from anoxia. These were
interpreted as endobionts but might also be parasites that could not be fended off, due to the drastically reduced
metabolism during dormancy under anoxia. A continuum of intracellular bacteria including prey in food vacuoles,
endobionts, parasites and necrophages has been documented before in benthic foraminifera from cold seeps
(Bernhard et al., 2010b). It already has been hypothesized by the authors that bacteria switched their function from
endobionts to predators, depending on the vitality of the host cell. Considering all the studies about dormancy, it
is likely that dormancy is a common survival strategy for foraminiferal species that either get exhausted of suitable
electron acceptors (i.e., $O_2$ or $NO_3^-$) or are exposed to periods of extreme environmental conditions. Since there is
evidence for dormancy in both *S. fusiformis* and *B. marginata* (Bernhard and Alve, 1996), it is likely that even
denitrifying species can get dormant under unfavorable conditions. Another *Stainforthia* sp. has been shown to
denitrify and *B. marginata* stores $NO_3^-$ in some environments (Piña-Ochoa et al., 2010b).

### 3 Trophic interactions in $O_2$ depleted environments

In general, benthic foraminifera show a wide range of trophic strategies. Gooday et al. (2008) suggested
that they can be separated according to their main trophic types (examples see fig. 5): A: Selective herbivores,
which include phytophagous species that consume only phytodetritus; B: Seasonal herbivores, which feed on fresh
phytodetritus, when available and consume sedimentary organic matter at other times; C: Detrivores that non-
selectively ingest sediment and consume the present degraded organic matter, bacteria and/or other organisms; D:
Selective bacterivores, that consume only bacteria; and E: Suspension feeders, that either erect from the sediments
or occur on elevated substrates. The latter two are not discussed in detail, since they mainly apply to abyssal species
that inhabit more oxygenated environments. Nevertheless, some *Cibicides* and *Planulina* species, can also inhabit
environments with relatively low $O_2$ concentrations (Erdem and Schönfeld, 2017; Rathburn et al., 2018;
Hoogakker et al., 2018b; Glock et al., 2022) and at least some of these *Cibicides* species are certainly suspension
feeders (Wollenburg et al., 2018, 2021). The trophic types that have been introduced above suggest that
foraminifera mainly feed on a low trophic level and it has been suggested that they constitute a trophic link to
higher levels in the food chain (Lipps and Valentine, 1970; Gooday et al., 1992; Nomaki et al., 2008).
There are a few studies that specifically focused on trophic interactions of foraminifera in environments
where $O_2$ is scarce or absent. Early observations have been documented by Nomaki et al. (2006), who conducted
an *in situ* feeding experiment at central Sagami Bay (1450 m), Japan, using $^{13}$C labeled algae and bacteria. Bottom
water $O_2$ concentration at this location is usually less than 60 µM and $O_2$ penetration depth into sediments varies
between 3 and 10 mm indicating that infaunal foraminifera in this habitat are regularly exposed to hypoxia and
anoxia (Glud et al., 2005). Nomaki et al. (2006) described three different feeding strategies by benthic foraminifera
in this environment. Since the bottom water $O_2$ concentrations at central Sagami Bay are fluctuating and not strictly
hypoxic, these observations likely apply to more oxygenated environments as well, especially for the shallow
infaunal species. *Uvigerina akitaensis, Bolivina spissa* and *Bolivina pacifica* selectively ingest fresh phytodetritus
and thus can be described as phytophagous species (selective herbivores). *Bulimina aculeata*, *Textularia*
*kattegatensis* and *Globobulimina affinis* ingest fresh phytodetritus selectively but feed on sedimentary organic
matter instead, when fresh phytodetritus is unavailable (seasonal herbivores). The species *Cyclammina cancellata*
and *Chilostomella ovoidea* ingest sedimentary organic matter at random and can thus be described as detrivores.
A later study confirmed these trophic types for most of the species at Sagami Bay by measuring the nitrogen
isotope fractionation ($\delta^{15}$N) of their amino acids, which is commonly used to trace the trophic position of an
organism in the food chain (Nomaki et al., 2015). Another feeding experiment at Sagami Bay by Nomaki et al.
(2011) revealed that all of the analyzed benthic species assimilated carbon from $^{13}$C labeled glucose and thus can
effectively utilize also dissolved organic carbon. The same study indicated that even the deep infaunal detrivores
can be selective regarding their food source. Four of the five analyzed species, except *C. cancellata*, incorporated
proportionally more $^{13}$C-labeled organic matter from the green algae *Dunaliella* sp. than from other carbon sources,
while *C. cancellata* preferentially incorporated carbon from *Chlorella* sp. (Nomaki et al., 2005, 2006, 2011).
Additional feeding experiments have been conducted at the Arabian Sea OMZ, where benthic foraminifera from
locations with different bottom water $O_2$ concentrations have been supplied with $^{13}$C and $^{15}$N labeled algae (Enge
et al., 2014, 2016). Nine out of nine analyzed species took up labeled phytodetritus during the four days
experimental phase (Enge et al., 2014). The foraminifera took up the highest amount of labeled carbon in the OMZ
center and the uptake decreased with distance from the OMZ (Enge et al., 2016). The authors hypothesized that
either the foraminifera from the core OMZ have a higher carbon demand or that there was less food competition
with macrofauna at the $O_2$ depleted locations. Similar to the studies by Nomaki et al. at Sagami Bay, the
experiments by Enge et al. (2014 & 2016) showed a more or less selective ingestion at the Arabian Sea OMZ
depending on the foraminiferal species. For example, several several *Uvigerina* species took up large amounts of
carbon from the labeled algae and are thus either selective or seasonal herbivores, while *Globobulimina* spp. took
up either no or only small amounts of the labeled carbon indicating their detritivore behavior (Enge et al., 2016).
Further examples for selective herbivores, opportunistic omnivores, which include seasonal herbivores, and
sediment detrivores are discussed by Gooday et al. (2008). It appears that many of the species that are considered
to be selective herbivores (e.g., *B. spissa*, *U. akitaensis*, *Eponides pusillus* or *Cassidulina carinata*) are living
epifaunal or shallow infaunal, although the selective herbivore *B. pacifica* can be also considered as intermediate
infauna (Gooday et al., 2008). The seasonal herbivores (or opportunistic omnivores; e.g., *U. peregrina*, *G. affinis*
or *G. pacifica*) can be found in a relatively wide range of microhabitats from shallow to deep infauna (Gooday et
al., 2008). Species that are considered to be sediment deposit feeders (or detrivores, e.g., *C. ovoidea* or *M.
barleeanum*) are usually found in the deeper habitats and belong to intermediate to deep infauna (Gooday et al.,
2008). This indicates that the selective herbivores must live closer to the source of fresh food supply, while the
less selective species can also feed on degraded organic matter or bacteria deeper in the sediments. Thus, the
specific trophic type is another control on the microhabitat of benthic foraminifera in addition to the availability
of $O_2$, $NO_3^-$ and the metabolic adaptations discussed in section 2. Indeed, the coupled diagenetic and ecologic
model of Jorissen et al. (2022) successfully uses different types of food particles as a controlling factor to simulate
the microhabitats of benthic foraminifera.
Although benthic foraminifera feed mainly on detritus and minute organisms there is also (less common)
evidence for carnivorous behavior when foraminifera prey on meiofauna (e.g., Lee, 1980; Bowser et al., 1986,
1992; Hallock and Talge, 1994). These observations have mainly been done on species that usually live in
oxygenated environments. Dupuy et al. (2010) documented carnivorous behavior in a laboratory experiment also
for the *Ammonia tepida* morphogroup (*A. aberdoveyensis* or *A. confertitesta* according to Hayward et al., 2021),
which is not uncommon in anoxic layers of tidal mudflats. A study on the trophic behavior of intertidal
foraminifera, using metabarcoding brought up evidence that *A. confertitesta* is actively preying on small
eukaryotes (e.g., nematodes) even in their natural environment (Panagiota-Chronopoulou et al., 2019). The
intracellular eukaryotic community in *A. confertitesta* varies with sediment depth but even up to 10 cm depth the
metabarcoding indicates freshly ingested eukaryotic prey in this species (Panagiota-Chronopoulou et al., 2019).
Still, the main eukaryotic prey of *A. confertitesta* appear to be diatoms (Panagiota-Chronopoulou et al., 2019).
Similar results have been documented by Schweizer et al. (2022). Recently, new evidence came up indicating
ingestion of nematodes by *Globobulimina auriculata* from the $O_2$ depleted Alsbäck Deep in Gullmar Fjord,
Sweden (Glock et al., 2019a). The species *G. auriculata* denitrifies and lives under $O_2$ depleted conditions (Woehle
& Roy et al., 2018). It is inconclusive, though, if the foraminifer preys on the nematode or vice versa but the
nematodes have most likely been ingested in the natural $O_2$ depleted habitat (Glock et al., 2019a). Although
predation is the main type of interaction in aerobic communities, it usually plays a much smaller role in anoxic
communities (Fenchel and Finlay, 1995). This is related to the low growth yields associated with the anaerobic
metabolism, which results in very short food chains. Thus, the decrease in energy flow along the anaerobic food
chains is higher than along the aerobic food chain (Fenchel and Finlay, 1995). The predatory isopod *Saduria
entomon* for example strongly reduces its predatory activity under hypoxia in comparison to aerobic conditions
(Sandberg, 1994) and the predator/prey biomass ratio has been shown to be 4 times lower in anoxic environments
compared to oxic environments (Fenchel and Finlay, 1995). There is evidence that foraminifera from the Namibian
shelf can perform phagocytosis (vacuolic ingestion of food particles) even under anoxic conditions, which usually
requires bursts of energy (Orsi et al., 2020). This study provides further evidence that the Namibian foraminifera
express enzymes for lysing digested prey cells inside food vacuoles after phagocytosis (schematic representations
for phagocytosis and predation on meiofauna shown in fig. 5). The evidence for phagotrophy and predation on or
by benthic foraminifera under $O_2$ depleted conditions, although it is rare, is thought-provoking and future studies
might shed more light on predator-prey interactions of benthic foraminifera in $O_2$ depleted environments. In
general, future metabarcoding studies to identify food sources of deep infauna or foraminifera that inhabit anoxia
might shed more light on trophic strategies in $O_2$ depleted environments.

## 4 The role of foraminifera in benthic nutrient cycling and biogeochemistry

Pina-Ochoa et al. (2010b) also suggested the possible importance of denitrifying foraminifera for the
benthic N-cycle, due partly to their high abundances in $O_2$ depleted environments. In some environments, such as
certain habitats in the Peruvian OMZ, foraminifera even seem to be the key players in benthic denitrification (Glud
et al., 2009; Glock et al., 2013, 2019b; Choquel et al., 2021). Complete heterotrophic denitrification produces non-
reactive (i.e., not bioavailable) $N_2$ gas. Denitrifying benthic foraminifera can thus be considered a sink for
bioavailable N. The recent genetic studies on denitrifying benthic foraminifera did not find transcripts for
homologues of enzymes that catalyze the last step of denitrification – the reduction of $N_2O$ to $N_2$ (Woehle & Roy
et al., 2018, 2022; Orsi et al., 2020; Gomaa et al., 2021). Some Globobuliminids from the $O_2$ depleted Alsbäck
Deep in the Swedish Gullmar Fjord have been shown to produce $N_2O$ gas as product of denitrification, although
the rates were lower than their rates for complete denitrification (Piña-Ochoa et al., 2010a). The $NO_3^-$ storage in
denitrifying foraminifera, but also in some sulfur bacteria, such as *Beggiatoa*, is of greater importance for benthic
biogeochemical cycling, due to the potential of biological transport of these intracellular reservoirs (Dale et al.,
2016). Most of the other diagenetic models that describe and calculate benthic N-cycling are based on (and limited
to) diffusive transport of the different N-species in bottom and pore water. Active biological transport of different
N-species can thus efficiently influence the benthic fluxes of different N-species (Dale et al., 2016).
The estimates of total benthic foraminiferal denitrification rates are mainly based on upscaling individual
species specific denitrification rates by the living abundances of benthic foraminifera in different environments
(Piña-Ochoa et al., 2010b; Glock et al., 2013, 2019b). This approach is limited by the availability of species
specific denitrification rates, although, various approximations can be used to calculate estimated denitrification
rates for species with unknown denitrification rates (Glock et al., 2013). A summary of all published benthic
foraminiferal denitrification rates can be found in tab. 1. Further data about species specific foraminiferal
denitrification rates will improve our estimates about the role of foraminifera in benthic N-cycling and, thus, also
models for benthic biogeochemical cycling.
Recently, it has been found that some benthic foraminifera are not only storing $NO_3^-$ for denitrification
but also store larger amounts phosphate (Glock et al., 2020). The intracellular phosphate concentration can exceed
the concentration in the surrounding pore waters by a factor of 10 to 100. The use of this intracellular phosphate
storage is still under debate. Hypotheses include the synthesis of polyphosphates or a reservoir for the synthesis of
phospholipids for the cell membranes (Glock et al., 2020). In addition, there is evidence that the intracellular
phosphate storage in foraminifera facilitates phosphogenesis in some environments, similar to the intracellular
polyphosphate enrichments in some sulfur bacteria (Schulz and Schulz, 2005). The release of phosphate after
breakdown of these polyphosphates to harvest energy in times of electron acceptor depletion results in apatite
supersaturation and initiates phosphogenesis (Schulz and Schulz, 2005). Sediments at the lower boundary of the
Peruvian OMZ contain many small phosphorite grains with similar size and shape of foraminifera (Manheim et
al., 1975; Glock et al., 2020). The sand fraction of the surface sediments in this region is a mixture of pristine
living foraminifer shells with dead tests that show a transition from shells that are filled with phosphorites until
small phosphorite grains that only retain the size and coarse shape of a foraminifer. It is likely that a *post mortem*
release of the intracellular phosphate storage results in a supersaturated microenvironment within the shells that
initiates apatite formation (Glock et al., 2020) in a similar way as it has been suggested for other organisms
(Kulakovskaya, 2014). The recent evidence for the potential of benthic foraminifera to use dephosphorylation of
an intracellular creatine phosphate storage to regenerate ATP under anoxic conditions might be another
explanation for the high intracellular phosphate storage (Orsi et al., 2020). It might be that this is an adaptation of
foraminifera to enable phagocytosis even under anoxic conditions.
4.1 Estimating the contribution of foraminifera to benthic nutrient budgets and fluxes
The intracellular $NO_3^-$ storage in benthic foraminifera from different environments shows a relatively
wide concentration range (Tab.2). In addition, species that lack intracellular $NO_3^-$ storage are relatively widespread
and there are species that, depending on the environment, either have or lack intracellular $NO_3^-$ (Tabs.2&3). Most

of the species that have been found both with and without intracellular $NO_3^-$ in different environments (bold species in Tab.3) are species that are typically shallow infaunal. They belong to the group of foraminifera that might partly be considered facultative anaerobe and likely are opportunistic species that are well adapted to transitional environments with periodic $O_2$ depletion, since they apparently can handle oxygenated and anoxic environments (see 2.1.1). In addition, the $NO_3^-$ is most likely stored in seawater vacuoles and the vacuole volume of foraminifera can have a large variability (LeKieffre et al., 2018).

Given this variation in $NO_3^-$ storage capability, the reliability of estimates for the foraminiferal contribution to $NO_3^-$ budgets depends crucially on the availability of data. The more data there is, the better we are able to calculate foraminiferal $NO_3^-$ budgets. Nevertheless, there are thousands of benthic foraminiferal species and a considerable amount of these species inhabit $O_2$ depleted environments and potentially store $NO_3^-$ and denitrify. It will be unrealistic to measure the intracellular nutrient content and metabolic rates for all foraminifera. Thus, functions to estimate the contribution of species with unknown denitrification rates or intracellular $NO_3^-$ will provide more data for better estimates of total foraminiferal budgets within the nitrogen cycle. Of course, it is not possible to strictly define, which foraminiferal species are able to denitrify or to store $NO_3^-$ without real measurements. If a foraminiferal species inhabits $O_2$ depleted environments and belongs to a genus of the species, listed in tab.1 or tab.2, as a rule of thumb, they are good candidates for potential denitrifiers. In addition, if a species is known to inhabit well oxygenated environments and/or belongs to a genus of the species shown in tab.3 it should be avoided to use equations presented below to estimate $NO_3^-$ storage or denitrification rates. Considering this, an analysis of published data about intracellular $NO_3^-$ content reveals a highly significant correlation between the intracellular $NO_3^-$ and the cell volume of denitrifying benthic foraminifera (Fig. 6; power regression; $R^2 = 0.59$; $F = 86$; $P = 3E-13$).

Thus, the intracellular $NO_3^-$ content of a potentially denitrifying foraminifer can be estimated from its biovolume according to the following equation:

$$Eq.2: \ln(NO_{3\ i}^-) = 1.07(\pm0.11) \times \ln(V_{cell}) - 11.5(\pm1.9)$$

where $NO_{3\ i}^-$ is the intracellular $NO_3^-$ content in pmol ind$^{-1}$ and $V_{cell}$ is the cell volume in $\mu m^3$. Note that only species from table 2 with an intracellular $[NO_3^-] \geq 1$ mM were considered for the power regression. In addition, two extreme datapoints were discarded as outliers (see supplementary note). Similar equations have been published to estimate foraminiferal denitrification rates (Glock et al., 2019b; here Eq.3) and intracellular dissolved inorganic phosphorous content (Glock et al., 2020, here Eq.4).

$$Eq.3: \ln(R_{den(ind)}) = 0.68(\pm0.12) \times \ln(V_{cell}) - 5.57(\pm1.9)$$

$$Eq.4: \ln(DIP_i) = 0.82(\pm0.03) \times \ln(V_{cell}) - 7.65(\pm0.52)$$

where $R_{den(ind)}$ is the individual denitrification rate in pmol ind$^{-1}$ day$^{-1}$ and $DIP_i$ is the intracellular dissolved inorganic phosphorous content in pmol ind$^{-1}$.

Further equations and principles for upscaling foraminiferal nitrogen- and phosphorous-budgets from abundances of living foraminifera can be found in Glock et al. (2013, 2019b and 2020) and (Xu et al., 2021). Formulae to estimate the biovolume of many different common shapes of foraminifera have recently been published (de Freitas et al., 2021). Due to the high uncertainties related to the natural variability in metabolic rates and nutrient storage, a thorough error estimation is recommended (see Appendix B in Glock et al. 2020). With an increasing amount of data about metabolic rates and intracellular nutrient storage more accurate models and equations might become available in the future that describe the role of benthic foraminifera within marine biogeochemistry. Similar models and equations might be also very helpful for exploring the role of planktonic foraminifera in pelagic biogeochemistry.

**6 Author contribution**

NG wrote the manuscript and did the data compilation and statistical analyses.

**7 Competing interests**

The author declare that they have no conflict of interest.

**8 Acknowledgements**

I would like to thank Gerhard Schmiedl for providing constructive feedback on an early draft of this manuscript. In addition, I acknowledge the extensive and constructive feedback Andrew Gooday, Frans Jorissen, another anonymous reviewer and the editor Lisa Levin, which significantly improved this manuscript. Funding was provided by the Deutsche Forschungsgemeinschaft (DFG) through Heisenberg grant GL 999/3-1 to N.G. Finally, I would like to thank all the authors and co-authors that are cited in this review, because of their pioneering research on benthic foraminifera from $O_2$ depleted environments.

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

**Figure captions:**
Figure 1: Schematic representations for three survival strategy examples performed by benthic foraminifera under
$O_2$ depleted conditions.
Figure 2: Schematic view of two alternative pathways suggested for foraminiferal denitrification. Abbreviations
above the reaction arrows indicate the enzymes that are catalyzing the respective step (see legend). Enzymes in
black have been found transcribed by eukaryotic (foraminiferal) RNA (Woehle & Roy et al., 2018). Enzymes in
grey are missing in the foraminiferal denitrification pathway and are likely performed by bacterial symbionts
(Woehle & Roy et al., 2022). The straight pathway above describes the normal heterotrophic denitrification
pathway. The junction, catalyzed by the Nod, which produces $O_2$, has been suggested as an alternative pathway
for foraminiferal denitrification (Woehle & Roy et al., 2018).
Figure 3: TROX model modified after Jorissen et al. (1995) and Xu et al. (2021). The supply of organic matter
and bottom water $O_2$ and $NO_3^-$ concentrations in different environments control the penetration depth of $O_2$ and
$NO_3^-$ into the sediment. Benthic foraminifera choose their microhabitat according to their metabolic preferences
for $O_2$ or $NO_3^-$ as an electron acceptor and the availability of food. Intermediate infauna is not specifically
schematized in the figure but peak between the shallow and deep infauna with an overlap to both directions. Note
that denitrifying foraminifera can actively transport intracellular $NO_3^-$ below the $NO_3^-$-penetration depth in the
sediments. The deeper regions where production of free sulfide occurs will mainly be avoided. For further details
see text.
Figure 4: Examples for molecules and processes that are relevant in the anaerobic metabolism of foraminifera. A:
Structural formula of creatine phosphate. B: The role of creatine kinase (Ck) and creatine phosphate in the
anaerobic metabolism. High energy creatine phosphate is produced by phosphorylation of creatine. Creatine
phosphate can rapidly recycle ADP to ATP to provide resources for rapid energy bursts. This pathway has been
described by Orsi et al.(2020). C: Fermentation has been found to be relevant in the anaerobic metabolism of
foraminifera by both Orsi et al.(2020) and Gomaa et al. (2021). The possibility of a $H_2$ producing fermentation
pathway, catalyzed by Fe-hydrogenase has been described by Gomaa et al. (2021).
Figure 5: A: Schematic representation of a *bolivinid* ingesting bacterial cells. Recent studies showed that benthic
foraminifera from $O_2$ depleted habitats have the capacity of phagocytosis even under anoxia (Orsi et al., 2020). B:
Schematic representation of *Ammonia* sp. preying on a nematode. Some benthic foraminifera are known to prey
on meiofauna (Dupuy et al., 2010) and there is evidence, that even some *globobuliminids* that usually thrive under
$O_2$ depleted conditions might prey on nematodes (Glock et al., 2019a).
Figure 6: Log-log plot and power regression of intracellular $NO_3^-$ content ($NO_{3\,i}^-$) against the biovolume ($V_{cell}$) of
benthic foraminifera from diverse environments (Tab. 2). Only species with an intracellular $[NO_3^-] \geq 1$ mM, where
both $NO_{3\,i}^-$ and $V_{cell}$ were published were considered for the power regression.







**Tables**

Table 1: Summary of foraminiferal denitrification rates (individual and volume specific), where Ind. refers to the
number of individuals used for one incubation. Individual denitrification rates refer to average rates per
individual while specific denitrification rates refer to rates normalized to the biovolume of the foraminifers.
Errors are given as standard deviations (1sd) [a]: Data from Piña-Ochoa et al. (2010b); [b]: Data from Bernhard et
al. (2012b); [c]: Data from Glock et al. (2019b); [d]: Data from Langlet et al. (2020); [e]: Data from Woehle & Roy
et al. (2018); [f]: Data from Risgaard-Petersen et al. (2006); [g]: Data from Choquel et al (2021); [h]: Data from
Høgslund et al. (2008). *: *Ammonia tepida* is a morphogroup of pseudocryptic species that recently had a
revision. Specimens earlier identified as *A. tepida* are likely either *A. aberdoveyensis* or *A. confertitesta*
1165              according to Hayward et al. (2021).

| Species | Location | Ind. | Denitrification (pmol nitrogen individual$^{-1}$ d$^{-1}$) | Specific Denitrification (pmol nitrogen $\mu$m$^{-3}$ d$^{-1}$) |
|---|---|---|---|---|
| *Ammonia tepida*\*[a] | (Aiguillon Bay) | 2 | 0 (n = 1) | 0 (n = 1) |
| *Bolivina argentea*[b] | (Santa Barbara) | 10 | 1976 ± 1103 (n = 8) | n.a. |
| *Bolivina costata*[c] | (OMZ, Peru) | 13-14 | 21 ± 8 (n = 3) | 3.42E-5 ± 1.53E-5 (n = 3) |
| *Bolivina plicata*[c] | (OMZ, Perú) | 5-8 | 105 ± 33 (n = 2) | 2.49E-5 ± 3.27E-6 (n = 2) |
| *Bolivina plicata*[a] | (OMZ, Perú) | 3 | 79 (n = 1) | 1.05E-5 (n = 1) |
| *Bolivina seminuda*[c] | (OMZ, Peru) | 6-13 | 86 ± 57 (n = 11) | 5.73E-5 ± 2.53E-5 (n = 10) |
| *Bolivina seminuda* [a] | (OMZ, Perú) | 3 | 216 (n = 1) | 4.15E-5 (n = 1) |
| *Bolivina spathulata*[d] | (Bering Sea) | 19 | 11 (n = 1) | 9.17E-7 (n = 1) |
| *Bolivina spissa*[c] | (OMZ, Peru) | 4-7 | 373 ± 205 (n = 5) | 9.12E-5 ± 3.66E-5 (n = 5) |
| *Bolivina subaenariensis*[a] | (B. Biscay) | 10-12 | 78 ± 2 (n = 2) | 3.12E-6 ± 5.43E-7 (n = 2) |
| *Cancris carmenensis*[c] | (OMZ, Peru) | 3-4 | 765 ± 306 (n = 3) | 1.86E-5 ± 4.25E-6 (n = 3) |
| *Cassidulina limbata*[c] | (OMZ, Peru) | 4-6 | 45 ± 16 (n = 4) | 7.62E-6 ± 9.25E-6 (n = 3) |
| *Fursenkoina cornuta*[b] | (Santa Barbara) | 10 | 1386 ± 320 (n = 2) | n.a. |
| *Globobulimina auriculata*[e] | (Gullmar fjord) | 4-5 | 75 ± 44 (n = 4) | 2.39E-6 ± 1.50E-6 (n = 4) |
| *Globobulimina pacifica*[d] | (Bering Sea) | 4-5 | 378 ± 471 (n = 2) | 1.63E-5 ± 2.07 E-5 (n = 2) |
| *Globobulimina turgida*[a] | (Gullmar fjord) | 2-3 | 358 ± 134 (n = 2) | 7.16E-7 + 5.16E-6 (n = 2) |
| *Globobulimina turgida*[f] | (Gullmar fjord) | 3 | 565 ± 339 (n = 10) | 1.13E-6 (n = 1) |
| *Globobulimina turgida*[e] | (Gullmar fjord) | 3-5 | 310 ± 573 (n = 8) | 9.34E-6 ± 1.34E-5 (n = 8) |
| *Nonionella auris*[c] | (OMZ, Peru) | 10 | 7 ± 1 (n = 1) | 2.70E-6 (n = 1) |
| *Nonionella* cf. *stella*[f,g] | (OMZ, Chile) | 3-5 | 84 ± 33 (n = 3) | 1.62E-5 ± 6.72E-6 (n = 3) |
| *Nonionella sp.* (T1)[g] | (Gullmar Fjord) | 5 | 38 (n = 1) | n.a. |
| *Stainforthia sp.* [a] | (OMZ, Perú) | 4 | 70 (n = 1) | n.a. |
| *Uvigerina phlegeri*[a] | (Rhône) | 10 | 46 ± 2 (n = 1) | 5.48E-6 (n = 1) |
| *Uvigerina striata*[c] | (OMZ, Peru) | 6-13 | 244 ± 35 (n = 3) | 9.26E-6 ± 1.50E-6 (n = 3) |
| *Valvulineria bradyana*[a] | (Rhône) | 10 | 183 ± 10 (n = 2) | 1.22E-5 ± 1.32E-6 (n = 2) |
| *Valvulineria* cf. *laevigata*[a] | (OMZ, Perú) | 10 | 248 ± 180 (n = 2) | 1.31E-5 + 9.81E-6 (n = 2) |
| *Valvulineria inflata*[c] | (OMZ, Peru) | 2-3 | 2241 ± 1825 (n = 2) | 3.50E-5 ± 2.49 E-5 (n = 2) |


Table 2: Summary of intracellular nitrate ($NO_3^-$) storage in benthic foraminifera and gromiids from different environments. Only species where intracellular [$NO_3^-$] was at least 0.1 mM are listed. Species with intracellular [$NO_3^-$] < 0.1 mM are listed in Tab.3. Errors are given as standard error of the mean (SEM) [a]: Data from Piña-Ochoa et al. (2010b); [b]: Data from Bernhard et al. (2012b); [d]: Data from Langlet et al. (2020); [f]: Data from Risgaard-Petersen et al. (2006); [h]: Data from Høgslund et al. (2008); [i]: Data from Bernhard et al. (2012a); [j]: Data from Xu et al. (2017); [k]: Data from Glock et al. (2020); [l]: Data from Xu et al. (2021); [m]: Data from Nomaki et al. (2015).

| Species | Location | $NO_3^-$ (pmol per cell) | 1SEM | Volume ($\mu m^3 \cdot 10^{-6}$) | 1SEM | [$NO_3^-$] (mM) | 1SEM |
|---|---|---|---|---|---|---|---|
| Foraminifera | | | | | | | |
| *Allogromia* sp.[i] | Santa Barbara Basin | 570 | 354 | n.a. | n.a. | 70.0 | 49.0 |
| *Ammonia* sp.[m] | Sagami Bay | 80 | 4 | n.a. | n.a. | n.a. | n.a. |
| *Bolivina alata*[a] | Bay of Biscay | 615 | 154 | 17.0 | 1.1 | 37.0 | 12.0 |
| *Bolivina argentea*[b] | Santa Barbara Basin | n.a. | n.a. | n.a. | n.a. | 195.1 | 160.3 |
| *Bolivina* cf. *abbreviata*[a] | OMZ-Peru | 1081 | 368 | 12.0 | 2.7 | 153.0 | 49.0 |
| *Bolivina* cf. *skagerrakensis*[a] | North Sea | 83 | n.a. | 17.0 | 0.0 | 5.0 | n.a. |
| *Bolivina costata*[k] | OMZ-Peru | 34 | 4 | 0.8 | 0.0 | 43.1 | 4.3 |
| *Bolivina interjuncta*[k] | OMZ-Peru | 1239 | 267 | 15.6 | 0.5 | 80.2 | 18.9 |
| *Bolivina plicata*[a] | OMZ-Peru | 478 | 72 | 7.5 | 1.0 | 79.0 | 15.0 |
| *Bolivina robusta*[j] | Yellow Sea | 212 | 46 | 6.1 | 0.4 | 35.0 | 6.0 |
| *Bolivina seminuda*[k] | OMZ-Peru | 140 | 45 | 1.6 | 0.1 | 88.6 | 29.8 |
| *Bolivina seminuda*[a] | OMZ-Peru | 564 | 135 | 5.2 | 1.8 | 118.0 | 18.0 |
| *Bolivina spathulata*[d] | Bering Sea | 154 | n.a. | 10.3 | n.a. | 14.9 | n.a. |
| *Bolivina spissa*[m] | Sagami Bay | 190 | 72 | n.a. | n.a. | n.a. | n.a. |
| *Bolivina subaenariensis*[a] | Bay of Biscay | 285 | 46 | 25.0 | 4.3 | 44.0 | 9.0 |
| *Bolivinellina pseudopunctata*[d] | Bering Sea | 133 | n.a. | 0.9 | n.a. | 148.1 | n.a. |
| *Bulimina aculeata*[a] | Bay of Biscay | 19 | 12 | 7.4 | 0.4 | 3.0 | 2.0 |
| *Bulimina* cf. *elongata*[a] | OMZ-Peru | 817 | 287 | 7.9 | 1.2 | 116.0 | 43.0 |
| *Bulimina marginata*[l] | Yellow Sea | 70 | 11 | 2.7 | 0.3 | 26.0 | 1.0 |
| *Bulimina marginata*[a] | Skagerrak | 5 | n.a. | 1.1 | 11.0 | 0.5 | 0.2 |
| *Bulimina marginata*[a] | Bay of Biscay | 40 | 4 | 32.0 | 1.1 | 4.0 | 1.0 |
| *Bulimina subula*[l] | Yellow Sea | 79 | 8 | 1.7 | 0.3 | 51.0 | 5.0 |
| *Buliminella tenuata*[b] | Santa Barbara Basin | n.a. | n.a. | n.a. | n.a. | 217.4 | 150.5 |
| *Cancris auriculus*[j] | East China Sea | 3211 | 1046 | 28.0 | 5.1 | 114.0 | 23.0 |
| *Cancris inflatus*[a] | OMZ-Peru | 263877 | 4253 | 120.0 | 24.0 | 262.0 | 37.0 |
| *Cassidulina carinata*[a] | Rhône Delta | 3 | 1 | 4.1 | 0.2 | 1.0 | 0.5 |
| *Cassidulina* cf. *laevigata*[a] | North Sea | 21 | n.a. | 4.1 | 0.0 | 5.0 | 5.0 |
| *Cassidulina* cf. *laevigata*[a] | OMZ-Peru | 523 | 289 | 12.0 | 3.6 | 41.0 | 12.0 |
| *Cassidulina limbata*[k] | OMZ-Peru | 1408 | 710 | 16.8 | 2.9 | 72.9 | 37.8 |
| *Chilostomella oolina*[a] | Bay of Biscay | 1124 | 520 | 20.0 | 2.0 | 65.0 | 36.0 |
| *Chilostomella ovoidea*[m] | Sagami Bay | 50 | 13 | n.a. | n.a. | n.a. | n.a. |
| *Clavulina cylindrica*[a] | Rhône Delta | 2202 | 480 | 35.0 | 1.0 | 48.0 | 13.0 |
| *Clavulina cylindrica*[a] | Bay of Biscay | 1941 | 314 | 37.0 | 5.8 | 61.0 | 12.0 |
| *Cyclammina cancellata*[a] | OMZ-Peru | 45563 | 45563 | 380.0 | 3.1 | 119.0 | 118.0 |
| *Fursenkoina cornuta*[b] | Santa Barbara Basin | n.a. | n.a. | n.a. | n.a. | 125.2 | 68.9 |

| | | | | | | |
|---|---|---|---|---|---|---|
| *Globobulimina affinis*[m] | Sagami Bay | 480 | 116 | n.a. | n.a. | n.a. | n.a. |
| *Globobulimina auriculata* cf. *arctica*[a] | Greenland | 10624 | 3555 | 100.0 | 17.0 | 113.0 | 43.0 |
| *Globobulimina cf. ovula*[a] | OMZ-Peru | 3,369 | 1602 | 1.0 | 2.3 | 375.0 | 174.0 |
| *Globobulimina pacifica*[j] | East China Sea | 1167 | 455 | 75.0 | 7.0 | 16.0 | 5.0 |
| *Globobulimina pacifica*[d] | Bering Sea | 6530 | 5563 | 34.2 | 8.9 | 243.9 | 203.6 |
| *Globobulimina turgida*[f] | Gullmar fjord | 18000 | 4852 | 500.0 | 360.0 | 10.0 | 2.0 |
| *Globobulimina turgida*[a] | Skagerrak | 8192 | 1497 | 100.0 | 17.0 | 71.0 | 13.0 |
| *Goesella flintii*[a] | OMZ-Peru | 459 | 424 | 100.0 | 27.0 | 24.0 | 23.0 |
| *Gyroidina neosoldanii*[a] | OMZ-Peru | 13190 | 480 | 27.0 | 12.0 | 241.0 | 46.0 |
| *Hanzawaia nipponica*[j] | Yellow Sea | 316 | 73 | 30.0 | 0.5 | 11.0 | 3.0 |
| *Hanzawaia nipponica*[l] | Yellow Sea | 296 | 49 | 16.2 | 4.9 | 25.0 | 9.0 |
| *Hyalinea balthica*[a] | North Sea | 8 | 2 | 8.0 | 120.0 | 1.0 | 0.3 |
| *Labrospira* cf. *kosterensis*[a] | OMZ-Peru | 3139 | 845 | 51.0 | 12.0 | 57.0 | 12.0 |
| *Melonis barleeanus*[a] | North Sea | 9 | 3 | 14.0 | 20.0 | 0.6 | 0.2 |
| *Nonionella* cf. *stella*[h] | OMZ-Chile | 186 | 24 | 5.2 | 0.7 | 35.0 | 5.0 |
| *Nonionella pulchella*[d] | Bering Sea | 31 | 7 | 6.7 | 2.0 | 7.6 | 2.2 |
| *Nonionella stella*[j] | Yellow Sea | 162 | 27 | 53.0 | 3.9 | 3.0 | 0.6 |
| *Nonionella stella*[l] | Yellow Sea | 178 | 28 | 5.5 | 0.9 | 34.0 | 3.0 |
| *Nonionella stella*[b] | Santa Barbara Basin | n.a. | n.a. | n.a. | n.a. | 11.6 | 15.7 |
| *Protelphidium tuberculatum*[l] | Yellow Sea | 232 | 26 | 3.7 | 0.5 | 68.0 | 9.0 |
| *Pyrgo elongata*[a] | Rhône Delta | 43 | 14 | 47.0 | 5.8 | 0.8 | 0.2 |
| *Pyrgo williamsoni*[a] | North Sea | 5 | n.a. | 47.0 | 0.0 | 0.1 | n.a. |
| *Pyrgoella sphaera*[a] | North Sea | 6 | 1 | 47.0 | 5.8 | 0.1 | 0.0 |
| *Stainforthia* sp. var. I[a] | OMZ-Chile | 60 | 46 | 0.3 | 0.0 | 180.0 | 29.0 |
| *Textularia* cf. *tenuissima*[a] | OMZ-Peru | 450 | 432 | 11.0 | 2.9 | 43.0 | 7.0 |
| *Uvigerina akitaensis*[m] | Sagami Bay | 210 | 73 | n.a. | n.a. | n.a. | n.a. |
| *Uvigerina elongatastriata*[a] | Bay of Biscay | 274 | 244 | 5.1 | 0.6 | 60.0 | 55.0 |
| *Uvigerina mediterranea*[a] | Bay of Biscay | 101 | 66 | 20.0 | 6.6 | 6.0 | 4.0 |
| *Uvigerina peregrina*[d] | Bering Sea | 74 | 20 | 9.9 | 4.1 | 10.0 | 4.7 |
| *Uvigerina peregrina*[a] | North Sea | 332 | 184 | 20.0 | 6.6 | 16.0 | 9.0 |
| *Uvigerina phlegeri*[a] | Rhône Delta | 444 | 44 | 8.4 | 0.2 | 209.0 | 48.0 |
| *Valvulineria bradyana*[a] | Rhône Delta | 1268 | 164 | 15.0 | 1.4 | 95.0 | 15.0 |
| *Valvulineria* cf. *laevigata*[a] | OMZ-Peru | 865 | 640 | 19.0 | 3.7 | 25.0 | 12.0 |
| *Valvulineria inflata*[k] | OMZ-Peru | 17666 | 5319 | 135.4 | 16.4 | 120.1 | 34.1 |
| *Verneuilinulla advena*[l] | Yellow Sea | 86 | 15 | 2.5 | 0.3 | 34.0 | 3.0 |
| Gromiids | | | | | | | |
| *Gromia* sp.[a] | Bay of Biscay | 2846 | 1275 | 93.0 | 20.0 | 35.0 | 21.0 |
| *Gromia* sp.[a] | Skagerrak | 35277 | 16546 | 510.0 | 110.0 | 53.0 | 19.0 |
| *Gromia* sp.[a] | Rhône Delta | 3889 | 1024 | 160.0 | 110.0 | 91.0 | 26.0 |
| *Gromia* sp.[a] | North Sea | 14682 | 4649 | 160.0 | 3500.0 | 140.0 | 46.0 |
| *Gromia* sp.[a] | Greenland | 12997 | 2954 | 80.0 | 23.0 | 163.0 | n.a. |
| *Gromia* spp.[d] | Bering Sea | 367 | 85 | 11.3 | 6.1 | 40.2 | 14.1 |

Table 3: Summary of benthic foraminifera from different environments that lack intracellular nitrate ($NO_3^-$) storage. Only species with intracellular $[NO_3^-] < 0.1$ mM are listed. Species in **bold** letters have been found to store $NO_3^-$ in other environments (see table 2). [a]: Data from Piña-Ochoa et al. (2010b); [j]: Data from Xu et al. (2017); [m]: Data from Nomaki et al. (2015). *: *Ammonia tepida* is a morphogroup of pseudocryptic species that recently had a revision. Specimens earlier identified as *A. tepida* are likely either *A. aberdoveyensis* or *A. confertitesta* according to Hayward et al. (2021).

| Species | Location | Species | Location |
|---|---|---|---|
| Agglutinated sp.[a] | Rhône Delta | *Hippocrepinella alba*[a] | Skagerrak |
| *Ammonia beccarii*[a] | Rhône Delta | ***Hyalinea balthica***[a] | North Sea |
| *Ammonia beccarii*[a] | Bay of Biscay | Komokiacea[a] | OMZ-Peru |
| *Ammonia* sp.[a] | Limfjorden | *Labrospira* cf. *L. subglobosa*[a] | OMZ-Peru |
| ***Ammonia* sp.**[m] | Sagami Bay | ***Melonis barleeanus***[a] | Rhône Delta |
| *Ammonia tepida*\*[a] | Aiguillon Bay | *Nonion scaphum*[a] | Rhône Delta |
| *Arenoparella asiatica*[j] | Yellow Sea | *Nonion scaphum*[a] | Bay of Biscay |
| *Bathysiphon* cf. *argenteus*[a] | OMZ-Peru | *Nouria polymorphinoides*[a] | Bay of Biscay |
| *Bathysiphon minutus*[a] | Skagerrak | *Pelosina variabilis*[a] | Skagerrak |
| *Biloculinella depressa*[a] | North Sea | *Pseudoeponides falsobeccarii*[a] | Rhône Delta |
| *Bolivinita quadrilatera*[a] | Bay of Biscay | *Quinqueloculina seminulum*[a] | Skagerrak |
| ***Bulimina aculeata***[a] | Rhône Delta | *Quinqueloculina seminulum*[a] | Bay of Biscay |
| ***Bulimina marginata***[a] | Rhône Delta | *Quinqueloculina seminulum*[a] | Rhône Delta |
| *Cibicidoides pachyderma*[a] | Bay of Biscay | *Quinqueloculina* sp.[a] | OMZ-Perú |
| *Crithionina hispida*[a] | OMZ-Peru | *Reophax micaceus*[a] | Bay of Biscay |
| *Cyclammina cancellata*[a] | Bay of Biscay | *Reophax* sp.[a] | OMZ-Perú |
| *Cypris subglobosus*[a] | Bay of Biscay | *Rhabdammina inaequalis*[a] | North Sea |
| *Dentalina* sp.[a] | Rhône Delta | *Saccammina* sp.[a] | Bay of Biscay |
| *Epistominella exigua*[a] | OMZ-Peru | *Technitella legumen*[a] | Skagerrak |
| *Gyroidina altiformis*[a] | Bay of Biscay | *Triloculina tricarinata*[a] | North Sea |
| *Haynesina germanica*[a] | Aiguillon Bay | ***Uvigerina peregrina***[a] | Bay of Biscay |

Figure 1:

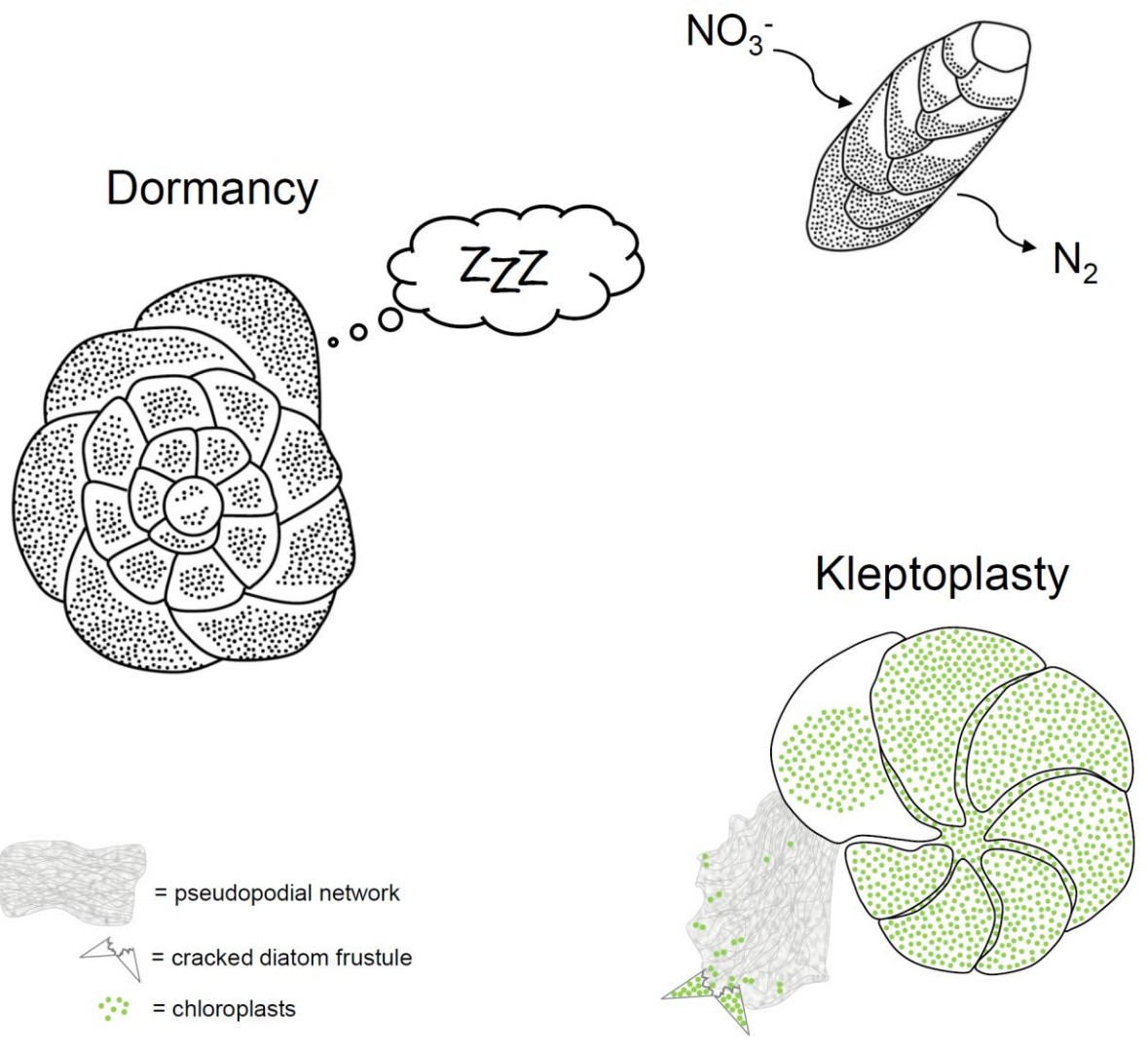

Fig. 1: Schematic representations for three survival strategy examples performed by benthic foraminifera under $O_2$ depleted conditions.

Figure 2:

$$2\,NO_3^- \xrightarrow{Nar} 2\,NO_2^- \xrightarrow{Nir} 2\,NO \xrightarrow{Nor} N_2O \xrightarrow{Nos} N_2$$

$$\searrow^{Nod}$$

$$N_2 + O_2$$

*Nar* = Nitrate reductase
*Nir* = Nitrite reductase
*Nor* = Nitric oxide reductase
*Nos* = Nitrous oxide reductase
*Nod* = Nitric oxide dismutase

Fig. 3: Schematic view of two alternative pathways suggested for foraminiferal denitrification. Abbreviations indicate the enzymes that are catalyzing the respective step (see legend). Enzymes in black have been found transcribed by eukaryotic (foraminiferal) RNA (Woehle & Roy et al., 2018, Orsi et al., 2020; Gomaa et al., 2021). Enzymes in grey are missing in the foraminiferal denitrification pathway and are likely performed by bacterial symbionts (Woehle & Roy et al., 2022). The straight pathway above describes the normal heterotrophic denitrification pathway. The junction, catalyzed by the Nod, which produces $O_2$, has been suggested as an alternative pathway for foraminiferal denitrification (Woehle & Roy et al., 2018).

Figure 3:

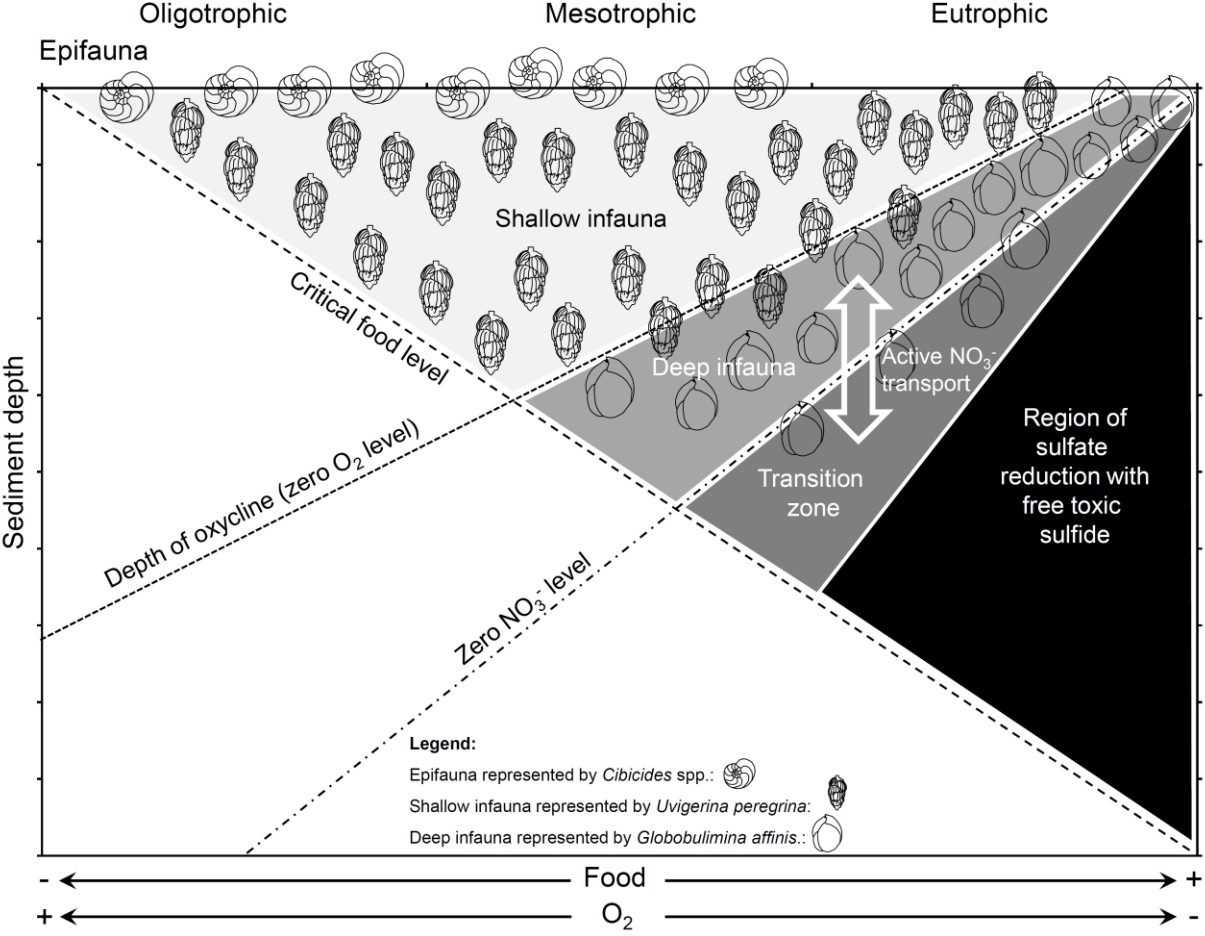

Fig.4: TROX model modified after Jorissen et al. (1995) and Xu et al. (2021). The supply of organic matter and bottom water $O_2$ and $NO_3^-$ concentrations in different environments control the penetration depth of $O_2$ and $NO_3^-$ into the sediment. Benthic foraminifera choose their microhabitat according to their metabolic preferences for $O_2$ or $NO_3^-$ as an electron acceptor and the availability of food. Intermediate infauna is not specifically schematized in the figure but peak between the shallow and deep infauna with an overlap to both directions and often peak within the $NO_3^-$ maximum (Jorissen et al., 2022). Note that denitrifying foraminifera can actively transport intracellular $NO_3^-$ below the $NO_3^-$-penetration depth in the sediments. The deeper regions where production of free sulfide occurs will mainly be avoided. For further details see text.

Figure 4:

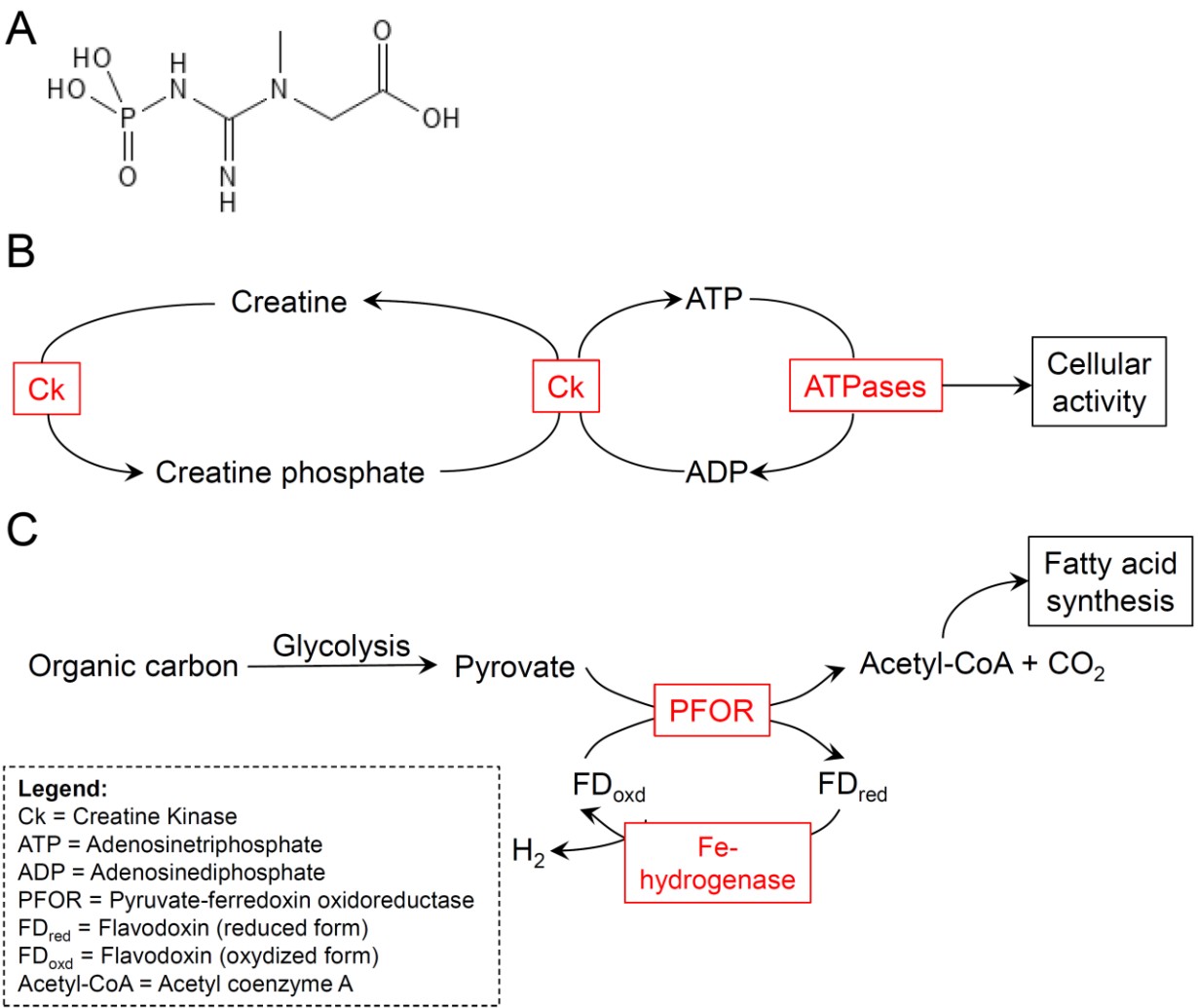

Fig. 5: Examples for molecules and processes that are relevant in the anaerobic metabolism of foraminifera. A: Structural formula of creatine phosphate. B: The role of creatine kinase (Ck) and creatine phosphate in the anaerobic metabolism. High energy creatine phosphate is produced by phosphorylation of creatine. Creatine phosphate can rapidly recycle ADP to ATP to provide resources for rapid energy bursts. This pathway has been described by Orsi et al.(2020). C: Fermentation has been found to be relevant in the anaerobic metabolism of foraminifera by both Orsi et al.(2020) and Gomaa et al. (2021). The possibility of a $H_2$ producing fermentation pathway, catalyzed by Fe-hydrogenase has been described by Gomaa et al. (2021).

Figure 5:

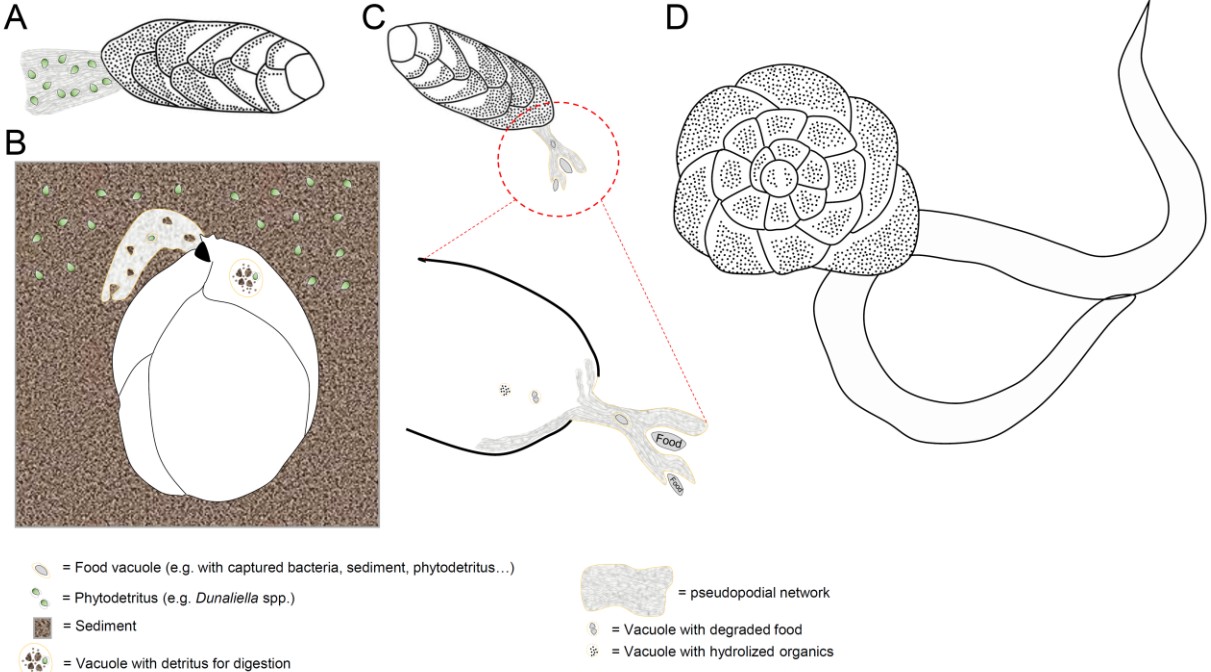

= Food vacuole (e.g. with captured bacteria, sediment, phytodetritus...)

= Phytodetritus (e.g. *Dunaliella* spp.)

= Sediment

= Vacuole with detritus for digestion

= pseudopodial network

= Vacuole with degraded food

= Vacuole with hydrolized organics

Fig. 6: Schematic representation of different trophic strategies by foraminifera: A: Strict herbivore species only ingest fresh phytodetritus. B: Non selective detrivores and seasonal herbivores ingest and partly digest detritus from the surrounding sediment (omnivores). C: Schematic representation of phagocytosis. Recent studies showed that benthic foraminifera from $O_2$ depleted habitats have the capacity of phagocytosis even under anoxia (Orsi et al., 2020). D: Schematic representation of *Ammonia* sp. preying on a nematode. Some omnivoric benthic foraminifera are known to prey on meiofauna (e.g., Dupuy et al., 2010) and there is first evidence, that some species can be carnivores even under $O_2$ depleted conditions (Glock et al., 2019a; Panagiota-Chronopoulou et al., 2019; Schweizer et al., 2022).

Figure 6:

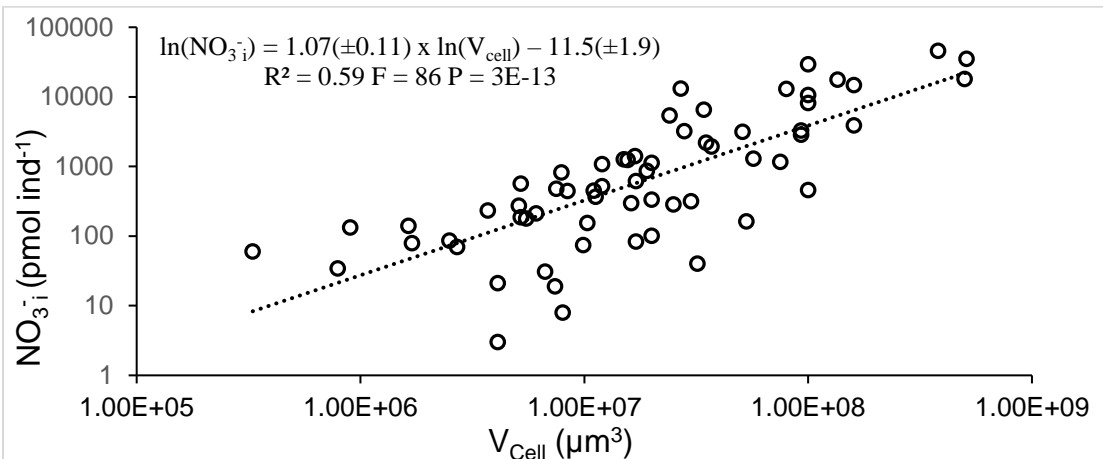

$$\ln(NO_{3\,i}^-) = 1.07(\pm0.11) \times \ln(V_{cell}) - 11.5(\pm1.9)$$
$$R^2 = 0.59 \; F = 86 \; P = 3E\text{-}13$$

Figure 7: Log-log plot and power regression of intracellular $NO_3^-$ content ($NO_{3\,i}^-$) against the biovolume ($V_{cell}$) of benthic foraminifera from diverse environments (Tab. 2). Only species with an intracellular $[NO_3^-] \geq 1$ mM were considered for the power regression.