# Peer review of "Reviews and syntheses: Benthic foraminifera and gromiids from oxygen depleted environments - Survival strategies, biogeochemistry and trophic interactions"

_Biogeosciences, 2023_

## Author Comment (AC1)

Dear reviewers, Dear Lisa Levin (Editor),

thank you very much for the feedback to this review paper. I am really glad that three experienced specialists from this field provided such detailed constructive revisions to the manuscript. Of course, a review article will strongly benefit from discussing different opinions and different points of view of several experts. I revised my manuscript thoroughly regarding the feedback to all three reviewers. Below you can find a detailed point by point response to the review by Frans Jorissen.

**Reviewer:** The review paper of Glock addresses concerns an exciting and rapidly evolving field of research. It comes very timely, and presents a good overview and perfect starting point for all scientists that want to know more about foraminifera living in anaerobic environments.

However, this first version is not perfect yet, and in my opinion, several points need to be addressed:

**1. The taxonomy used in this paper**.

Although taxonomy is not a central topic here, in order to avoid confusion, justice should be done to the important recent advances, largely resulting from the contribution of molecular studies. One of the problems is the treatment of species of the genus In some parts of the text, the author talks about *A. tepida*, a species name that was often used in the past for a complex of cryptic and pseudocryptic species that has only be sorted out recently. On lines 189-200 the author gives more detailed information and says that in that case, the text concerns "*Ammonia* sp. (T6)", which is an informal name of a phylotype previously placed in the *A. tepida* morphocomplex. According to the most recent revision of this group of species (Hayward et al., 2021, *Micropaleontology*, 67; p. 109-313) the correct name of this phylotype is *Ammonia confertitesta*. I think that in the whole text, the taxonomy of Ammonia should be standardized, according to the 2021 revision of Hayward and coauthors.

A very similar case concerns the morphospecies *Uvigerina peregrina*. Also here, we are very probably confronted with a pseudocryptic species complex, but in case, things haven't been sorted out yet. The consequence is that is impossible to treat U. peregrina as a single biological species, as the author is doing. When the author writes (lines 86-87) "*Also, the NO3 - storage in U. peregrina shows a high variability, depending on the environment.*", it is very well possible that different biological species are concerned, and that the differences in nitrate storage are species-specific and have nothing to do with the environmental conditions.

A very similar situation, of species complexes with a multitude of cryptic and pseudocryptic species can be expected in the genera *Nonionella* and *Globobulimina*.

Although there is no ready solution for this problem, the author should take it into account, and realise that different species may be designed by the same species name, or inversely, different species names may indicate the same biological species. The author should show in the text that he is aware of this potential problem, and some of the conclusions should be reconsidered and be formulated with somewhat more reserve.

**Reply:** This is indeed a complex problem and it might be complicated to find a ready solution how to deal with older literature that considered these morphogroups as single species with ecophenotypic plasticity. I tried to address this topic in several parts of the revised manuscript and also tried to adapt the latest revision of the *Ammonia tepida* taxonomy. When *A. tepida* is mentioned in tables, I mark them with a * and add into the figure caption:

"*Ammonia tepida* is a morphogroup of pseudocryptic species that recently had a revision. Specimens earlier identified as *A. tepida* are likely either *A. aberdoveyensis* or *A. confertitesta* according to Hayward et al. (2021)."

When *A. tepida* is mentioned in the text, I also tried to address this problem:

"However, not all benthic foraminifera are able to denitrify, even if they live in environments that are periodically exposed to anoxia such as representatives of the intertidal species morphogroup *Ammonia tepida* (either *Ammonia venata, Ammonia aberdoveyensis* or *Ammonia confertitesta* according to Hayward et al., 2021), which neither store $NO_3^-$ nor show any denitrification activities (Piña-Ochoa et al., 2010b)."

"LeKieffre et al. (2017) did a feeding experiment with specimens from the *Ammonia tepida* morphogroup (*A. confertitesta* according to Koho et al., 2018 and Hayward et al., 2021) using a $^{13}C$-labeled diatom film as food source."

Also, I wrote a paragraph, where I discuss, that this problem might concern other taxa, too:

"The observations that some species store $NO_3^-$ and denitrify in some environments and in others not might have two reasons. One reason could be that these species belong to an opportunistic group of foraminifera that can well adapt to both oxygenated environments where they respire $O_2$ and do not denitrify and $O_2$ depleted environments where they switch to denitrification. The other reason could be that some of these foraminifera belong to morphogroups that are identified as a single species but indeed are a mixture of cryptic and pseudocryptic species that include denitrifying and non-denitrifying species. An example for such a morphogroup that has recently had a revision is *A. tepida*. This morphogroup includes three species (*Ammonia venata*, *A. aberdoveyensis* or *A. confertitesta*) that now can be morphologically distinguished (Richirt et al., 2019; Hayward et al., 2021). A similar case concerns the morphogroup *Nonionella stella*, where representatives have been found to denitrify (Høgslund et al., 2008; Choquel et al., 2021) but also consists out of several cryptic to pseudocryptic species (Deldicq et al., 2019). The situation might be similar with other *Nonionella* species and the widespread species *U. peregrina*."

**Reviewer:**

**2. The synonymising of microhabitat and metabolic categories.**

Both in lines 146-165 and in figure 4, the author states that epifaunal taxa are aerobic, shallow infaunal taxa are facultative anaerobes whereas deep infauna are facultative aerobes. Although this is certainly partly true, this seems to be a major oversimplification, for which the author gives insufficient justification.

**Reply:** This section indeed suffered a bit from overgeneralizations by my side. I tried to clarify, that there definitely are exceptions from this concept within each category of microhabitat. Detailed examples are discussed in each of the microhabitat sections. Before the individual microhabitat sections I added the following sentence:

"Due to our increasing understanding about the anaerobic metabolism of foraminifera we can now assume that $NO_3^-$ availability is another controlling factor (Fig.4). This is also indicated by a study coupled early diagenetic modeling with foraminiferal ecology to model the microhabitats of benthic foraminifera (Jorissen et al., 2022). According to their metabolic preference for $NO_3^-$ or $O_2$ as electron acceptors many benthic foraminifera species that typically occupy a certain microhabitat (epifauna, shallow infauna and deep infauna) might **partly** be assigned to three different attributes (Aerobe,

facultative anaerobe and facultative aerobe). **Most likely there are exceptions to these classifications that will be discussed below. Another controlling factor on the microhabitat is most likely the trophic strategy of the foraminiferal species, which is further discussed in section 3.**"

**Reviewer:** First, it is not evident that all "epifauna" have necessarily an aerobic metabolism. In cases where bottom waters are strongly hypoxic, some typically epifaunal taxa may very well be facultative anaerobes or even facultative aerobes. As an example, I think of *Epistominella smithi* from the California foreland basins and the Peruvian margin. This large-sized species, which has never been described from well oxygenated sites, can reach high densities in strongly hypoxic settings, and its morphology very strongly suggests an epifaunal lifestyle.

**Reply:** This is a good point. I already discussed in the original part of this section that certain *Cibicides* spp. might be able to denitrify under certain circumstances. Now, I tried to integrate the example suggested by the reviewer and another epifaunal species *Planulina limbata* that also seems to be restricted to low oxygen environments. Also I adapted the first sentence of this section to prevent overgeneralization:

"Many **epifaunal species** can most likely be considered as **aerobes** that typically occur at the sediment-water interface or on elevated surfaces."

This is the new part that describes examples of epifaunal species that are restricted to low oxygen environments:

"In the same way as for the other microhabitats, not all species with an epifaunal lifestyle should be automatically considered as aerobes. There are examples of epifaunal benthic foraminifera that have not been found in well oxygenated environments but reach high abundances in $O_2$ depleted environments. One example is *Epistominella smithi*, which has been described in low $O_2$ environments, such as the Santa Barbara Basin (Harman, 1964) or the Peruvian OMZ (Erdem and Schönfeld, 2017). Nevertheless, the morphology of *E. smithi* strongly suggests an epifaunal lifestyle. Another example is the epifaunal species *Planulina limbata*. This species is abundant only in $O_2$ depleted environments on continental margins within the East Pacific (Natland, 1938; Erdem and Schönfeld, 2017; Glock et al., 2022). Recent *P. limbata* specimens are present in severely $O_2$ depleted water masses within the Peruvian OMZ ($[O_2]$ = 3 - 12 µmol/kg, Glock et al., 2022). Nevertheless, *P. limbata* adapts its pore density to the availability of $O_2$ (Glock et al., 2022), which might indicate that it has an aerobic metabolism, despite that its presence appears to be limited to low $O_2$ environments. Another possibility is that species such as *E. smithi* or *P. limbata* may denitrify under certain circumstances and therefore can also be considered as facultative anaerobes. Hopefully, measurements of metabolic rates, intracellular nutrient content and enzymatic activity might bring further evidence in the future, if at least some epifaunal species can switch to an anaerobe metabolism, when $O_2$ is too depleted."

**Reviewer:** Next, it is not evident either that all shallow infaunal taxa are facultative anaerobes, some definitely are not. Representatives of the *Ammonia tepida* morphogroup (*A. aberdoveyensis, A. confertitesta*) are good examples. Until to date, all tests trying to show anaerobic metabolism were negative for these taxa. Nevertheless, they are found massively in deeper sediment layers in estuarine mudflats, were they may survive the most adverse conditions by dormancy. But I would definitely consider them as shallow infaunal taxa with an aerobic metabolism.

**Reply:** Again, this is a good point and overgeneralizations are certainly not very helpful. Within the revised manuscript I tried to clarify that, of course, we cannot generalize that all shallow infaunal species are facultative anaerobes. Regarding the *Ammonia tepida* morphogroup: It is true that they neither can store nitrate nor denitrify, but it might be that they have other adaptations to withstand $O_2$ depletion. Recent genetic studies indicated that some foraminifera have the capacity for ananaerobe metabolism by using fermentation. Other recent studies revealed that several foraminifera species store large amounts of phosphate, which is likely another adaptation to $O_2$ depletion. Actually, I found that specimens of the *Ammonia tepida* morphogroup from tidal mudflats

have the highest intracellular phosphate concentrations of more than 20 species I analyzed from various environments. Since these are unpublished results for a manuscript in preparation, I cannot discuss those in this review but there is actually a lot of evidence that these *Ammonia* species have adaptations to stay mobile, even within anoxia. I adapted the first sentence of this paragraph to prevent overgeneralization and added several sentences, where I discussed possible exceptions:

"Shallow infauna can **in many cases** be considered as facultative anaerobes that are well adapted to the presence of low $O_2$ concentrations but can switch to denitrification if they are exposed to anaerobic conditions or need to enter the deeper sediment parts to find food or avoid competitive stress."

And the text that discusses the exceptions:

"Of course, it cannot be generalized that all foraminifera from a shallow infaunal habitat are indeed facultative anaerobes. At least some species that can be considered shallow infaunal have been shown neither to be able to store $NO_3^-$ nor to denitrify. As mentioned above all specimens from the *Ammonia tepida* morphogroup that have been analyzed so far lack an intracellular $NO_3^-$ storage and cannot denitrify (Piña-Ochoa et al., 2010b). Nevertheless, these taxa are often exposed to anoxia and can sometimes even be found alive in 4 to 26 cm sediment depth (Alve and Murray, 2001; Thibault de Chanvalon et al., 2015). It is possible that these foraminifera indeed only have an aerobe metabolism and just become dormant under exposure to anoxia (dormancy is discussed in another section). Though, another possibility is that intertidal species such as *A. venata*, *A. aberdoveyensis* or *A. confertitesta* have other adaptations to anoxia than denitrification. Recent studies revealed other possible anaerobic metabolic pathways in foraminifera such as fermentation or dephosphorylation of creatine phosphate which are discussed in section 2.1.4 (Orsi et al., 2020; Gomaa et al., 2021). Eventually, studies on the transcriptome of non-denitrifying species from infaunal environments might be able to show, if some of these species can switch an alternative anaerobe metabolism under exposure to anoxia."

**Reviewer:** Third, I have the least problems with considering deep infaunal taxa as facultative aerobes. But there is actually not a lot of evidence for the fact this. You would expect this, but it would be good if the author could give some arguments to strengthen this point.

**Reply:** I extended this part significantly to provide more arguments (new parts are marked in red):

"**Deep infaunal** species can most likely be considered as **facultative aerobes** that have a metabolic preference of $NO_3^-$ over $O_2$ (Glock et al., 2019c) and try to avoid trace amounts of $O_2$. They cannot be accounted as obligate anaerobes, though, since they can withstand periods of oxygenation. Many experiments show that denitrifiying foraminifera can switch to $O_2$ respiration, if they are exposed to $O_2$ (i.e. Piña-Ochoa et al., 2010b). Still, they follow the oxycline in the sediments to avoid the inhibition of denitrification by trace amounts of $O_2$. The $\delta^{13}C$ signature of shells of deep infaunal globobuliminids also indicates that they calcify in sediment depth where the pore water $O_2$ level reaches zero or even deeper in the sediments. The offset between $\delta^{13}C$ of *Globobulima* spp. tests and $\delta^{13}C$ of epifaunal foraminifera or of bottom water dissolved inorganic carbon (DIC) is nearly equal to the offset between DIC at the zero $O_2$ layer and the bottom water (Schmiedl and Mackensen, 2006) and often can be even higher (Costa et al., 2023), indicating that many globobuliminids live even below the oxycline. Even though they can switch to $O_2$ respiration (Piña-Ochoa et al., 2010b), these species most likely would try to avoid crossing the oxycline since denitrification would be already inhibited by nM $O_2$ concentrations (Dalsgaard et al., 2014) and the $O_2$ concentration slightly above the oxycline is not high enough to fulfil their metabolic demands. Indeed, the model by Jorissen et al. (2022) describes the distribution of deep infauna very well, by using the presence of $O_2$ as an inhibiting factor, which also promotes that they can rather be considered faculatative aerobes instead of facultative anaerobes. Taxa belonging to the deep infaunal group that might be considered as facultative aerobes that prefer $NO_3^-$ over $O_2$ include for example *Valvulineria inflata* and *bradyana*,

*Bolivina seminuda*, *Globobulimina pyrula* and *Cancris carmenensis* (e.g. Jorissen et al., 1995; Mojtahid et al., 2010; Glock et al., 2019c)."

**Reviewer:** Finally, I strongly regret that the author doesn't mention "intermediate infauna". The maximum abundance of these taxa is systematically found in the nitrate maximum zone; *Melonis barleeanus* is a typical example. These taxa are interesting, because most of them do not store nitrate (maybe because there is no reason to do so when you live in the nitrate maximum), but appear to be capable to denitrify.

**Reply:** In the revised version, I will add a small paragraph about the intermediate infauna. I also discuss now that they live close to the $NO_3^-$ maximum, since $NO_3^-$ increases by remineralization before it decreases again due to denitrification (Jorissen et al., 2022). Though, denitrification would be likely inhibited by the presence of $O_2$ in these depths. It could be a possibility that it simply does not store $NO_3^-$ but denitrifies the $NO_3^-$ directly after the uptake from the seawater, when $O_2$ decreases below the inhibition threshold. This definitely provides some questions for future research. I added the following paragraph to the revised manuscript:

"The **intermediate infauna** is somehow an exceptional case. Common representatives of intermediate infaunal taxa are *Melonis barleeanus* or *Pullenia* spp. (Corliss, 1991). The typical example for intermediate infaunal species *M. barleeanus* is interesting, since it either stores no or only very small amounts of $NO_3^-$ (See table 2 and 3). Still, several studies indicate that *M. barleeanus* lives deeper in the sediments than some *Uvigerina* spp. (Corliss, 1991; Ní Fhlaithearta et al., 2018) although many *Uvigerina* species have been shown to store $NO_3^-$ and denitrify (Tab. 1&2). This might give room to speculate if *M. barleeanus* has other metabolic adaptations to $O_2$ depletion than denitrification or if it simply does not store large amounts of $NO_3^-$ but denitrifies $NO_3^-$ directly after the uptake from the seawater. Indeed, a recent study predicted the microhabitats of infaunal benthic foraminifera using an early diagenetic model and showed that the intermediate infauna clusters around the $NO_3^-$ maximum in the pore water (Jorissen et al., 2022). Future perspectives on understanding the biology of intermediate infauna might include transcriptome analyses to decipher other anaerobe metabolic pathways and testing the denitrification capacity after incubation in $NO_3^-$-free and $NO_3^-$-containing seawater."

**Reviewer:** And a detail: the legend of figure 4 is way too long, and repeats the running text.

**Reply:** Agreed. The figure caption was indeed too long. I shortened it for the revised version. In addition, I removed the terms "aerobic", facultative

**3. Chapter 5: Applications in paleoceanography**

**Reviewer:** This chapter didn't convince me. The texts are very concise, and repeatedly the reader refers to the paper of Hoogakker et al., who treats this topic in much more detail. As it is, this chapter doesn't add anything useful. I think the author should delete it altogether and expand some of the topics that would become more robust with a more deep-going treatment.

**Reply:** This is a good point. I deleted chapter 5 and condensed the whole chapter to one paragraph, that is now moved into the introduction:

"Benthic foraminifera from low $O_2$ environments have also been established as an invaluable archive for paleoceanography. However, I will touch on summarizing redox proxies based on benthic foraminifera only briefly, since there is work in progress to give a comprehensive review about proxies for $O_2$ concentrations in paleoceanography (Hoogakker et al., in prep). Due to their ability to precipitate their calcitic tests even under anoxic conditions, fossil benthic foraminifera became routine tools in paleoceanography to reconstruct past redox conditions (Nardelli et al., 2014; Orsi et al., 2020). Some

morphological adaptations are very common for benthic foraminifera that thrive in $O_2$ depleted habitats. Small, more elongated and flattened morphologies are often characteristic for $O_2$ depletion, while more spherical forms can indicate oxygenated conditions (Bernhard, 1986; Bernhard et al., 1997). In addition, high porosity and thin test walls seem to be characteristic for foraminifera that live in low $O_2$ environments (Kaiho, 1994). The porosity, including pore size and pore density, of foraminiferal tests recently received more attention as possible paleoceanographic tool. Different foraminiferal species seem to adapt their pore characteristics in a different way to environmental conditions. *Cibicides* spp. for example mainly thrive in well oxygenated environments (Mackensen et al., 1995) and the porosity in epifaunal *Cibicides* spp. and *Planulina* spp. is significantly negatively correlated to the $O_2$ concentrations in the bottom water (Rathburn et al., 2018; Glock et al., 2022). If $O_2$ is too depleted, these foraminifers increase their porosity to optimize the $O_2$ uptake. Furthermore, the mechanism of biomineralisation in foraminifera can preserve the chemical signature of ambient seawater in their test calcite. These species precipitate their test calcite directly from vacuolized seawater (Erez, 2003; de Nooijer et al., 2014; Toyofuku et al., 2017)) and thus the chemical composition of the test calcite reflects the chemical composition of the surrounding water in their habitats. Different element/Ca ratios are used as proxy for various parameters. Over the past decades several redox sensitive element/Ca ratios in foraminiferal calcite were identified as potential $O_2$ proxies, where Mn/Ca (Reichart et al., 2003; Barras et al., 2018; Brinkmann et al., 2021) and I/Ca (e.g. Zhou et al., 2014, 2022; Lu et al., 2016; Glock et al., 2019d; Winkelbauer et al., 2021; Cook et al., 2022) are amongst the most prominent examples. The offset of the stable carbon isotope fractionation ($\delta^{13}C$) between the tests of epifaunal and deep infaunal benthic foraminifera can also be used as a quantitative $[O_2]_{BW}$ proxy (e.g. McCorkle and Emerson, 1988; Schmiedl and Mackensen, 2006; Hoogakker et al., 2014, 2018). Finally, species compositions of benthic foraminifera assemblages are used to reconstruct past environmental conditions. Kaiho et al. (1994) developed the first benthic foraminifera $O_2$ index (BFOI). Further development of this index is still going on with recent developments by Tetard et al., 2021 and Kranner et al., 2022."

**4. Minor points:**

**Reviewer:** The title: I think that it is somewhat overdone: I don't understand the plural form of the first words, and I would say that a review is always a synthesis. I would recommend something like : "Foraminifera from anaerobic environments - survival strategies, biogeochemistry and ecology – an overview".

**Reply:** A similar recommendation has been given by anonymous reviewer #2. To be honest, I agree with these statements and did not have "Reviews and syntheses" in my original title. Though, after submission to Biogeosciences as a review paper, I had the request by the editorial office to change my title accordingly. "Reviews and syntheses" is mandatory in the title of review papers that are submittet to Biogeosciences.

**Reviewer:** The cartoons presented in figures 1 and 6: these drawings are very nice (although somewhat simplistic), would be perfect in a text for a larger public, but are in my opinion not suitable for a scientific review paper.

**Reply:** I disagree in this point and actually corresponded with some other colleagues about exactly this point before submitting the paper. A review paper is often a starting point for either early career scientists or scientists that have to become acquainted with a new topic in their field. For both groups it is much easier to memorize context by looking at figures that are easy to follow and not too sterile. There are enough original papers out there that show SEM images of foraminifera, which are surely aesthetic but can also be very sterile. Nevertheless, I tried to keep the "cartoons" scientifically

correct. If the reviewers find anything in the figures misleading or plain wrong: Please let me know and I correct everything accordingly.

**Reviewer:** Tables 2 and 3: these two tables are really the heart of this review paper. First, it is essential that the authors make it very clear that these tables are exhaustive, that they contain ALL published info available today. For me this became only progressively clear when reading the text. Next, unlike table 3, which is perfect, table 2 is a mess, all taxa are mixed without any visible order. Like for table 3, the taxa should be presented in alphabetical order. Finally, the signs used to indicate the different studies from which the data are taken are certainly artistic, but also unreadable. My mind is totally unfit to memorise such symbols, and I think it will be the same with many of our colleagues.

**Reply:** In table 2 the taxa were already in alphabetical order but they were not in table 1. I changed this and the taxa are sorted in alphabetical order in table 1, too. In addition, I substituted the cryptic signs that indicate the references with superscripted letters in brackets behind the species names. Every reference got its own letter, which has been kept in all the different tables. For example: Piña-Ochoa et al., 2010b is now indicated as "[(a)]" in all the tables.

**Reviewer:** On several places the author is going too quickly, simply backs up a statement with a reference, but doesn't tell on what evidence the statement is based. Some examples:

Lines 118-119: "It is noteworthy that denitrifying foraminifera from the Peruvian OMZ show a metabolic preference of NO3 - over O2 as an electron acceptor (Glock et al., 2019c)."

**Reply:** This part has been extended for clarification:

"It is noteworthy that denitrifying foraminifera from the Peruvian OMZ show a metabolic preference of $NO_3^-$ over $O_2$ as an electron acceptor (Glock et al., 2019c). These foraminifera show an increasing cell volume with increasing ambient $NO_3^-$ and decreasing $O_2$ concentrations. Similar observations have been made at the Califorinian Borderlands, where some benthic foraminifera increase their cell-volume with decreasing ambient $O_2$ concentrations (Keating-Bitonti and Payne, 2017). Additional evidence for the metabolic $NO_3^-$ preference came from comparing denitrification and $O_2$ respiration rates and scaling them to their cell volume (Glock et al., 2019c). The scaling is lower for $O_2$ respiration than for denitrification, indicating that the $NO_3^-$ metabolism during denitrification is more efficient than the $O_2$ metabolism during aerobic respiration in foraminifera from the Peruvian OMZ. This might explain, why some infaunal denitrifying foraminifera follow the oxycline within sediments (Linke and Lutze, 1993; Duijnstee et al., 2003)."

**Reviewer:** Lines 125-126: "Larger amounts of O2 might supply this demand but also harm the cell."

**Reply:** Added a bit text for clarification:

"Larger amounts of $O_2$ might supply this demand but also harm the cell. For example, $O_2$ can inhibit the growth of some obligate anaerobes poison enzymes that are important for their metabolism (Lu and Imlay, 2021). Also for aerobes $O_2$ can be harmful. "Hyperoxia", an excess supply of $O_2$, leads to damaging effects by highly-reactive metabolic products of $O_2$ (free $O_2$ radicals) that inactivate enzymes in the cell, damage DNA and destroy lipid membranes (Frank and Massaro, 1980)."

**Reviewer:** Lines 134-135: "The fact that some foraminifera prefer NO3 over O2 as electron acceptor (Glock et al., 2019c)".

**Reply:** See above.

**Reviewer:** Line 147: "since they are also able to respire O2."

**Reply:** Deleted this part of the sentence and added the following two sentences for clarification:

"They cannot be accounted as obligate anaerobes, though, since they can withstand periods of oxygenation. Many experiments show that denitrifiying foraminifera can switch to $O_2$ respiration, if they are exposed to $O_2$ (i.e. Piña-Ochoa et al., 2010b)."

**Reviewer:** In none of these cases the author explains on what evidence these statements are based. Many similar examples are present in the text.

In all these cases, I would like that the author explains what evidence these important conclusions are based on. Since this is a review paper, the author should not oblige the reader to find such important information in another paper.

**Reply:** For all examples that have been addressed by the reviewer I added some sentences and references for further clarification.

**Reviewer:** Lines 116-117: I feel that there is a slight contradiction here. On lines 92-94 the author indicates that foraminifera can denitrify without bacterial symbionts, whereas from lines 110-117 I understand that complete denitrification is not possible without bacterial symbionts.

**Reply:** That´s again a valid point. Bernhard et al. (2012b) did not measure $N_2$ production after the antibiotic treatment but nitrate consumption. This is not a proof for complete denitrification but a strong evidence, since the rate was similar as in the species that have been shown to denitrify. To discuss this controversy, I added the following text:
"Recent metagenomics and transcriptomics results  of denitrifying foraminifera indicate that bacterial symbionts might perform the missing steps in the foraminiferal denitrification pathway or that they at least partly contribute to the amount of $NO_3^-$ that is denitrified within foraminiferal cells (Woehle & Roy et al., 2022). This seems contradictory to the results by Bernhard et al. (2012b) who showed that *B. argentaea* consumed its intracellular $NO_3^-$ storage (likely for denitrification) even after the antibiotics treatment. Gomaa et al. (2021) confirmed that *B. argentaea* also lacks the first and last denitrification step in its transcriptome, although it lacks intracellular bacterial symbionts (Bernhard et al., 2012b). Future studies might decipher, if indeed bacteria are responsible for the missing denitrification step and be immune to such antibiotic treatment, if an oxygenic nitric oxide dismutase skips the last denitrification step as discussed by Woehle & Roy et al. (2018) and/or if foraminifera have unknown enzymes that catalyze the missing steps as suggested by Gomaa et al. (2021)."

**Reviewer:** Lines 207-208: "*it is likely that dormancy is a common survival strategy under anoxia for foraminiferal species that don´t have an anaerobic metabolism*". I agree that dormancy is a logical alternative in case of absence of  anaerobic metabolism, like in the various species of the *A. tepida* morphogroup. However, on line 182, as examples of dormancy, the author mentions *Bulimina marginata, Stainforthia fusiformis* and *Adercotryma glomerata.* The first two species have been shown to concentrate intracellular nitrate, and *Stainforthia* sp. has been shown to be able to denitrify. This would mean that we have also indications of dormancy in species which can denitrify. What makes the addition "*that don't have an anaerobic metabolism*" in the line cited above, incorrect, or at least incomplete. The author has to clarify this!

**Reply:** This is a very good point, since this basically provides solid evidence that even denitrifying foraminifera can get dormant under unfavorable conditions. I rewrote this part as follows:

"Considering all these studies, it is likely that dormancy is a common survival strategy for foraminiferal species that either get exhausted of suitable electron acceptors (i.e. O2 or NO3-) or are exposed to periods of extreme environmental conditions. Since there is evidence for dormancy in both S. fusiformis and B. marginata (Bernhard and Alve, 1996), it is likely that even denitrifying species can get dormant under unfavorable conditions. Another Stainforthia sp. has been shown to denitrify and B. marginata stores NO3- in some environments (Piña-Ochoa et al., 2010b)."

**Reviewer:** Lines 225-227: "*Intertidal foraminifera are often exposed to hypoxic or even anoxic conditions, when water stagnates during low tide or if they are transported to deeper anoxic sediment layers by bioturbation* (Rybarczyk et al., 1996; Cesbron et al., 2017)". These lines suggest that the author is not very familiar with intertidal environments. Things are much worse: on most intertidal mudflats, oxygen penetration is less than 1 mm, so that all forams are confronted with anoxia.

**Reply:** I added the following sentence to clarify, that intertidal foraminifera are often exposed to anoxia, even within the first cm of the sediment column: "Oxygen penetration depths in tidal flats can vary between a few mm during low tide to several cm during high tide (Jansen et al., 2009). Thus, intertidal foraminifera are often exposed to anoxia, even within the first cm of the sediment column."

**Reviewer:** Lines 211-231: It should perhaps be useful to explain here that *Haynesina germanica* is very often found together with taxa of the *Ammonia tepida* morphogroup (*Ammonia aberdoveyensis* or *Ammonia confertitesta*), taxa which don't have the ability to keep the ingested chloroplasts active.

**Reply:** The following sentence has been added to this part: "*H. germanica* often shares the habitat with species from the *Ammonia tepida* morphogroup (*Ammonia aberdoveyensis* or *Ammonia confertitesta* according to Hayward et al., 2021) which also tend ingest chloroplasts but these chloroplasts do not show any photosynthetic activity anymore (Jauffrais et al., 2016)."

**Reviewer:** Lines 308-310: "*It is remarkable that there is so much evidence for phagotrophy on or by benthic foraminifera under anaerobic conditions and future studies might shed more light on predator prey interactions of benthic foraminifera in O2 depleted environments*." I think this is very much exaggerated. As far as I can see, evidence for foraminiferal phagotrophy under anaerobic conditions is very scarce. It seems to be limited to (rather indirect) evidence of Orsi et al. (2020).

**Reply:** This chapter will get an extended overhaul according to the review by Andrew Gooday. It will briefly all kinds of trophic strategies of benthic foraminifera, summarize what is known about the forams from O2 depleted environments and show that predation is more an exception, but possible.

**Reviewer:** As a final remark, although I appreciated the review, and found it very useful, I also think it emphases what we (think to) know, but doesn't talk too much about the things we don't know (yet). It would be good if the author could from time to time indicate some unsolved resolved questions, and, if possible, research strategies to answer these questions.

**Reply:** In the revised manuscript I will address some unsolved questions in several parts of the paper. Some of these statements are already mentioned above in my responses to the reviewer's previous points of revision. Here are some examples of such sentences which I added to the revised MS:

"Future perspectives on understanding the biology of intermediate infauna might include studying the transcriptome to decipher other anaerobe metabolic pathways and testing the denitrification capacity after incubation in $NO_3^-$-free and $NO_3^-$-containing seawater."

"Eventually, studies on the transcriptome of non-denitrifying species from infaunal environments might be able to show, if some of these species can switch an alternative anaerobe metabolism under exposure to anoxia."

"Hopefully, measurements of metabolic rates, intracellular nutrient content and enzymatic activity might bring further evidence in the future, if at least some epifaunal species can switch to an anaerobe metabolism, when $O_2$ is too depleted."

"Future studies might decipher, if indeed bacteria are responsible for the missing denitrification step and be immune to such antibiotic treatment, if an oxygenic nitric oxide dismutase skips the last denitrification step as discussed by Woehle & Roy et al. (2018) and/or if foraminifera have unknown enzymes that catalyze the missing steps as suggested by Gomaa et al. (2021)."

"In general, future metabarcoding studies to identify food sources of deep infauna or foraminifera that inhabit anoxia might shed more light on trophic strategies in $O_2$ depleted environments."

---

## Author Comment (AC2)

Dear reviewers, Dear Lisa Levin (Editor),

thank you very much for the feedback to this review paper. I am really glad that three experienced specialists from this field provided such detailed constructive revisions to the manuscript. Of course, a review article will strongly benefit from discussing different opinions and different points of view of several experts. I revised my manuscript thoroughly regarding the feedback to all three reviewers. Below you can find a detailed point by point response to the review by Andy Gooday.

**Reviewer**: This manuscript gives a very useful review of biology of foraminifera that live where oxygen is absent or scarce. I found the section on survival strategies(1.3), together with the data summarized in Tables 1-3 and the figures, very interesting and informative. Section 1.5 on the role of foraminifera in nutrient cycling and biogeochemistry is also useful (although see the comment below). However, sections 1.4 and 1.6 need some thought. Section 1.4 is headed 'ecology' but that's misleading. It's actually a very brief and rather confusing treatment of diets/feeding. The final section about palaeoceanography is rather superficial and somewhat redundant, since a more detailed review is on the way.

**Reply:** Thanks a lot for the detailed, constructive feedback to this manuscript. The paleoceanography section has been deleted and condensed to a short paragraph in the introduction according to the feedback of another reviewer (see response to Frans Jorissen). The "ecology" part has been carefully rewritten according to your suggestions. Below you can find a detailed point by point response to all points of revision.

**Reviewer**: **Specific comments**

**Reviewer**: line 79, Table 2. Since the paper is about foraminifera, you should mention somewhere in the text that gromiids are not foraminifera but a separate group of protists within the Rhizaria.

**Reply:** This is a good point and has also been addressed by another reviewer. I adapted the part in the text, where gromiids are mentioned for the first time accordingly:

"Other studies showed that bacterial endobionts likely perform denitrification in some allogromiid foraminifera and gromiid species (Bernhard et al., 2012a; Høgslund et al., 2017). Gromiida are a separate group of protists within the Rhizaria and closely related to foraminifera."

**Reviewer**:

1. 8. Section 1.4 Ecology.

This is a brief and selective section about feeding, not a review of foraminiferal ecology. I would change the title to 'Trophic interactions in oxygen-depleted environments' or something similar. At the beginning, you might want to cite a few general reviews of feeding and diets in foraminifera, such as those of Lipps (1973, cited elsewhere) and Gooday et al. (1993, 2008). This would provide some context for feeding and diets in general. I have several other comments.

**Reply:** This is a very helpful and constructive feedback. I rewrote the whole section and took the new title as suggested. This section now synergizes much better with the rest of the manuscript, since the different feeding strategies can also be used to discuss the microhabitat preferences that have been discussed in the section about survival strategies. The text of the reworked section can be found below:

[revised manuscript text omitted]

**Reviewer**: 1) The Sagami Bay habitat is not strongly hypoxic, at least in terms of bottom water, and the observations of Nomaki probably apply to more oxygenated settings as well, particularly for the shallow infaunal species (*Uvigerina akitaensis* and *Bolivina spissa*).

**Reply:** This is a valid point and I tried to clarify this in this part of the review by adding the following sentence:

"Since the $O_2$ concentrations at central Sagami Bay are not extremely low, these observations likely apply to more oxygenated environments as well, especially for the shallow infaunal species."

**Reviewer:** 2) In lines 325-326, you say that predation ('phagotrophy') is the main type of interaction in aerobic communities. Is phagotrophy synonymous with predation? I thought it applied to any ingested particle, not necessarily prey?

**Reply:** Of course, the reviewer is totally right that phagotrophy is not synonymous with predation. This sentence was formulated in a very misleading way. I deleted the "phagotrophy" from the brackets in this sentence. In addition, to avoid further misunderstanding, I adapted the end of the paragraph accordingly:

"There is further evidence by Orsi et al. (2020) that foraminifera from the Namibian shelf can perform phagocytosis (vacuolic ingestion of food particles) even under anoxic conditions. They provide further evidence that the Namibian foraminifera express enzymes for lysing digested prey cells inside food vacuoles after phagocytosis (schematic representations for phagocytosis and predation on meiofauna shown in fig. 6)."

**Reviewer:** 3) The same sentence seems to imply that foraminifera are mainly predators, which to me gives the impression that they are carnivores (as discussed in lines 319-324). I don't believe this is correct. It's true that there is increasing evidence that some foraminifera eat metazoans or other foraminifera, at least sometimes (e.g., review of Culver and Buzas, 2003), but in general, most forams seem to feed most of the time at a low trophic level on algae such as diatoms, bacteria, and decaying organic material, and form a link to higher trophic levels (Gooday et al., 1993; Nomaki et al., 2008). By predation you seem to mean consumption of mainly algae, bacteria, and sediment/degraded material (as described by

Nomaki et al., 2005), which I would call herbivory, bacterivery and deposit feeding. Also, lines 325-333 are apparently about benthic communities generally, not specifically about forams, but the meaning is not clear and may give the wrong impression. I found this whole paragraph rather confusing.

**Reply:** As stated above, the whole section has been completely reworked. It should be clear now, that foraminifera aren´t mainly predators but that some species can show predatory behavior (text changes see above).

**Reviewer:** 4) What about dissolved organic matter (DOM)? This may also be important, including in Sagami Bay (Nomaki et al., 2011).

**Reply:** The possible utilization of DOM is now discussed in the revised MS:

"Another feeding experiment at Sagami Bay by Nomaki et al. (2011) revealed that all of the analyzed benthic species assimilated carbon from $^{13}C$ labeled glucose and thus can effectively utilize also dissolved organic carbon."

**Reviewer:** Culver S J, Buzas MA (2003). Predation on and by foraminifera. In Predator-Prey Interactions in the Fossil Record, ed. Patricia H. Kelley, Michael Kowalewski & Thor A. Hansen Kluwer Academic/Plenum Publishers, New York; pp. 7-32.

Gooday, AJ, Levin LA, Linke P, Heeger T. (1992b) The role of benthic foraminifera in deep-sea food webs and carbon cycling. In: Rowe GT and Pariente V (eds) Deep-Sea Food Chains and the Global Carbon Cycle. Kluwer Academic Publishers, p. 63-91.

Gooday, A.J., Nomaki, H. & H. Kitazato. 2008. Modern deep-sea benthic foraminifera: a brief review of their biodiversity and trophic diversity. In: Austin, W. E. N. & James, R. H. (eds) Biogeochemical Controls on Palaeoceanographic Environmental Proxies. Geological Society, London, Special Publications, 303, 97–119. DOI: 10.1144/SP303.8 0305-8719/08/

Nomaki, H., Ogawa, N.O., Ohkouchi, N., Suga, H., Toyofuku, T., Shimanaga, M., Nakatsuka, T., Kitazato, H., 2008. Benthic foraminifera as trophic links between phytodetritus and benthic metazoans: carbon and nitrogen isotopic evidence. Mar. Ecol. Prog. Ser. 357: 153–164, 2008

Nomaki, H., Ogawa, N.O., Takano, Y., Suga, H., Ohkouchi, N., Kitazato, H., 2011. Differing utilization of glucose and algal particulate matter by deep-sea benthic organisms of Sagami Bay, Japan. Mar. Ecol. Prog. Ser. 431, 11–24.

**Reviewer:** Section 1.5 Lines 376-379. Are the pristine shells live and those filled with phosphorite dead? Or are they dead in both cases? If the shells are dead, it might suggest that the phosphorite was deposited by some inorganic process. I thought that modern phosphorites in anoxic sediments were precipitated from interstitial porewater (e.g., Kolodny, 1981, Phosphorites. In: The Sea v. 7, The Oceanic Lithosphere). Please explain further or delete.

**Reply:** This section might indeed have been a bit too short and some important information was missing. There have been several studies that showed that the large intracellular polyphosphate enrichments in sulfur bacteria can facilitate phosphorite formation at the upper boundary of the Peruvian OMZ that is rich in bacterial mats and phosphorite deposits. The phosphorite deposits at the lower boundary of the Peruvian OMZ are different, because these

bacterial mats are usually not present and the phosphorite grains have a similar size and shape of foraminifera. These phosphorite grains are also abundant in the surface fraction of the sediments. In the Peruvian OMZ, living foraminifera abundances are very high. The sediments are thus a mixture of living forams, phosphorite grains with a coarse shape and size of a foram and every intermediate step in between. It is likely that their high intracellular phosphate storage, together with the calcium storage to precipitate their tests results in a supersaturated apatite microenvironment within their shells and initiates apatite formation. This has been also suggested for other organisms before. All this is discussed in detail in my 2020 GCA paper ("A hidden sedimentary phosphate pool inside benthic foraminifera from the Peruvian upwelling region might nucleate phosphogenesis"). To address these issues and avoid further misunderstandings, I extended the text in this section:

"In addition, there is evidence that the intracellular phosphate storage in foraminifera facilitates phosphogenesis in some environments, similar to the intracellular polyphosphate enrichments in some sulfur bacteria (Schulz and Schulz, 2005). The release of phosphate after breakdown of these polyphosphates to harvest energy in times of electron acceptor depletion results in apatite supersaturation and initiates phosphogenesis (Schulz and Schulz, 2005). Sediments at the lower boundary of the Peruvian OMZ contain many small phosphorite grains with similar size and shape of foraminifera (Manheim et al., 1975; Glock et al., 2020). The sand fraction of the surface sediments in this region is a mixture of pristine living foraminifer shells with dead tests that show a transition from shells that are filled with phosphorites until small phosphorite grains that only retain the size and coarse shape of a foraminifer. It is likely that a *post mortem* release of the intracellular phosphate storage results in a supersaturated microenvironment within the shells that initiates apatite formation (Glock et al., 2020) in a similar way as it has been suggested for other organisms (Kulakovskaya, 2014)."

**Reviewer:** Subsection 1.5.1, p. 9-10. You finish this subsection rather abruptly with equations for estimating cell biovolume, denitrification rates, and P content. You precede these equations by presenting a regression between intracellular NO3- and cell volume (Fig. 7). It would make more sense to present Fig. 7 after you have outlined the equations, since the regression depends on equation 1. This would give the subsection a more logical structure.

**Reply:** I understand that this subsection ends rather abruptly, especially since the following section is cut out in the revised version of the paper (see next comment below). Nevertheless, I thoroughly read through this section again and I think it is important to present figure 7 before the equations or at least before equation 1, which might be moved further down in the paper. Equation 1 is a result from the power regression that is shown in figure 7. Equations 2 & 3 are equations that are independent from this regression and have been published before in the cited papers. Since the reviewer is totally right with the abrupt ending, I added two more general sentences to the end, that adjust the whole section into a bit of a bigger context:

"With an increasing amount of data about metabolic rates and intracellular nutrient storage more accurate models and equations might become available in the future that describe the role of benthic foraminfera within marine biogeochemistry. Similar models and equations might be also very helpful for exploring the role of planktonic foraminifera in pelagic biogeochemistry."

**Reviewer:** Section 1.6.3. Obviously, this is a very extensive topic that a short paragraph cannot do justice to. If you want to keep it, then there are reviews that you could cite (e.g., Gooday 2003, Jorissen et al. 2007). The OMZ review of Levin (2003), which deals with benthos as whole but includes foraminifera, is also relevant. Please note that it's not just the

taxonomic composition that reflects oxygen concentrations. Other assemblage attributes, such as diversity and dominance, are also strongly influenced. However, I would agree that the problem of disentangling the relative effects of organic matter and oxygen on the composition, diversity, dominance etc of foraminiferal assemblages in hypoxic settings is an important point to make.

**Reply:** According to the suggestions of another reviewer I completely cut the part about applications in paleoceanography in this paper and only provide a brief summary of all three subsections in the introduction of the paper. As mentioned in the original draft by my paper there is a larger community review paper in progress that focuses on the paleo-applications (Hoogakker et al., *in prep*). This paper in preparation will also provide more details regarding the ecology and paleoceanographic application of foraminiferal assemblages from $O_2$ depleted environments.

Gooday, A.J. (2003). Benthic foraminifera (Protista) as tools in deep-water palaeoceanography: a review of environmental influences on faunal characteristics. Advances in Marine Biology, **46,** 1-90.

Levin, L.A. (2003) Oxygen minimum zone benthos: adaptation and community response to hypoxia. Oceanography and Marine Biology: An Annual Review 41: 1-45

Jorissen, F., Fontanier, C., and Thomas, E. 2007. Paleoceanographical proxies based on deep-sea benthic foraminiferal assemblage characteristics. in: Hillaire-Marcel, C., and de Vernal, A.: Proxies in Late Cenozoic Paleoceanography, Elsevier, Amsterdam, Boston, Heidelberg, London, New York, Oxford, Paris, San Diego, San Francisco, Singapore, Sydney, Tokyo, 263-325.

**Editing suggestions**

Some parts of the text are well written, but others are not and need careful editing, for which suggestions are made below.

Abstract and Introduction

**Reviewer:** Line 12. Delete 'ongoing'

**Reply:** Done.

**Reviewer:**

1. 'Since several species….' (no need to repeat foraminifera)

**Reply:** Done.

**Reviewer:** 23, 24. Delete the repetitions of 'even'.

**Reply:** Done.

**Reviewer:** 24-25. 'Finally, since foraminifera can calcify under anaerobic conditions, I will briefly review proxies for O2 based on their shell composition and assemblage composition.

**Reply:** The structure of the abstract has slightly been changed according to the changes in the manuscript.

**Reviewer:**

1. 'More than a decade later…..published…'

**Reply:** Done.

**Reviewer:**

2. 'Nevertheless, advances in methods to analyse the metabolic rates, intracellular nitrate storage and molecular genetics of foraminifera has changed our understanding of strategies such as anaerobic metabolism that help them to withstand O2 depletion.'

**Reply:** Done.

**Reviewer:**

3. Are foraminifera really 'microeukaryotes? This term would be more appropriate for eukaryotic microbes such as flagellates. Better to just call them 'small eukaryotes' or 'meiofaunal eukaryotes' (although not all are meiofaunal or particularly small).

**Reply:** "Microeukaryotes" has been deleted.

**Reviewer:**

4. '….2021). As a result, G. hexagonus has…'

**Reply:** Done.

**Reviewer:**

43-44. 'Benthic foraminifera ….have also been established ..'

**Reply:** Done.

**Reviewer:**

1. However, I will touch on …..benthic foraminifer only briefly, since…'

**Reply:** Done.

**Reviewer:**

48-52. 'The first part of this paper reviews recent advances in our understanding of the diverse strategies that foraminifera use to withstand O2 depletion, focussing mainly on denitrification, dormancy and kleptoplasty. I also incorporate denitrification into the

conceptual TROX model of Jorissen et al. (1995), which explains the sediment microhabitats of benthic foraminifera in terms of an interplay…..'

**Reply:** Done.

**Reviewer:**

1. Next, I briefly discuss….'
2. 'foraminifera in marine…'
3. You've already mentioned foraminifera as oxygen proxies in paleoceanography in lines 44-47

**Reply:** All done.

**Reviewer:**

Section 1.3

60-65. You've already explained some of this paragraph in the Introduction. I think you can limit it to the first sentence (but please change 'examples see' to 'see examples').

**Reply:** Done.

**Reviewer:**

1. 'first evidence emerged…'
2. 'can store substantial amounts of…'
3. 'of the presence..'

**Reply:** All done.

**Reviewer:**

4. 'storage capacity'

**Reply:** I am not completely sure, which part of this paragraph is meant, because the nitrate storage is mentioned several times.

**Reviewer:**

108-109. 'This indicates that foraminifera use other enzymes to catalyze these steps, or they rely on…., or…..' (you can only use 'either…or' for two alternatives, not three)

**Reply:** Done.

**Reviewer:**

1. Woehle & Roy et al, 2022. Is this the same as Woehle et al 2022?

**Reply:** Yes, it is. Both papers Woehle & Roy et al. 2018 & 2022 were papers with doubled first authorships. I wanted to do the second first author justice, because otherwise, mainly the

first name will be shown as citation in other papers. Unfortunately, this was not uniform along my manuscript, due to the use of a reference manager. I corrected this and now use "Woehle & Roy" all along the manuscript.

**Reviewer:**

2. Replace second 'likely' with 'possibly' or 'probably'.
3. Better to delete 'Furthermore' and start sentence with 'The'.

**Reply:** All done.

**Reviewer:**

149, 154. 'epifaunal' (not 'Epifaunal' – you are not starting a new sentence).

168 'its shallow infaunal…' (not 'their')

1. 'certain circumstances'

**Reply:** All done.

**Reviewer:**

182-186. These last two sentences can be simplified – 'Research to measure denitrification rates in different benthic foraminiferal species continues. This will add to the scarce available data and contribute to estimates of the role of foraminifera in …….'

**Reply:** Done.

**Reviewer:**

195-196. 'some studies suggested that some foraminifera may become dormant when …'

1. 'putative anoxic habitats
2. 'had' (not 'has')
3. Delete 'own'
4. 'fended'

**Reply:** All done.

**Reviewer:**

247-248. Insert commas after brackets.'

1. No comma after 'test'
2. Delete 'rest'.

**Reply:** All done.

**Reviewer:**

277-281. You could condense these two sentences as follows – 'Several recent publications based on advances in molecular biological methods……have revealed some other metabolic adaptations of foraminifera that thrive under…..'

281-283. Again, this could be simplified -  'In N. stella and B. argentea, Gomaa et al. (2021) found evidence for the expression of proteins, including pyruvate……hydrogenase, that are characteristic of anaerobic metabolism. The PFOR sequences…..'

   1.   'already came from a study by Nardelli…'

**Reply:** All done.

**Reviewer:**

297-299. This sentence more or less repeats the previous one in lines 295-296. I suggest you run together the next two sentences to say – 'These processes (calcification and the ingestion of prey cells by phagocytosis) require bursts of high energy, which the authors suggest is generated by dephosphorylation of….to generate ATP.'

   1.   'metatranscriptonomes' (spelling)

303-304. 'might serve as a ….'

**Reply:** All done.

**Reviewer:**

305-307. 'Orsi also found evidence for an anaerobic metabolism….'. I may not be understanding correctly, but isn't anaerobic metabolism what you describe on line 300? If so, perhaps you should run these two sentences together – 'Orsi et al. (2021) also found evidence that foraminifera on the Namibian shelf metabolize hydrolyzed…..'

**Reply:** Not really. The previous part describes the dephosphorylation of creatine phosphate, which is more like a "lifeline" for the forams, when conditions are really bad. They use it as a reservoir to generate ATP, when electron acceptors are depleted. The anaerobic metabolism, described at the end of the paragraph describes the "typical" anaerobic metabolism that is based on fermentation and is happening all the time and not only, when the conditions are "bad".

**Reviewer:**

   1.   'conducted' (not 'made')

**Reply:** Done.

Section 1.4.

Please see comments above.

**Reviewer:**

Section 1.5

339-342. These two opening sentences more or less repeat lines 74-76. I would start this section instead with the third sentence ('Pina-Ochoa et al. (2020b) pointed out the possible importance of….').

1. 'due partly to their high abundances…'
2. 'Globobuliminids' is not a genus. If you want to use italics, then this should be 'Some species of *Globobulimina…*'
3. Add comma after 'foraminifera'.

**Reply:** All done.

**Reviewer:**

363-365. These equations are the subject of subsection 1.5.1, so perhaps they should not be mentioned here. Also, .'….to calculate estimated denitrification rates,…'

1. 'data' (not 'Data').

**Reply:** All done.

**Reviewer:**

372-373. 'Hypotheses include…' (Delete 'about the use of the stored phosphate' – you've said this already)

1. 'contain' (not 'bear').

**Reply:** All done.

**Reviewer:**

389-390. '…are typically shallow infaunal and …'

394-395. This could be reworded for clarity, e.g. – 'Given this variation in NO3- storage capability, the reliability of estimates for the foraminiferal contribution to NO3- budgets depends crucially on the availability of data. The more data there is, the better …..'

1. '…including the contribution of species with….'

404, 412. Maybe remove the comma at the beginning of the line. It looks odd.

**Reply:** All done.

**Reviewer:**

Section 1.6

1. 'This section (not chapter) briefly discusses morphological and geochemical aspects of benthic foraminiferal tests, and foraminiferal assemblage attributes, that can be used……. A more extensive review will summarize…'
2. 'that live in deep infaunal microhabitats below….' Also 'spp., for example, are….' (spp. is not in italics).

434-5. 'Though, Globobuliminidae…..' Delete this sentence. You already said this a few lines earlier. Also please note that although species of *Globobulimina* (and *Praeglobobulimina*) are large and are not flattened, none of them are actually spherical. I would say that members of this genus tend to have an ovate or globular shape, sometimes with a pointed proximal end.

**Reply:** According to the remarks by another reviewer (Frans Jorissen) the whole section about paleoceanographic applications has been deleted and there is only a brief summary of this section left in the introduction (see response to review by Frans Jorissen).

**Reviewer:**

1. 'is the porosity, including pore size and pore density, of foraminiferal tests.'
2. '…characteristics began in the 1950s…'
3. '…the first attempts were made to use the test porosity of PLANKTONIC foraminifera…'
4. 'Here, I focus on the…'

451-452. '…environments with ………and fossil specimens from periods of..'

1. Delete comma after 'known'
2. 'were identified as potential…'

481-482. 'This offset depends on the vertical separation of the species within the sediment column…'

**Reply:** According to the remarks by another reviewer (Frans Jorissen) the whole section about paleoceanographic applications has been deleted and there is only a brief summary of this section left in the introduction (see response to review by Frans Jorissen).

**Reviewer:** Table 2 caption. Foraminifera and gromiids (or *Gromia,* but not 'gromiida') are mentioned separately, which is correct, but in the table itself it might be clearer to put them in a separate section, just to be clear.

**Reply:** Done.

**Reviewer:**

Table 3 caption line 1. 'foraminifera'

Fig. 4 caption line 5. Delete 'supply'.

Fig. 4 caption line 8. '..when they have to, although if the food supply is too low, they …'

**Reply:** All done.

**Reviewer:**

Fig. 6 caption, bottom line. 'Foraminifera in hypoxic'

**Reply:** I am not really sure, which part you mean because this phrase does not really fit into the bottom line of the Fig. 6 caption.

---

## Author Comment (AC3)

Dear reviewers, Dear Lisa Levin (Editor),

thank you very much for the feedback to this review paper. I am really glad that three experienced specialists from this field provided such detailed constructive revisions to the manuscript. Of course, a review article will strongly benefit from discussing different opinions and different points of view of several experts. I revised my manuscript thoroughly regarding the feedback to all three reviewers. Below you can find a detailed point by point response to the review by Anonymous Reviewer#2.

**Reviewer:** The manuscript by Glock is a good initial attempt to review current knowledge about benthic foraminifera inhabiting anoxic AND oxygen-depleted (hypoxic, dysoxic) habitats. In fact, roughly half of the situations described regard foraminifera living in low – perhaps even moderately high— oxygen conditions. Further, manuscript portions discuss gromiids / *Gromia*, which are not foraminifera. With these two things in mind, the present title must be changed in terms of "anaerobic" and "foraminifera". In particular, the title must include low oxygen (or synonym) and "gromiids" or the contribution must be stricken of all discourse about low-oxygen habitats and the gromiids. Further, the use of plural ("Reviews and syntheses") is grammatically incorrect. In my opinion, the title should also include "benthic" as there are only a few sentences about planktic foraminifers. The abstract too requires edits after considering the detailed comments that follow.

**Reply:** Regarding the plural in "Reviews and syntheses": A similar recommendation has been given by another reviewer (Frans Jorissen). I agree with these statements and did not have "Reviews and syntheses" in my original title. After my initial submission but before the upload of the preprint I had the request by the editorial office to change my title accordingly. "Reviews and syntheses" is mandatory in the title of review papers that are submitted to this journal.

For the rest: I added "Benthic foraminifera" to the title and changed "anaerobic" to "oxygen depleted" to be less restricting. I am not opposed to add "gromiids" as well. I also clarified now specifically in the text that gromiids aren´t foraminifera, when gromiids are first mentioned in the text:

"Other studies showed that bacterial endobionts likely perform denitrification in some allogromiid foraminifera and gromiid species (Bernhard et al., 2012a; Høgslund et al., 2017). Gromiida are a separate group of protists within the Rhizaria and closely related to foraminifera."

In addition, I separated foraminifera and gromiids in table 2.

I would change the title of the paper as follows, which also addresses the points of revision by the other two reviewers:

"Reviews and syntheses: Benthic foraminifera and gromiids from oxygen depleted environments - Survival strategies, biogeochemistry and trophic interactions"

**Reviewer:** It is not clear why certain topics received much attention while other "survival strategies" for inhabiting oxygen-depleted habitats were effectively ignored (symbionts, for example).

**Reply:** I discussed symbiosis in several parts of the "survival strategies chapter" (examples see below). It was certainly not my intention to ignore this point. Due to the statement by the reviewer, I assume that this discussion was not comprehensive and to address this issue, I added the following parts to the revised manuscript:

"There is strong evidence for symbiosis between foraminifera and prokaryotes in many hosts from $O_2$ depleted environments, which most likely are an adaptation to survive within the steep geochemical gradients close to the oxic/anoxic boundary (Bernhard et al., 2000; Bernhard, 2003; Bernhard et al.,

2006; Nomaki et al., 2014; Bernhard et al., 2018). Most of the observed prokaryotic associates are endobionts within the foraminiferal cytoplasm but some are ectobionts that often are observed close to the pores in the foraminiferal shell (Bernhard et al., 2001, 2010a, 2018). For about a decade after the first discovery of foraminiferal denitrification it remained unclear if foraminifera indeed denitrify themselves, or if the bacterial symbionts are responsible for the denitrification. Evidence came up for both hypotheses."

"Recent metagenomics and transcriptomics results of denitrifying foraminifera indicate that bacterial symbionts might perform the missing steps in the foraminiferal denitrification pathway or that they at least partly contribute to the amount of $NO_3^-$ that is denitrified within foraminiferal cells (Woehle & Roy et al., 2022). It has already been hypothesized before that the ectobionts, found on *Bolivina pacifica* from the Santa Barbara Basin are either sulfate reducing or sulfur oxidizing bacteria (Bernhard et al., 2010a)."

"A continuum of intracellular bacteria including prey in food vacuoles, endobionts, parasites and necrophages has been documented before in benthic foraminifera from cold seeps (Bernhard et al., 2010b). It already has been hypothesized by the authors that bacteria switched their function from endobionts to predators, depending on the vitality of the host cell."

In addition, two other studies that discuss alternative adaptations of foraminfera to O2 depletion (Bernhard & Bowser, 2008 & Powers et al., 2022) are now discussed in a bit more detail in the manuscript:

"Most foraminifera species from $O_2$ depleted habitats possess numerous peroxisomes that are usually associated with mitochondria and the endoplasmatic reticulum (Bernhard and Bowser, 2008). Bernhard and Bowser (2008) hypothesized that these peroxisome proliferations might be used to either metabolize $H_2O_2$ and other highly reactive oxygen species that are produced within the chemocline close to the oxic/anoxic boundary or to reduce the oxidative stress by these compounds. Indeed, they showed in an experiment that ATP concentrations in foraminifera increased proportional to ambient $H_2O_2$ concentrations. A recent study on transcriptome and metatranscriptome of *N. stella* and *B. argentaea* from the Santa Barbara Basin revealed that these species utilize an adaptable mitochondrial and peroxisomal metabolism, depending on the chemical treatment in the experiment (Powers et al., 2022). The high plasticity of their peroxisomal and mitochondrial metabolism might be substantial for survival at the highly variable conditions at the chemocline in the sediments. The results by Powers et al. (2022) indicate that at least some processes that are involved in foraminiferal denitrification are associated with mitochondria. Interestingly, the expression of denitrification related genes in both species was upregulated after incubation with elevated $H_2O_2$ but without $NO_3^-$ and downregulated, if they were incubated without $H_2O_2$ but with $NO_3^-$, compared to a control treatment with both $H_2O_2$ and $NO_3^-$. In the same way several peroxisomal processes were upregulated in the $H_2O_2$ only treatment. In addition, despite that both species are able to denitrify, Powers et al. (2022) found distinct metabolic adaptations to anoxia in both species. For example, a quinol:fumarate oxidoreductase, which is considered as an adaptive mechanism for anaerobic respiration in eukaryotic organisms, was present in *N. stella* but not in *B. argenaea*. Vice versa, *B. argentaea* has the capacity to digest food vacuole contents under $O_2$ depletion, while *N. stella* was lacking food vacuoles (Powers et al., 2022)."

As I mentioned above, I did not ignore symbiosis in the original manuscript and it was not my intention to ignore any of the present studies. Here are examples, where I already discussed symbiosis in the original manuscript:

Line 90-95: "For about a decade after the first discovery of foraminiferal denitrification it remained unclear if foraminifera indeed denitrify themselves, or if they host **bacterial symbionts** that are responsible for the denitrification Bernhard et al. (2012a) showed that *Bolvina argentaea* was able to denitrify, even after a very harsh treatment with antibiotics, which indicates that this species can denitrify even, when potential **bacterial symbionts** are killed. Other studies showed that **bacterial endobionts** likely perform denitrification in some allogromiid species and gromiids (Bernhard et al., 2012a; Høgslund et al., 2017)."

Line 100-104: "Nevertheless, the homologues of the enzymes that catalyze the first and the last step of foraminiferal denitrification (Reduction of $NO_3^-$ to nitrite ($NO_2^-$) and reduction of nitrous oxide ($N_2O$) to $N_2$ gas; fig. 3) have not been identified, yet. This indicates either that foraminifera use other enzymes to catalyze these steps, that they rely on **bacterial symbionts** for these steps or that they use an alternative denitrification pathway in general."

Line 110-112: "Recent metagenomics and transcriptomics results of denitrifying foraminifera indicate that **bacterial symbionts** might perform the missing steps in the foraminiferal denitrification pathway or that they at least partly contribute to the amount of $NO_3^-$ that is denitrified within foraminiferal cells (Woehle et al., 2022)."

**Reviewer:** In reality, there are few studies that have shown bonafide "complete denitrification" as Risgaard-Petersen et al. did in their 2006 paper. To date, the 'omics have shown the process lacks the final step, so those studies technically have not shown "complete denitrification". This point should be made clear and elaborated upon as necessary throughout the contribution.

**Reply:** This is a valid point. The incomplete foraminiferal denitrification pathway is discussed later in the paper but it might be confusing, if this is not clearly stated already at the beginning. I adapted the part where I introduce Risgaard-Petersen's results as follows:

"The discovery by Risgaard-Petersen et al. (2006) was also the first evidence for complete denitrification in eukaryotic cells in general and it also showed that they likely take up $NO_3^-$ from the surrounding pore water and store it within intracellular seawater vacuoles. Nevertheless, no later study could actually proof a bonafide "complete" denitrification pathway in foraminifera and the eukaryotic foraminiferal denitrification pathway is today considered to be incomplete (Woehle et al., 2018; Orsi et al, 2020; Gomaa et al., 2021; see discussion below). Other eukaryotes that are known to perform incomplete denitrification are the primitive eukaryote *Loxodes* (Finlay et al., 1983) and two species of fungi (Usuda et al., 1995)."

**Reviewer:** The contribution should include a short synopsis where definitions of the terms used in the manuscript are defined. For instance, the term "anaerobic" typically refers to a physiological or metabolic process, not a habitat—meaning, "anaerobic environments" is unconventional phraseology. As the author likely knows, there are a plethora of terms designating environments with low concentrations of oxygen; these must be defined as the author interprets. For example, when one "low oxygen" concentration was presented, it was really quite high (60 uM), so it really is necessary that details be included.

**Reply:** The use of "anaerobic" might indeed be misleading as there are different definitions for this term and "anaerobic environments" are often better referred as "hypoxia". Nevertheless, also the "definitions" of "hypoxic" or "suboxic" conditions often show a broad $O_2$ concentration range that

differs between various literature sources. Therefore I will avoid these terms throughout the text of the revised manuscript to avoid further confusion or misunderstandings. Every time when the paper uses the term "anaerobic" it now refers to a metabolic process. "Anaerobic environments" are now termed either O2 depleted environments, or in some cases, when O2 is absent in these environments as "anoxic environments" or "anoxia". "Anoxia" and "anoxic" are now defined in the introduction within the following sentence:

"The present paper will sometimes along the text refer to "anoxia" or "anoxic conditions", which are now defined as the absence of oxygen."

**Reviewer:** There are a number of overgeneralizations that require literature support or the assertions much be curtailed. For example, lines 13-14 state that benthic foraminifera as a group can benefit from ocean deoxygenation. This is vastly overstated. While some species may compete well in very low oxygen to anoxia, there is little, if any, evidence to suggest the entire group will benefit from deoxygenation. A second example involves the genus Bolivina, which is considered a model "deep infaunal" taxon in Fig. 4 yet there is no universally accepted agreement that all Bolivina are deep infaunal / anaerobes. Lines 149-150 notes four genera that are designated "deep infaunal facultative anaerobes" yet all species of each of these genera have not been assessed in this context so this assertion is premature.

**Reply:** The parts addressed by the reviewer have been adapted. The part in the abstract (originally line 13-14) has been rewritten to make clear that not all benthic foraminifera will benefit from ocean deoxygenation:

"Benthic foraminifera are a group of protists that include taxa with adaptations to partly extreme environmental conditions. Several species possess adaptations to $O_2$ depletion that are rare amongst eukaryotes and these species might benefit from ongoing ocean deoxygenation."

The part where I discuss the deep infauna states that some *Bolivina* spp. are examples for deep infauna. This is not wrong, since *Bolivina seminuda* for example can certainly be considered deep infaunal. *Bolivina spissa* is certainly a shallow infaunal species due to the fact, that it selectively feeds on fresh phytodetritus. Figure 4 used schematic representations of *Bolivina* individuals as an example for deep infauna. This might indeed be misleading, because certainly not all *Bolivina* species are deep infaunal. In the revised manuscript, I removed the *Bolivina* drawings from the figure, although I have to say that the figure was visually more appealing before. In addition, I adapted the last line about the deep infauna. Now the text does not provide genera as examples for deep infauna but species names and refers that these "might be considered as facultative aerobes":

"Taxa belonging to the deep infaunal group that might be considered as facultative aerobes that prefer $NO_3^-$ over $O_2$ include for example *Valvulineria inflata* and *bradyana*, *Bolivina seminuda*, *Globobulimina pyrula* and *Cancris carmenensis* (e.g. Jorissen et al., 1995; Mojtahid et al., 2010; Glock et al., 2019c)."

**Reviewer:** Other statements are incorrect. For example, lines 66-68 note that foraminifera are the only eukaryote to perform denitrification. This is not true—a number of fungi also perform denitrification, and this was first described in the 1990s (Shoun et al. 1992 FEMS Microbiology Letters).

**Reply:** This is likely a misunderstanding. Of course, some fungi and for example also the primitive eukaryote Loxodes can perform incomplete denitrification. The part in 66-68 stated that "the discovery by Risgaard-Petersen et al. (2006) was also the first evidence for **complete** denitrification in eukaryotic cells in general...". And up to date it is the only evidence for complete denitrification in

eukaryotic cells. Nevertheless, the reviewer is right that this study is also the only study that showed bonafide complete denitrification in foraminifera, yet. So this part has been adapted, according to my response of the earlier comment above and I also refer to the other eukaryotes that perform (incomplete) denitrification.

**Reviewer:** Also, the passage spanning lines 218 to 219 is simply wrong: Grzymski et al. (2003) – not Pillet et al., 2011— was the first to sequence foraminiferal kleptoplasts, documenting via molecular methods they were from diatoms. Further, the studies documenting kleptoplast morphology via TEM should not be discounted or belittled as diatom chloroplasts have distinctive morphologies. Molecular methods are not required to establish all facts.

**Reply:** My apologies for this unfortunate wording. It was neither my intention to discount or belittle morphological approaches, nor to discredit the author who sequenced the foraminiferal kleptoplasts before. The two sentences have been changed accordingly:

"The kleptoplasts in foraminifera orginate from diatoms, which has been confirmed on the basis of the chloroplast shape in TEM-observations and by sequencing the chloroplasts with molecular biological methods (Lopez, 1979; Lee et al., 1988; Cedhagen, 1991; Lee and Anderson, 1991; Bernhard and Bowser, 1999; Grzymski et al., 2002; Goldstein et al., 2004)."

**Reviewer:** The fact that there is strong natural variability in nitrate storage causes the estimations of rates and contributions (lines 364) to be merely statistical manipulations that may mean little. This should be elaborated upon.

**Reply:** I disagree with the reviewer in this point. Yes, it is true that the nitrate storage and the metabolic rates of foraminifera show a wide range. This range is covered within the errors of the functions that are used to upscale to total budgets. These can be calculated, using classic statistical methods, such as propagation of uncertainty. I even stated this in the original text:

"Due to the high uncertainties related to the natural variability in metabolic rates and nutrient storage, a thorough error estimation is recommended (see Appendix B in Glock et al. 2020)."

Of course, we need more data to provide better estimates. I state this in several parts of the manuscript. For example:

"Given this variation in $NO_3^-$ storage capability, the reliability of estimates for the foraminiferal contribution to $NO_3^-$ budgets depends crucially on the availability of data. The more data there is, the better we are able to calculate foraminiferal $NO_3^-$ budgets including the contribution of species with unknown denitrification rates or intracellular $NO_3^-$. "

If the reviewer states that these are "merely statistical manipulations", we need to bury a whole scientific discipline whose main tool is earth system modeling. For example, all "bottom-up" models that scale up different sources and sinks for different greenhouse gases, would be redundant! Models, that use knowledge about plankton ecology and chlorophyll distributions in the ocean to estimate global marine primary productivity: Redundant, since all kinds of phytoplankton species have different metabolic rates. Obviously, these models are often based on better studied groups of organisms. That´s why I cannot state enough that we need more "real" data to lower the uncertainty for estimating to total budgets. There are already biogeochemical models that successfully include active biological nitrate transport by foraminifera and sulfur bacteria and denitrification by foraminifera (e.g. Dale et al., 2016) and these will likely improve in the future.

**Reviewer:** Referring to lines 334-336, the author needs clarify why it is sensible to: (1) estimate the denitrification rate for species that are NOT documented to denitrify and (2) why it is sensible to think that the volume of a foraminiferal test reflects denitrification rate (think of the LBFs [Large Benthic Foraminifera] like Amphistegina—by this argument, it would have an extremely high denitrification rate, yet these species live on coral reefs which are not known for anoxic bottom waters).

**Reply:** It is obviously not my intention that researchers start to calculate a hypothetical nitrate storage for species that obviously do not denitrify, such as *Amphestigina* species that live on coral reefs. To clarify, why it is important to estimate these budgets also for the unknown species and that denitrification rates and nitrate storage should not be estimated for candidates that are unlikely to denitrify, I added the following part to the text:

"Given this variation in $NO_3^-$ storage capability, the reliability of estimates for the foraminiferal contribution to $NO_3^-$ budgets depends crucially on the availability of data. The more data there is, the better we are able to calculate foraminiferal $NO_3^-$ budgets. Nevertheless, there are thousands of benthic foraminiferal species and a considerable amount of these species inhabit $O_2$ depleted environments and potentially store $NO_3^-$ and denitrify. It will be unrealistic to measure the intracellular nutrient content and metabolic rates for all foraminifera. Thus, functions to estimate the contribution of species with unknown denitrification rates or intracellular $NO_3^-$ will provide more data for better estimates of total foraminiferal budgets within the nitrogen cycle. Of course, it is not possible to strictly define, which foraminiferal species are able to denitrify or to store $NO_3^-$ without real measurements. If foraminiferal species inhabit $O_2$ depleted environments and belongs to a genus of the species, listed in tab.1 or tab.2, as a rule of thumb, they are good candidates for potential denitrifiers. In addition, if a species is known to inhabit well oxygenated environments and/or belongs to a genus of the species shown in tab.3 it should be avoided to use equations presented below to estimate $NO_3^-$ storage or denitrification rates."

**Reviewer:** The section on kleptoplasty is mostly about species living in shallow, aerated environments— if there is proof of these species inhabiting anoxic conditions, that should be presented in this Review. It should be noted that the experimental conditions used by LeKieffre et al. (2018) were aerated (lines 223-225) so discussing this paper in the context of uptake during anoxia is misleading. Further, Jauffrais et al. (2019) did not incubate in anoxia either (Lines 245-246).

**Reply:** The reviewer is right, that a part of this section concerns kleptoplastic foraminifera from aerated environments and that some of the discussed experiments were done under aerated conditions. This is necessary, though, since kleptoplasty is a complex topic and, especially the kleptoplasty by the species from aphotic anoxic zones is not well understood, yet. Therefore, it is a good idea to discuss, what we know about kleptoplasty (even if it is from aerated environments), before we switch to the things, we don´t know, yet. Also I already had some text in the original manuscript, where I stated that at least the intertidal kleptoplastic species, which usually live in oxygenated environments, sometimes are buried in the deeper anoxic layers in the sediment. This is quite common for *Haynesina germanica* but also for some *Elphidium* species. For example, I am working on samples from a hypoxic Canadian Fjord basin at the moment (unpublished data, that is not discussed in this manuscript). The only two species, present in these sediments are *Stainforthia fusiformis* (denitrifier) and *Elphidium albiumbilicatum* (no nitrate storage but kleptoplastic). I extended the section about the periodic exposure of intertidal species to O2 depleted condition in the revised MS to address this concern by the reviewer:

"Intertidal foraminifera are often exposed to $O_2$ depleted or even anoxic conditions, when water stagnates during low tide or if they are transported to deeper anoxic sediment layers by bioturbation (Rybarczyk et al., 1996; Cesbron et al., 2017). Oxygen penetration depths in tidal flats can vary

between a few mm during low tide to several cm during high tide (Jansen et al., 2009). Thus, intertidal foraminifera are often exposed to anoxia, even within the first cm of the sediment column. *H. germanica* is also supposed to occur in black sediments of the British salt marsh tide pools (Bernhard and Bowser, 1999), which likely become anoxic during a tidal cycle (Rybarczyk et al., 1996) and it was among the first recolonizers of a Fjord suffering of organic pollution (Cato et al., 1980; Bernhard and Bowser, 1999)."

To address the concern by the reviewer that LeKieffre et al. (2018) and Jauffrais et al. (2019) did not incubate under anoxia, I adapted these sections slightly. For the LeKieffre study, I added that it has been done under aerated conditions:

"Recently, LeKieffre et al. (2018) showed in **(aerated)** incubation experiments with $H^{13}CO_3^-$ and $^{15}NH_4^+$ during a light/dark cycle that *Haynesina germanica* is indeed able to fix inorganic carbon and nitrogen under light exposure."

Regarding Jauffrais et al. (2019), I intended to cite their discussion, where they speculate that kleptoplasty in photic environments might also be an adaptation to periodic $O_2$ depletion. They have a whole paragraph about this hypothesis in the discussion of their paper. I did not want to discuss their (aerated) experiment in my sentence. To make this clear I adapted the sentence in the revised manuscript accordingly:

"Kleptoplasty might thus be an additional adaptation of foraminifera from photic environments to stay active during periods of $O_2$ depletion, **which already has been hypothesized by Cesbron et al., 2017**."

**Reviewer:** The paragraph about kleptoplastidic foraminifers from aphotic anoxic zones are relevant to this Review as two species inhabit anoxic conditions: Nonionella stella from Santa Barbara Basin and Virgulinella fragilis from a few habitats (Venezuela, Japan, Namibia; unfortunately the New Zealand population has become decimated as eutrophication remediation has progressed). It is surprising that V. fragilis has not been discussed at all in this Review.

**Reply:** *Virgulina fragilis* is now immediately mentioned in the first sentence of this paragraph:

"Less well understood is the phenomenon of kleptoplasty, observed in the benthic foraminifers *Nonionella stella, Virgulina fragilis* and *Nonionellina labradorica* that can thrive below the photic zone and often inhabit $O_2$ -depleted sediments (Cedhagen, 1991; Bernhard and Bowser, 1999; Grzymski et al., 2002; Bernhard, 2003; Tsuchiya et al., 2015; Jauffrais et al., 2019; Gomaa et al., 2021; Powers et al., 2022)."

I don´t see, why eutrophication remediation is a bad thing.

**Reviewer:** Why is Figure 2 a figure? It is merely an equation and should be included as such.

**Reply:** Figure 2 is now included into the paper as an equation and not as figure anymore.

**Reviewer:** Figure 3 shows a common biogeochemical pathway, denitrification (do a Google search!)—why include it here as a figure? Further, given Orsi et al. and Gomaa et al. also documented this pathway for other foraminifera communities/species, why are these not cited as well as the Woehle et al. papers? If this figure is similar to a previously published image, copyright permission may be required.

**Reply:** Figure 3 not only shows the common denitrification pathway but also includes the alternate oxygenic pathway that might be catalyzed by the nitric oxide dismutase (Nod). Since it is a biogeochemical pathway, as the reviewer states in the first sentence, it cannot be copyrighted. I actually don´t even know a publication with a similar depiction of this pathway. The image in the original figure by Woehle & Roy et al. (2018) is completely different and neither Orsi et al. (2020) or Gomaa et al. (2021) use a similar depiction. Woehle & Roy et al. (2018) was mainly cited here since it was the first paper that described the possible alternative oxygenic denitrification. I have no objections against adding Orsi et al. (2020) and Gomaa et al. (2021) that later confirmed the presence of these enzymes and will cite them in this figure, too.

**Reviewer:** The point of Fig, 6 is not clear.  For example, bacterivory by benthic foraminifera was established in the 1960s (work of JJ Lee and colleagues), so this is nothing new.  Foraminiferal preying on metazoans also is not new—see, e.g., Alan Be papers (1980s), Bowser on carnivory by Antarctic foraminifera (1990s), etc.  If insist on only citing Dupuy then must add "e.g.," as it is only an example.  Term "bolivinids" should be capitalized without italics; same for "globobuliminids".

**Reply:** The whole "ecology" section has been rewritten. According to the constructive feedback by another reviewer (Andrew Gooday). The name of the section has been changed to "Trophic interactions in $O_2$ depleted environments". For the detailed changes in the text of this section see response letter to Andrew Gooday. Also fig.6 has been adapted to the new text in this section. It will provide schematic depictions of the different trophic strategies of foraminifera from $O_2$ depleted environments such as selective herbivory, seasonal herbivory and detrivory. The depiction of phagocytosis will be kept in this figure, because it is important for all these strategies and certainly present in anoxia, as shown by Orsi et al., 2020. I will swap the text "prey" with "food" to clarify, that not only prey organisms can be vacuolized. I am not completely opposed to remove the depiction of the foraminifer preying on nematodes but I would prefer to keep it for the sake of completeness.

**Reviewer:** Fig. 7 presents a subset of data, only those that fit some threshold.  The important question is What does this regression look like when including the data that was excluded??  Remove one of the "from" in "from from" of the caption.  Figure captions are generally far too long.

**Reply:** The threshold was chosen to exclude foraminifera that do not store nitrate in their cells. In my opinion it does not make too much sense to include species that don´t store nitrate to show the correlation between cell volume and nitrate storage in foraminifera that store nitrate. Nevertheless, only a total of six datapoints have been excluded in the original regression. Most of the time there is no cell volume given in literature for species with zero nitrate storage (see for example Pina Ochoa et al., 2010; PNAS). Nevertheless, I did an additional regression, where I included the 6 datapoints that fell below the threshold (<1 mM). All these datapoints included foraminifera with a nitrate content of less than 10 pmol/cell, which is very low and in my opinion not related to nitrate storage for denitrificatio. The regression itself is still significant and looks very similar but the $R^2$ is a bit lower. A comparison of both regressions is shown below. I would prefer to keep the original figure in the manuscript. Figure captions, especially the one concerning the microhabitats are generally shortened now.

[Figure]

Above: Left: Original figure that excluded 6 data-points, which fell below the threshold (<1 mM). Right: Regression that includes all available datapoints, including foraminifera with intracelular nitrate concentrations of < 1 mM.

**Grammatical and other minor specifics:**

The first person singular ("I") is used in many places (e.g., lines 43, 46, 51, 52), which is unconventional for scientific literature, which typically is in the third person plural.

**Reviewer:** Line 11: "artificial fertilizer" is a peculiar phrase. Most would say "eutrophication".

**Reply:** Done.

**Reviewer:** Line 15 notes that certain foraminifera are "unique amongst eukaryotes" without elaboration of details.  Again, this is an overstatement without supporting details. Such phraseology must be used with caution.

**Reply:** Good point. "unique" has been changed to "rare".

**Reviewer:** What is "heterotrophic denitrification"?  First use: line 17.

**Reply:** Heterotrophic denitrification is described in detail within the text and even in two figures (2&3). Line 17 is within the abstract. The abstract would be a bit too long, of every used term would be explained in detail. Otherwise I would also have to explain terms like "anaerobic metabolism", "kleptoplasty", "dormancy" or "phagocytosis" that are all mentioned in the abstract.

**Reviewer:** Line 34 would read better if it was changed to "Nevertheless, much has changed in our perspective about…" (less "wordy").

**Reply:** Another reviewer suggested to change this sentence as well and merge it with the next sentence to assure that this part is not too wordy. This sentence has been changed to:

"Nevertheless, advances in methods to analyze the metabolic rates, intracellular nitrate storage and molecular genetics of foraminifera have changed our understanding of strategies such as anaerobic metabolism that help them to withstand $O_2$ depletion."

**Reviewer:** Line 50 equates organic matter to food. This is an arguable point as organic matter may be refactory, etc and otherwise not necessarily "food".

**Reply:** This is a good point. To avoid any misunderstanding, I changed the sentence to:

"…non refractory organic matter that can be used as food."

**Reviewer:** Line 51 notes that there is scarce knowledge of the ecological interactions of benthic foraminifera from low oxygen to anoxic habitats. This, simply, is not true. Indeed, in Section 3, (lines 285-310) the authors notes that there is much to review on the ecology (although this discussion is about trophics).

**Reply:** The whole "ecology" section has been rewritten. According to the constructive feedback by another reviewer (Andrew Gooday). The name of the section has been changed to "Trophic interactions in $O_2$ depleted environments". For the detailed changes in the text of this section see response letter to Andrew Gooday.

**Reviewer:** Line 52 would read better as "…role of foraminifera in marine biogeochemical…"

**Reply:** Done.

**Reviewer:** The statement spanning lines 58-59 is redundant with the section header, so it should be omitted.

**Reply:** Done. This statement has been deleted.

**Reviewer:** Line 61 would be better as "metabolism of benthic foraminifera."

**Reply:** According to the suggestion of another reviewer this sentence has been deleted.

**Reviewer:** Line 76 mentioned gromids, which are not foraminifera. If discussion of this group remains, the title much include this taxon name.

**Reply:** See my answer from above:

"I am not opposed to add the *"Gromia"* as well but the title is already quite long and gromiids are only a minor part of this review. Nevertheless, I clarified now specifically in the text that gromiids aren´t foraminifera, when gromiids are first mentioned in the text:

"Other studies showed that bacterial endobionts likely perform denitrification in some allogromiid foraminifera and gromiid species (Bernhard et al., 2012a; Høgslund et al., 2017). Gromiida are a separate group of protists within the Rhizaria and closely related to foraminifera."

In addition, I separated foraminifera and gromiids in table 2."

**Reviewer:** Line 77 states the presence or size of the intracellular nitrate, but nitrate cannot be "sized". Better alternatives include concentration, amount or magnitude.

**Reply:** Changed "size" to "magnitude".

**Reviewer:** Line 92 cites Bernhard et al. 2012a when Bernhard et al. 2012b should be cited.

**Reply:** Changed accordingly.

**Reviewer:** Line 94 incorrectly infers that bacteria are "killed" by antibiotics, when in reality bacterial activities are inhibited when exposed to the appropriate antibiotics.

**Reply:** This sentence has been changed accordingly:

"...when the activity of potential bacterial symbionts would be inhibited."

**Reviewer:** Line 108 states that genomes were obtained by Gomaa et al. (2021) but this is not the case and may not be the case in Orsi et al. (2020) either.

**Reply:** This is right. "genomes" has been changed to "transcriptomes".

**Reviewer:** Why is a publication listed with two author names plus "et al." (line 113)? There is no need of this given there are no other 2022 papers with Woehle and Roy as the first two authors.

**Reply:** Both papers Woehle & Roy et al. 2018 & 2022 were papers with shared first authorships. I wanted to do the "second-first author" justice, because otherwise, mainly the first name will be shown as citation in other papers. Unfortunately, this was not uniform along my manuscript, due to the use of a reference manager. I corrected this and now use "Woehle & Roy et al." all along the manuscript.

**Reviewer:** Line 122 requires a literature citation(s) for the sentence ending "…strong reactivity".

**Reply:** According to another reviewer (Frans Jorissen) the text was going to fast over some statements and he already criticized that I have to extend the statement about the oxygen reactivity. Now this part is longer and more citations have been added:

"Larger amounts of O2 might supply this demand but also harm the cell. For example, O2 can inhibit the growth of some obligate anaerobes poison enzymes that are important for their metabolism (Lu and Imlay, 2021). Also for aerobes O2 can be harmful. "Hyperoxia", an excess supply of O2, leads to damaging effects by highly-reactive metabolic products of O2 (free O2 radicals) that inactivate enzymes in the cell, damage DNA and destroy lipid membranes (Frank and Massaro, 1980). Furthermore, foraminifera can store NO3- within vacuoles, due to its lower reactivity and still have an electron acceptor reservoir if NO3- is depleted in their microhabitat. This is not possible for O2 due to its high reactivity (Auten and Davis, 2009)."

**Reviewer:** Line 133 should remove "denitrifying" as that word appears later in the sentence.

**Reply:** Done.

**Reviewer:** Line 139 should cite the original literature that documented "epifauna, shallow infauna, and deep infauna" foraminifera (Bruce Corliss *Nature* 1985).

**Reply:** I added the following sentence to this section:
"The presence of this species specific microhabitat structure has first been documented by Corliss (1985)."

**Reviewer:** Line 146 and elsewhere, the use of the verb "prefer" (or preference) should be avoided as foraminifers do not have conscious thought.

**Reply:** This is a rather philosophical debate, if a "preference" prerequisites a conscious thought. We all agree that certain foraminifera species have certain food or microhabitat preferences. Some species selectively ingest phytodetritus, if available. If fresh phytodetritus is not available they ingest other food particles. Isn´t this a preference for phytodetritus? Nevertheless, in the revised version of the manuscript, I tried to reduce the usage of "prefer" or "preference".

**Reviewer:** On line 150, the species of Globobulimina that are considered to be deep infuanal / facultative aerobes are not listed.

**Reply:** Added "spp." after *Globobulimina*.

**Reviewer:** Use of "basically" on line 151 is colloquial and superfluous; the term should be omitted.

**Reply:** "basically" has been deleted.

**Reviewer:** The phrase "competitional stress" (line 153) is unconventional. Perhaps the intention was "competitive stress"?

**Reply:** Changed "competitional" to "competitive".

**Reviewer:** The references are often placed at the end of a sentence when the papers do not all show the same thing.  This approach should be avoided. For example, the sentence spanning lines 155-158 should have Schmiedl and Mackensen (2006) cited after "shallow infaunal lifestyle", not at the end as that paper did not assess denitrification or nitrate storage.

**Reply:** I divided the references in this sentence and moved Schmiedl and Mackensen (2006) as suggested by the reviewer.

**Reviewer:** The statement on line 159 is awkward because "used" can mean "utilized", but it seems the intent here is more like "accustomed".  A suggested edit is "that typically occur at the sediment-water interface or on elevated surfaces."

**Reply:** Edited and changed as the reviewer suggested.

**Reviewer:** Line 164 should read "spp. may denitrify under".

Line 167 should read "…if they must, due to their…"

Line 170 should read "research still continues…"

**Reply:** All done.

**Reviewer:** Throughout, "chapters" should be called "sections".

**Reply:** Changed "chapter" to "section" throughout the MS.

**Reviewer:** The sentence spanning lines 169-171 is poorly worded as published papers have already determined denitrification rates, meaning the wording cannot include any literature citations.

**Reply:** This sentence has been changed, according to suggestions of another reviewer: "Research to measure denitrification rates in different benthic foraminiferal species continues (Langlet et al., 2020; Choquel et al., 2021). This will add to the scarce available data and contribute to estimates of the role of foraminifera in oceanic N-cycling

**Reviewer:** The order of subsections in Section 2.1 is illogical as it starts discussing an active process (denitrification), then discusses a inactive process (dormancy), and then discusses an active process again (kleptoplasty).

**Reply:** The order was supposed to be alphabetical (foraminiferal) "denitrification", "dormancy", "kleptoplasty" and "other recent developments…". It was not supposed to be ordered after active and inactive processes. I changed the first header to "denitrification by foraminifera" to make the alphabetical order more accessible.

**Reviewer:** Line 183 should read "well-aerated condition. They interpreted this…" (two changes)

**Reply:** Done.

**Reviewer:** The paragraph spanning lines 189 to 200 consistently italicizes "sp.", which is incorrect.  This should never be italicized. Line 201 italicizes "spp." incorrectly.

**Reply:** Done.

**Reviewer:** Line 197 should read "…acids, which was not the case…"

**Reply:** Done.

**Reviewer:** It Is not clear why paragraph spanning lines 201-209 is included in the dormancy section. If there is a connection to dormancy, this must be explicitly stated.

**Reply:** This paragraph dealt with ultrastructural changes of *Ammonia* sp. exposed to anoxia. A survival strategy by *Ammonia* sp. under exposition to anoxia is dormancy, as discussed in other parts of this section. So some of the documented ultrastructural changes might be related to dormancy. I apologize that this was not clear enough in this paragraph. To address this, I state this directly now in one of the sentences:

"These were interpreted as endobionts but might also be parasites that could not be fended off, due to the drastically reduced metabolism during dormancy under anoxia."

**Reviewer:** Line 206 should read "…could not be fended off" (although this is a rather colloquial statement).  Further, this statement should cite the original publication that showed the transition from endobionts to parasites to necrophagy (i.e., Bernhard et al. 2010 Paleoceanography).

**Reply:** Done. Added the following sentence:

"A continuum of intracellular bacteria including prey in food vacuoles, endobionts, parasites and necrophages has been documented before in benthic foraminifera from cold seeps (Bernhard et al., 2010). It already has been hypothesized by the authors that bacteria switched their function from endobionts to predators, depending on the vitality of the host cell."

**Reviewer:** Line 214 should read "…this research originated in the 1970's (Lopez…"

**Reply:** Done.

**Reviewer:** Line 232 must spell the genera name properly:  *Nonionella,* for *N. stella.* The genus for *N. labradorica* is *Nonionellina* (not the same as *Nonionella*).  Further, Powers et al. (2022) Frontiers in Mar. Sci Sect on Mar Microbiol and Ecol. should be cited in the string of citations on line 234.

**Reply:** Corrected the typos in the genera name and added the citation.

**Reviewer:** The spelling of Grymsky on line 241 must be corrected (it is Grzymski).

**Reply:** Done.

**Reviewer:** Line 242 should read "…calculated that the required amount…"

**Reply:** Done.

**Reviewer:** Line 248 should read "… Gomaa et al. (2021) support observations of Grzymski et al (2002)."

**Reply:** Done.

**Reviewer:** It is surprising that Powers et al. (2022 Frontiers) is not discussed at all in this Review.  '

**Reply:** This paper is now discussed in detail in the revised manuscript:

"Most foraminifera species from $O_2$ depleted habitats possess numerous peroxisomes that are usually associated with mitochondria and the endoplasmatic reticulum (Bernhard and Bowser, 2008). Bernhard and Bowser (2008) hypothesized that these peroxisome proliferations might be used to either metabolize $H_2O_2$ and other  highly reactive oxygen species that are produced within the chemocline close to the oxic/anoxic boundary or to reduce the oxidative stress by these compounds. Indeed, they showed in an experiment that ATP concentrations in foraminifera increased proportional to ambient $H_2O_2$ concentrations. A recent study on transcriptome and metatranscriptome of *N. stella* and *B. argentaea* from the Santa Barbara Basin revealed that these species utilize an adaptable mitochondrial and peroxisomal metabolism, depending on the chemical treatment in the experiment (Powers et al., 2022). The high plasticity of their peroxisomal and mitochondrial metabolism might be substantial for survival at the highly variable conditions at the chemocline in the sediments. The results by Powers et al. (2022) indicate that at least some processes that are involved in foraminiferal denitrification are associated with mitochondria. Interestingly, the expression of denitrification related genes in both species was upregulated after incubation with elevated $H_2O_2$ but without $NO_3^-$ and downregulated, if they were incubated without $H_2O_2$ but with $NO_3^-$, compared to a control treatment with both $H_2O_2$ and $NO_3^-$. In the same way several peroxisomal processes were upregulated in the $H_2O_2$ only treatment. In addition, despite that both species are able to denitrify, Powers et al. (2022) found distinct metabolic adaptations to anoxia in both species. For example, a quinol:fumarate oxidoreductase, which is considered as an adaptive mechanism for anaerobic respiration in eukaryotic organisms, was present in *N. stella* but not in *B. argentaea*. Vice versa, *B. argentaea* has the capacity to digest food vacuole contents under $O_2$ depletion, while *N. stella* was lacking food vacuoles (Powers et al., 2022)."

**Reviewer:** Line 261 should write out the genus name for *B. argentea* (Bolivina).

**Reply:** Done.

**Reviewer:** Line 268 should read "…observation was made by…"

**Reply:** Done.

**Reviewer:** For line 270, the term "presumably" should be inserted to read "Presumably living foraminifera…".

**Reply:** Done.

**Reviewer:** Line 276 "Though this requires…energy." is not a sentence.  Please rewrite.

**Reply:** This part has been rewritten, due to the suggestion of another reviewer:

"These processes (calcification and the ingestion of prey cells by phagocytosis) require bursts of high energy, which the authors suggest is generated by dephosphorylation of an intracellular creatine phosphate storage to regenerate ATP from ADP."

**Reviewer:** Please correct spelling of "metatranskriptome" on line 279.

**Reply:** Done.

**Reviewer:** Line 281 should read "…found evidence of another anaerobic metabolism…" (as denitrification is an anaerobic metabolism).

**Reply:** Done.

**Reviewer:** Line 282-283 should include Gomaa et al. (2021) as these processes were also documented.

**Reply:** Changed this part to: "Orsi et al. (2020) and Gomaa et al. 2021 also found evidence for another anaerobic metabolism in foraminifera from the Namibian shelf. Their data indicates that the foraminifers metabolize hydrolyzed organics to produce ATP using fermentation and fumarate reduction."

**Reviewer:** Line 287 discusses a study with 60 uM O2, which is very high from many "hypoxia" perspectives.

**Reply:** I am a bit confused, since hypoxia are not mentioned in this context. I only mentioned "oxygen depleted environments" in a sentence before and I guess we can agree that "less than 60 µM" in the bottom water can be considered as oxygen depleted. In addition, the oxygen penetration depth into the sediments at this location is very low. I added a statement about the $O_2$ penetration depth at this location and also stated that many of these observations, especially regarding shallow infauna also apply for foraminifera from well oxygenated environments.

**Reviewer:** Line 292 should read "cancellata and Chilostomella".

**Reply:** Done.

**Reviewer:** Consider using the typical term for deposit feeders (detritivore) on line 293.

**Reply:** Done.

**Reviewer:** Line 294 must include "e.g.," in the beginning of the ctations as there are additional studies of carnivory by foraminifers.

**Reply:** Done.

**Reviewer:** Line 298 uses the term "thrive" (used earlier in the manuscript too) yet there is no documentation that this species is truly abundant in the setting being discussed.

**Reply:** *Globobulimina auriculata* is relatively abundant in the oxygen depleted Alsbäck Deep within the Gullmar Fjord. It shares the habitat with the closely related *Globobulimina turgida*. Both species have a very similar appearance and have thus been lumped in some early studies about foraminifera assemblages in the Gullmar Fjord. Now we have seen the differences between both species in the trancriptome (Woehle & Roy et al., 2018), in their feeding strategies (Glock et al. 2019) and in subtle morphological features, such as appearance and color of the cytoplasm and the shape of the aperture. Nevertheless, since no study is published, yet, about the abundances of both species, I decided to change this sentence accordingly. I avoided "thrives" and wrote:

"The species *G. auriculata* denitrifies and lives under oxygen depleted conditions (Woehle & Roy et al., 2018)."

**Reviewer:** "Predator prey" must be hyphenated on line 310.

**Reply:** Done.

**Reviewer:** The sentence spanning lines 312-313 should be omitted as it is highly redundant.

The use of "not exotic" on line 314 requires explanation. Exotic to what?

**Reply:** Done. Both sentences have been deleted, due to being redundant.

**Reviewer:** Line 316 should use either "established" or "suggested" instead of "pointed out" which is colloquial. Consider replacing one of the "some" on line 317 ("In some environments, such as some habitats…" (possible replacements are "certain", "selected").

**Reply:** Done.

**Reviewer:** It would be good to add other publications on line 318 that have calculated the contribution of foraminifera to total denitrification such as Choquel et al., Glud et al., etc. While it is understood that the author is promoting his publications, it gets a bit much sometimes.

**Reply:** Done. I added both references. Glud et al. showed that foraminifera have only a minor contribution to the nitrogen cycle at Sagami Bay but they also showed that it is significant. So I guess this paper can be cited here. I apologize that I forgot to cite Choquel et al. 2021. It was not my intention to only promote my own publications.

**Reviewer:** "keyplayers" is two words (line 318).

**Reply:** Done.

**Reviewer:** The text spanning lines 327 to 330 should explain why diagenetic models are being discussed. Most people will be interested in biogeochemical modeling vs diagenetic models.

**Reply:** I don´t really understand the statement regarding biogeochemical modeling vs diagenetic modeling. I mean, if redox processes that happen in the upper sediment column during early diagenesis are modelled, using a diagenetic model, this is also biogeochemical modeling, right? Why should we make a difference here. The "diagenetic model" is mainly discussed, because the model used by Dale et al. models early diagenesis by including biological nitrate transport, instead of just using a diffusion-reaction model. To assure that the "diagentic model" part regards the Dale et al. study I added "other" before diagenetic models:

"The $NO_3^-$ storage in denitrifying foraminifera, but also in some sulfur bacteria, such as *Beggiatoa*, is of greater importance for benthic biogeochemical cycling, due to the potential of biological transport of these intracellular reservoirs (Dale et al., 2016). Most of the **other** diagenetic models that describe and calculate benthic N-cycling…"

**Reviewer:** Line 337: change capitalized "D" in "Data" to lower case.

**Reply:** Done.

**Reviewer:** The observation that foraminifera have served as a nucleating site for phosphorites (lines 348 to 350) does not necessarily indicate their active use of precursors as there is known presence of other mineral associations in foraminifer tests (e.g., pyrite framboids).

**Reply:** This section might indeed have been a bit too short and some important information was missing. There have been several studies that showed that the large intracellular polyphosphate enrichments in sulfur bacteria can facilitate phosphorite formation at the upper boundary of the Peruvian OMZ that is rich in bacterial mats and phosphorite deposits. The phosphorite deposits at the lower boundary of the Peruvian OMZ are different, because these bacterial mats are usually not present and the phosphorite grains have a similar size and shape of foraminifera. These phosphorite grains are also abundant in the surface fraction of the sediments. In the Peruvian OMZ, living foraminifera abundances are very high. The sediments are thus a mixture of living forams, phosphorite grains with a coarse shape and size of a foram and every intermediate step in between. It is likely that their high intracellular phosphate storage, together with the calcium storage to precipitate their tests results in a supersaturated apatite microenvironment within their shells and initiates apatite formation. This has been also suggested for other organisms before. All this is discussed in detail in my 2020 GCA paper ("A hidden sedimentary phosphate pool inside benthic foraminifera from the Peruvian upwelling region might nucleate phosphogenesis"). To address these issues and avoid further misunderstandings, I extended the text in this section:

"In addition, there is evidence that the intracellular phosphate storage in foraminifera facilitates phosphogenesis in some environments, similar to the intracellular polyphosphate enrichments in some sulfur bacteria (Schulz and Schulz, 2005). The release of phosphate after breakdown of these polyphosphates to harvest energy in times of electron acceptor depletion results in apatite supersaturation and initiates phosphogenesis (Schulz and Schulz, 2005). Sediments at the lower boundary of the Peruvian OMZ contain many small phosphorite grains with similar size and shape of foraminifera (Manheim et al., 1975; Glock et al., 2020). The sand fraction of the surface sediments in this region is a mixture of pristine living foraminifer shells with dead tests that show a transition from shells that are filled with phosphorites until small phosphorite grains that only retain the size and coarse shape of a foraminifer. It is likely that a post mortem release of the intracellular phosphate storage results in a supersaturated microenvironment within the shells that initiates apatite

formation (Glock et al., 2020) in a similar way as it has been suggested for other organisms (Kulakovskaya, 2014)."

**Reviewer:** The single sentence spanning lines 356 to 361 must be broken into two or three sentences.

**Reply:** Done. Divided into two sentences.

**Reviewer:** In section 4.1, the author does not address the very large variability of vacuole volume in cytoplasm from varied foraminifers (see Fig 2 in LeKieffre et a 2018 Mar Micropaleo).

**Reply:** I don´t think the vacuole volume has too much to do with the intracellular nutrient storage but rather the nitrate concentration in the vacuoles. Vacuoles are also present in foraminifera that do not store nitrate. In addition, vacuoles can have a variety of functions and most are not related to nitrate storage. For example: Some vacuoles are related to calcification and others to digestions. In addition, not all structures that are considered as vacuoles are indeed seawater vacuoles but vesicles with various functions. For example, many of the small vesicles that are observed with TEM in for example Ammonia spp. are indeed acidocalcisomes that lost their content during preparation for TEM.

**Reviewer:** The Section on paleoceanography (Sect 4) needs to be shortened as an "in prep" manuscript (Hoogakker et al.) is cited in most sentences (appears at least 5 times over ~75 lines). Such passages are irritating—just leave that information to the Hoogakker team to present. Further, most of this discussion is about environments with oxygen, which is not anoxia, impacting the decision to change the title or focus of the Review, as discussed earlier in this critique.

**Reply:** The paleo-part will be deleted and only briefly addressed in the introduction. So many of the remarks below (from line 399 to 450) will not be relevant anymore and not being replied in detail.

**Reviewer:** Do not italicize family names (line 399) and do not italicize "spp." (line 397 and as noted earlier).

Line 399 is not a sentence. Please rewrite.

The format "C. spp." is not allowed (line 416). Here, the genus name must be fully written.

Line 424: is 11 years still considered "recent"?

Aside from citing Erez, newer references should be cited regarding foraminiferal calcification mechanisms (e.g., works by Toyofuku and/or de Nooijer).

The list of 22 publications in one sentence (lines 432-435) is excessive and not the approach used in other places earlier in the manuscript.

Statement on line 439 should read "This offset is referred to as…" because that notation was used in the McCorkle paper from decades ago—the terminology is not new. Line 441 should cite McCorkle papers also, not solely the new publications.

"Height difference" (line 442) is a peculiar way to discuss what I believe the author intends, which is depth of calcification.

Line 445 should present values for "lower [O2] range".

Line 454 should read "…this index are ongoing , with recent developments… Tetard et al. (2021) and Kranner et al. (2022)."  (five changes)

Line 450 uses the phrase "appears obvious" which is rather rude.  Hindsight is 20-20 vision, correct?  Meaning of course things seem obvious now but no one / few thought foraminifera could be anaerobes back then.  The author is urged to rewrite this.

**Reply:** As mentioned above, the paleo-part will be deleted and only briefly be addressed in the introduction. So many of the remarks above (from line 399 to 450) will not be relevant anymore and not being replied in detail.

**Reviewer:** Line 464 should be singular as there is only one author.

**Reply:** Done.

**Reviewer:** The proper way to cite one's grant in Acks (line 467 is "…(DFG) through Heisenberg grant GL 999/3-1 to N.G."  (Initials can be used if the grant recipient is an author of the manuscript).

**Reply:** Done.

**Reviewer:** Species names must be italicized in the cited literature section.

Journal name is missing from van Dijk et al. 2019 (and most would alphabetize that under V).  Same for de Frietas (should appear in the D's).

Proper citation for Glock et al 2012b is:

Glock, N., Schönfeld, J., Mallon, J. (2012). The Functionality of Pores in Benthic Foraminifera in View of Bottom Water Oxygenation: A Review. In: Altenbach, A., Bernhard, J., Seckbach, J. (eds) Anoxia. Cellular Origin, Life in Extreme Habitats and Astrobiology, vol 21. Springer, Dordrecht. https://doi.org/10.1007/978-94-007-1896-8_28

Line 583: do not capitalize "auriculata"

Why is Glock et al. (2019) PNAS repeated twice in the Literature cited section (2019a and 2019b)?

The citation for Grzymski et al. (2002) has added seven extraneous authors.  Why?  The author list is Grzymski, Schofield, Falkowski and Bernhard.

The "o" in Jorgensen has a slash through it (Scandinavian letter).

Line 849 should read "pH", not "PH".

**Reply:** I thank the reviewer for this thorough check of the reference list. All references will be corrected carefully in the revised manuscript.

**Reviewer:** Names in Table 1 must be alphabetized.  Terms must be defined ("specific denitrification" is truly a rate presented on a per volume basis).

**Reply:** Done. Terms are defined as follows: "Individual denitrification rates refer to average rates per individual while specific denitrification rates refer to rates normalized to the biovolume of the foraminifers."

**Reviewer:** Table 2 should remove gromiida unless the title changes.  And, proper: Gromiida, or "gromids"

**Reply:** Done.

**Reviewer:** Italicize 'H' in Haynesina in Table 3.  "Labrospira cf. subglobosa" should read "Labrospira cf *L.* subglobosa" (add first letter of genus name  italicized and without italics for "cf").

**Reply:** Done.